# Scaf-GRPO: Scaffolded Group Relative Policy Optimization for Enhancing LLM Reasoning

**Xichen Zhang[1],\*, Sitong Wu[2],\*, Yinghao Zhu[3], Haoru Tan[3], Shaozuo Yu[2], Ziyi He[3] & Jiaya Jia[1],†**

[1]The Hong Kong University of Science and Technology
[2]The Chinese University of Hong Kong
[3]The University of Hong Kong

## Abstract

Reinforcement learning from verifiable rewards has emerged as a powerful technique for enhancing the complex reasoning abilities of Large Language Models (LLMs). However, these methods are fundamentally constrained by the "learning cliff" phenomenon: when faced with problems far beyond their current capabilities, models consistently fail, yielding a persistent zero-reward signal. In policy optimization algorithms like GRPO, this collapses the advantage calculation to zero, rendering these difficult problems invisible to the learning gradient and stalling progress. To overcome this, we introduce Scaf-GRPO (Scaffolded Group Relative Policy Optimization), a progressive training framework that strategically provides minimal guidance only when a model's independent learning has plateaued. The framework first diagnoses learning stagnation and then intervenes by injecting tiered in-prompt hints, ranging from abstract concepts to concrete steps, enabling the model to construct a valid solution by itself. Extensive experiments on challenging mathematics benchmarks demonstrate Scaf-GRPO's effectiveness, boosting the pass@1 score of the Qwen2.5-Math-7B model on the AIME24 benchmark by a relative 44.3% over a vanilla GRPO baseline. This result demonstrates our framework provides a robust and effective methodology for unlocking a model's ability to solve problems previously beyond its reach, a critical step towards extending the frontier of autonomous reasoning in LLM.

## 1 Introduction

Large Language Models (LLMs) have demonstrated remarkable capabilities in complex reasoning tasks across diverse domains such as mathematics, programming, and logic (Guo et al., 2025; Jaech et al., 2024; Muennighoff et al., 2025; Min et al., 2024). A key driver of these advancements is Reinforcement Learning from Verifier Rewards (RLVR) (Guo et al., 2025; Zeng et al., 2025; Liu et al., 2025b), a paradigm where models learn to generate sophisticated reasoning paths by exploring diverse strategies and receiving feedback on their final outcomes. This approach eliminates the need for expensive, step-by-step human annotations by rewarding only the final correct answer, enabling models to autonomously discover effective problem-solving procedures (Guo et al., 2025).

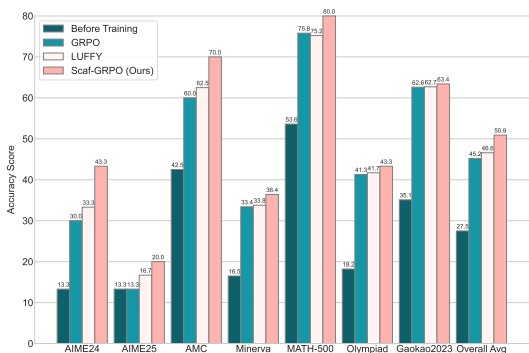

Figure 1: Scaf-GRPO overcomes the learning cliff with minimal guidance, outperforming vanilla GRPO (Shao et al., 2024) and the prefix-based LUFFY (Yan et al., 2025) across challenging math benchmarks on Qwen2.5-Math-7B. By injecting strategic, hierarchical hints, our method unlocks the model's potential on difficult problems, achieving superior overall performance.

However, the efficacy of RLVR is severely constrained by a fundamental challenge we formalize as the "learning cliff." This phenomenon occurs when a model confronts a subset of problems that lie significantly beyond its current capabilities. For these problems, all exploratory attempts

---

\*Equal contribution
†Corresponding author: Jiaya Jia (jia@cse.ust.hk)

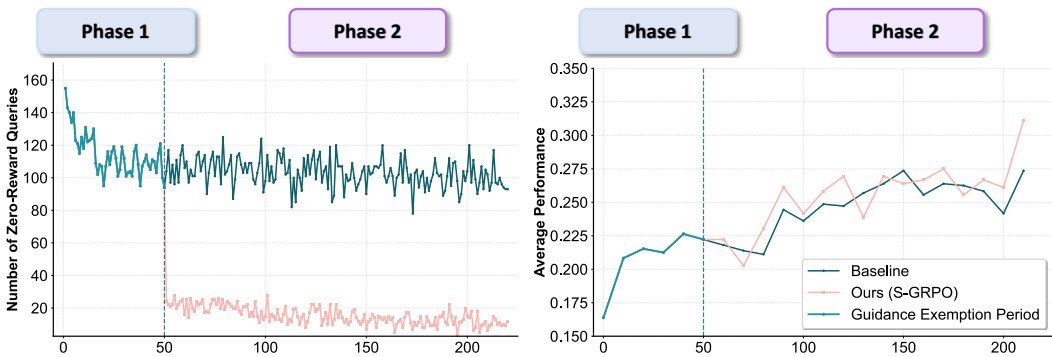

Figure 2: Training dynamics of Qwen2.5-Math-1.5B. (a) Scaf-GRPO overcomes the learning cliff by continuously solving zero-reward problems where vanilla GRPO plateaus. (b) This translates to sustained and superior validation accuracy for Scaf-GRPO throughout training.

consistently fail, leading to two critical and cascading consequences: (1) Reward Signal Loss: The model receives a persistent zero-reward signal for this entire subset of problems. (2) Vanishing Gradients: In algorithms like GRPO (Shao et al., 2024), the advantage signal provides the learning gradient. When all rewards are zero, the advantage collapses to zero, providing no gradient for the policy to learn from (Yu et al., 2025).

Consequently, these difficult problems become "invisible" to the policy update. As our empirical analysis in Figure 2 illustrates, these problems form a persistent "long tail" of challenges that the model cannot conquer autonomously. This long tail represents a critical bottleneck, as it prevents the model from leveraging the most difficult examples to achieve a higher level of competence.

To address the learning cliff, a prevailing strategy has emerged: incorporating off-policy guidance from a more capable "teacher" policy (Yan et al., 2025; Huang et al., 2025; Zhang et al., 2025a;b). These methods typically work by providing the student model with a prefix of a correct "golden" solution and tasking it with generating the remainder of the reasoning path. While this ensures a positive reward signal, this prefix-continuation paradigm introduces significant issues. It creates a distributional mismatch between the teacher-generated prefix and the student-generated suffix, necessitating complex algorithmic corrections like policy shaping (Yan et al., 2025) or hybrid SFT-RL objectives (Huang et al., 2025) that can introduce bias and training instability. More critically, this "on-rails" guidance forces the model down a predetermined path, stifling its ability to explore alternative, potentially more novel or efficient, reasoning strategies.

To address this challenge, we propose Scaf-GRPO (Scaffolded Group Relative Policy Optimization). Our framework is inspired by the pedagogical theory of Scaffolding (Berk & Winsler, 1995), a teaching method of providing temporary support that fades as learners improve. We apply this principle by providing hierarchical, minimal, progressive assistance to help the model bridge its capability gaps, rather than enforcing a rigid solution prefix. This in-prompt scaffolding approach is guided by two primary objectives: first, to maintain policy consistency by having the model process both the problem and the hint under a single, unified policy, thereby avoiding the distributional mismatches of prefix-based methods. Second, to preserve exploration flexibility, as our hints act as "signposts" rather than "railroads," guiding the model without fixing its path and allowing it to discover its own unique solution strategies.

Our framework operates in two carefully designed phases. It first employs a guidance exemption period to distinguish "true-hard" problems from "pseudo-hard" ones that the model can solve on its own with more training. For true-hard problems, it then activates hierarchical hint-guided exploration, providing progressively concrete hints (from abstract concepts to concrete steps) until the model can generate a correct solution. By rewarding the model for succeeding with the most abstract hint possible, Scaf-GRPO encourages the internalization of reasoning skills rather than the memorization of solutions. Our contributions are as follows:

- We propose Scaf-GRPO, a novel training framework inspired by pedagogical scaffolding addressing the "learning cliff" issue in RLVR. It provides hierarchical, minimal, and progressive guidance via in-prompt hints instead of fixed solution prefixes. This approach maintains policy consistency while preserving the model's exploratory autonomy, thereby overcoming the key limitations of existing guidance methods.

- We demonstrate the effectiveness of Scaf-GRPO through extensive experiments on several challenging mathematics benchmarks. On the Qwen2.5-Math-7B model, our method achieves a significant relative improvement of 12.6% over the vanilla GRPO baseline and a 9.2% relative gain over strong prefix-based guidance methods like LUFFY.

- We demonstrate the broad applicability and robustness of Scaf-GRPO across diverse models. Our experiments show consistent performance gains on different architectures (Qwen, Llama), scales (1.5B to 7B), and specializations (math-tuned, instruction-tuned, and Long-Chain-of-Thought), establishing Scaf-GRPO as a versatile and model-agnostic framework enhancing LLM reasoning.

## 2 RELATED WORK

**Reinforcement learning from verifier reward.** The success of DeepSeek-R1 (Guo et al., 2025) establishes Reinforcement Learning from Verifier Reward (RLVR) as a paradigm for enhancing the reasoning capabilities of Large Language Models (LLMs). In RLVR, models are trained using feedback from an external verifier that provides an outcome-based reward (e.g., correct/incorrect) for a generated solution. The success of DeepSeek-R1 (Guo et al., 2025) demonstrates that even with sparse, binary rewards, models can learn reasoning strategies. Subsequent research has built upon this foundation, focusing on enhancing algorithmic stability through debiasing techniques (Liu et al., 2025b; Yu et al., 2025), or designing more informative rewards to improve sample efficiency, such as using length penalties to mitigate overthinking (Aggarwal & Welleck, 2025; Jin et al., 2025; Chen et al., 2024) or token-level signals to provide denser feedback (Wang et al., 2025b;a).

**RLVR with off-policy guidance.** To overcome the learning cliff, a phenomenon where a persistent lack of positive rewards renders difficult problems invisible to the learning gradient (Yu et al., 2025), researchers incorporate guidance from a "teacher" policy. The prevailing strategy is to provide the student model with a prefix of a "golden" trajectory and task it with generating the continuation (Yan et al., 2025; Huang et al., 2025; Zhang et al., 2025a;b; Ma et al., 2025). Different methods introduce variations on this theme. For instance, Yan et al. (2025) mix a complete expert trajectory with multiple model-generated rollouts in one batch. Huang et al. (2025) employ a cosine decay schedule to adjust the length of the guiding prefix. More recently, Zhang et al. (2025a) provide multi-level hints of varying lengths, allowing the model to explore from multiple starting points. However, this prefix-continuation paradigm introduces challenges. It breaks policy consistency by mixing trajectories from two different distributions, which necessitates complex algorithmic patches (Yan et al., 2025; Huang et al., 2025; Zhang et al., 2025a). Furthermore, forcing the model down a predetermined path stifles exploration, limiting its ability to discover novel reasoning strategies. Our work provides effective guidance while circumventing these issues.

## 3 METHODOLOGY

Our framework, **Scaffolded Group Relative Policy Optimization (Scaf-GRPO)**, illustrated in Figure 3, overcomes the learning cliff inherent in reinforcement learning by providing hierarchical, minimal, and progressive guidance. Unlike methods that alter the fundamental RL objective with off-policy data, Scaf-GRPO maintains the on-policy nature of GRPO. It operates by strategically augmenting the model's rollout buffer when learning stagnates, ensuring that the learning signal is both meaningful and derived from the most efficient reasoning path the model can achieve with assistance. Our framework operates in two phases: an initial guidance exemption phase to diagnose "true-hard" problems, and a subsequent cyclical phase of hierarchical hint-guided exploration.

### 3.1 PRELIMINARIES: GROUP RELATIVE POLICY OPTIMIZATION (GRPO)

Group Relative Policy Optimization (GRPO) (Shao et al., 2024) is an on-policy RL algorithm for training LLMs that eliminates the need for a trainable value function. For a given prompt $q$, the policy $\pi_\theta$ generates a group of $N$ trajectories, $\mathcal{G} = \{o_1, \ldots, o_N\}$. After obtaining a terminal reward $R(o_i)$ for each trajectory from a verifier, GRPO computes a normalized advantage $\hat{A}_i$ as: $\hat{A}_i = \frac{R(o_i) - \mu_\mathcal{G}}{\sigma_\mathcal{G} + \epsilon_{\text{std}}}$, where $\mu_\mathcal{G}$ and $\sigma_\mathcal{G}$ are the mean and standard deviation of rewards in the group $\mathcal{G}$, and $\epsilon_{\text{std}}$ is a small constant for numerical stability. The policy is then updated by maximizing a clipped

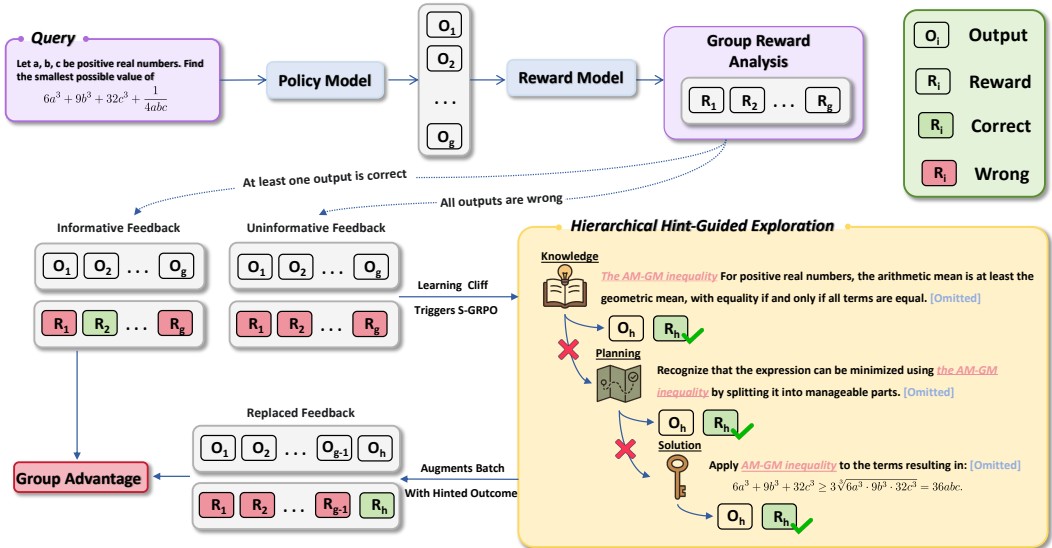

Figure 3: Overview of the Scaf-GRPO framework. For a given query, the model generates multiple solutions. (Left) If any solution is correct, standard GRPO proceeds. (Right) If all solutions fail (the learning cliff), Scaf-GRPO initiates hierarchical hint-guided exploration. It injects progressively concrete in-prompt hints until a correct solution is found. This successful, minimally-guided trajectory replaces a failed one, restoring the learning gradient and enabling on-policy updates to resume.

surrogate objective:

$$J_{\text{GRPO}}(\theta) = \hat{\mathbb{E}}_{i,t}\left[\min\left(r_{i,t}(\theta)\hat{A}_i,\ \text{clip}(r_{i,t}(\theta), 1-\epsilon, 1+\epsilon)\hat{A}_i\right)\right], \tag{1}$$

where $r_{i,t}(\theta) = \frac{\pi_\theta(o_{i,t}|o_{i,<t},q)}{\pi_{\theta_{\text{old}}}(o_{i,t}|o_{i,<t},q)}$ is the probability ratio between the current and old policies, and $\epsilon$ is the clipping hyperparameter. The key limitation arises when all trajectories in $\mathcal{G}$ receive a zero reward, causing $\hat{A}_i$ to collapse to zero and stalling the learning process—the learning cliff.

## 3.2 THE SCAF-GRPO FRAMEWORK

Scaf-GRPO modifies the training process by strategically augmenting the trajectory group $\mathcal{G}$ when a learning cliff is detected. The process consists of a conditional batch construction procedure followed by the application of the standard GRPO loss.

**Phase 1: Diagnosing true-hard problems.** A key principle of effective teaching is to avoid providing help when a learner can succeed independently. Not all initial failures indicate a fundamental capability gap; many are what we term pseudo-hard samples, arising from unfamiliarity with output formats or nascent reasoning skills. To address this, Scaf-GRPO incorporates a guidance exemption period, empirically set to the initial 15% of training steps. During this phase, the model attempts solutions purely through on-policy exploration. As shown in Figure 2, this period is characterized by a rapid decrease in zero-reward queries. We algorithmically determine when this independent learning has plateaued by monitoring the rate of solving zero-reward queries. Once this rate stagnates, any problem the model still consistently fails is classified as "true-hard," making it a candidate for guidance. This ensures hints are reserved for genuine learning cliffs.

**Phase 2: Hierarchical hint-guided exploration.** Once a problem is identified as "true-hard," Scaf-GRPO activates its guidance mechanism using a pre-defined, three-tiered hint hierarchy, $H = \{H_{\text{knowledge}}, H_{\text{planning}}, H_{\text{solution}}\}$. The tiers offer distinct levels of guidance: (1) $H_{\textbf{knowledge}}$ (Knowledge Hint): Points to the key concept or formula required. (2) $H_{\textbf{planning}}$ (Planning Hint): Outlines a high-level strategic framework for the solution. (3) $H_{\textbf{solution}}$ (Solution Hint): Provides a concrete calculation step.

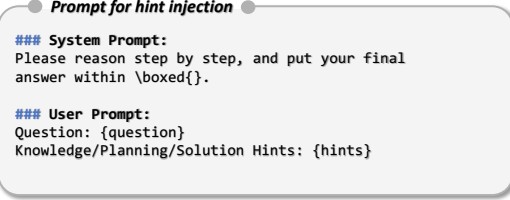

Figure 4: Prompt for hint injection.

To provide the minimal necessary guidance, the framework executes a deterministic search through this hierarchy, proceeding from the most abstract to the most concrete hint ($H_{\text{knowledge}} \rightarrow H_{\text{planning}} \rightarrow H_{\text{solution}}$). Within each tier, guidance is offered incrementally. The search terminates as soon as the model generates a correct solution, thereby identifying the minimal effective guidance required. A detailed description of this progressive exploration algorithm is provided in Appendix D.1.

**On-policy batch augmentation and unified loss.** The core of Scaf-GRPO is its on-policy intervention, reactivating the learning signal during a learning cliff. When all initial trajectories $\mathcal{G} = \{o_1, \ldots, o_N\}$ from $\pi_\theta(\cdot|q)$ yield zero reward, the advantage $\hat{A}_i$ collapses, halting the gradient update. Scaf-GRPO intervenes by finding a minimal hint $h^*$ that enables policy $\pi_\theta$ to generate a successful trajectory $o_h^* \sim \pi_\theta(\cdot|q \oplus h^*)$. This successful trajectory replaces a random failed trajectory $o_j \in \mathcal{G}$ to form an augmented group, $\mathcal{G}_{\text{final}} = (\mathcal{G} \setminus \{o_j\}) \cup \{o_h^*\}$.

The key insight is that Scaf-GRPO does not alter the mathematical form of the GRPO loss function. Instead, it modifies the data used for the loss computation. The advantage calculation is performed on this conditionally augmented batch:

$$\hat{A}'_i = \frac{R(o'_i) - \mu_{\mathcal{G}_{\text{final}}}}{\sigma_{\mathcal{G}_{\text{final}}} + \epsilon_{\text{std}}} \quad \text{for } o'_i \in \mathcal{G}_{\text{final}}. \tag{2}$$

The learning objective remains the clipped surrogate objective, but it is now applied to the trajectories in $\mathcal{G}_{\text{final}}$. The probability ratio for a given trajectory $o'_i \in \mathcal{G}_{\text{final}}$ at timestep $t$ is denoted as $r'_{i,t}(\theta)$. The overall objective is:

$$J_{\text{Scaf-GRPO}}(\theta) = \hat{\mathbb{E}}_{i,t} \left[ \min \left( r'_{i,t}(\theta)\hat{A}'_i, \text{clip}(r'_{i,t}(\theta), 1 - \epsilon, 1 + \epsilon)\hat{A}'_i \right) \right], \tag{3}$$

where the probability ratio $r'_{i,t}(\theta)$ is critically computed with respect to the trajectory's specific originating prompt:

$$r'_{i,t}(\theta) = \begin{cases} \frac{\pi_\theta(o'_{i,t}|o'_{i,<t},q)}{\pi_{\theta_{\text{old}}}(o'_{i,t}|o'_{i,<t},q)} & \text{if } o'_i \in \mathcal{G}_{\text{final}} \text{ and } o'_i \neq o_h^* \\ \frac{\pi_\theta(o'_{i,t}|o'_{i,<t},q \oplus h^*)}{\pi_{\theta_{\text{old}}}(o'_{i,t}|o'_{i,<t},q \oplus h^*)} & \text{if } o'_i = o_h^*. \end{cases} \tag{4}$$

This on-policy augmentation ensures the batch contains non-zero reward variance, restoring a meaningful advantage signal and allowing learning to resume on previously intractable problems.

**Conservative nature and on-policy integrity.** A crucial property of Scaf-GRPO is its conservative nature; the framework does not alter the fundamental GRPO optimization objective but rather operates as a targeted data augmentation strategy. Its impact on the policy gradient can be formalized by analyzing two distinct cases based on the initial sampling results for a given prompt $q$.

In the first case, where at least one successful trajectory is generated initially ($\exists o_i \in \mathcal{G}$ such that $R(o_i) > 0$), the batch already contains a valid learning signal. The condition for intervention is not met, so the batch remains unchanged ($\mathcal{G}_{\text{final}} = \mathcal{G}$). Consequently, the objective function is mathematically identical to standard GRPO, ensuring our framework does not interfere when the model can learn on its own:

$$J_{\text{Scaf-GRPO}}(\theta) \equiv J_{\text{GRPO}}(\theta). \tag{5}$$

In the second case, the learning cliff scenario ($\forall o_i \in \mathcal{G}, R(o_i) = 0$), standard GRPO fails. The uniform zero rewards cause the advantage calculation to collapse ($\mu_{\mathcal{G}} = 0, \sigma_{\mathcal{G}} = 0$), leading to a null advantage $\hat{A}_i = 0$ and a vanishing policy gradient. Here, Scaf-GRPO intervenes by constructing the augmented batch $\mathcal{G}_{\text{final}}$. This restores the gradient by ensuring $\mu_{\mathcal{G}_{\text{final}}} > 0$, which in turn guarantees a non-zero advantage signal $\hat{A}'_i$. Critically, this intervention preserves the on-policy principle. Unlike off-policy methods that import trajectories from a different policy $\pi_\phi$ and require high-variance importance sampling corrections (e.g., using a ratio $\frac{\pi_\theta}{\pi_\phi}$), the guided trajectory $o_h^*$ is sampled directly from the current policy $\pi_\theta$. The probability ratio is therefore a standard on-policy ratio computed on a modified input, which is inherently more stable. We explicitly avoid the distributional mismatch of off-policy alternatives defined by ratios like $\frac{\pi_\theta(\cdot|q)}{\pi_{\theta_{\text{old}}}(\cdot|q \oplus h^*)}$, which are shown to destabilize training in Appendix F.5. Instead, by conditioning both the current and old policies on the identical hint-augmented prompt, Scaf-GRPO ensures a stable learning signal. This targeted, on-policy intervention transforms an unproductive, zero-gradient sample into a valuable learning opportunity without compromising the integrity of the optimization process.

Table 1: Overall performance on seven benchmarks. We compare our method, SCAF-GRPO, against vanilla GRPO baselines across diverse architectures, including the Qwen2.5 series, a non-Qwen model (Llama-3.2-8B-Instruct), and a specialized long-CoT model (DeepSeek-R1-Distill-Qwen-1.5B). Scores: pass@1 (%). Best results are in **bold**. The background color of Scaf-GRPO cells indicates performance change vs. Vanilla GRPO (**green** for improvement, **red** for decline).

| Model | AIME 24 | AIME 25 | AMC | Minerva | MATH-500 | Olympiad | Gaokao2023en | Avg. |
|---|---|---|---|---|---|---|---|---|
| *Qwen2.5-Math-1.5B* | | | | | | | | |
| Qwen2.5-Math-1.5B | 7.2 | 3.3 | 32.5 | 14.7 | 32.8 | 20.6 | 20.0 | 18.7 |
| Vanilla GRPO | 13.3 | 10.0 | 47.5 | 28.3 | 72.2 | 34.8 | 57.4 | 37.6 |
| Scaf-GRPO | **20.0** | **13.3** | **60.0** | **29.1** | **73.4** | **36.6** | **57.9** | **41.5** |
| *Qwen2.5-Math-7B* | | | | | | | | |
| Qwen2.5-Math-7B | 13.3 | 13.3 | 42.5 | 16.5 | 53.6 | 18.2 | 35.1 | 27.5 |
| Vanilla GRPO | 30.0 | 13.3 | 60.0 | 33.4 | 75.8 | 41.3 | 62.6 | 45.2 |
| SimpleRL-Zero Zeng et al. (2025) | 23.3 | 13.3 | 55.0 | 31.6 | 76.8 | 37.2 | 60.8 | 42.6 |
| Oat-Zero Liu et al. (2025a) | 30.0 | 16.7 | 62.5 | 34.6 | 78.0 | 41.0 | 62.9 | 46.5 |
| LUFFY Yan et al. (2025) | 33.3 | 16.7 | 62.5 | 33.8 | 75.2 | 41.7 | 62.7 | 46.6 |
| Scaf-GRPO | **43.3** | **20.0** | **70.0** | **36.4** | **80.0** | **43.3** | **63.4** | **50.9** |
| *Qwen2.5-7B* | | | | | | | | |
| Qwen2.5-7B | 10.0 | 6.7 | 37.5 | 26.4 | 61.8 | 34.4 | 42.6 | 31.3 |
| Vanilla GRPO | 10.0 | 10.0 | 50.0 | 38.5 | 77.6 | 40.4 | **64.2** | 41.5 |
| Scaf-GRPO | **13.3** | **20.0** | **60.0** | **38.6** | **77.8** | **40.8** | 63.8 | **44.9** |
| *Llama-3.2-3B-Instruct* | | | | | | | | |
| Llama-3.2-3B-Instruct | 6.7 | 0.0 | 20.0 | 11.8 | 38.3 | 12.6 | 33.5 | 17.6 |
| Vanilla GRPO | 13.3 | 0.0 | 35.0 | 18.7 | 51.8 | 18.3 | 45.7 | 26.1 |
| Scaf-GRPO | **16.7** | **3.3** | **40.0** | **19.1** | **56.2** | **20.3** | **46.0** | **28.8** |
| *DeepSeek-R1-Distill-Qwen-1.5B* | | | | | | | | |
| DeepSeek-R1-Distill-Qwen-1.5B | 28.9 | 20.0 | 67.5 | 26.1 | 83.9 | 45.8 | 62.1 | 47.7 |
| Vanilla GRPO | 30.0 | 21.1 | 67.5 | 30.1 | 83.9 | 50.2 | 71.4 | 50.6 |
| Scaf-GRPO | **33.3** | **23.3** | **77.5** | **32.4** | **85.8** | **50.7** | **72.3** | **53.6** |

# 4 EXPERIMENTS

## 4.1 EXPERIMENTAL SETUPS

**Training dataset.** Our training data is derived from the DeepScaleR-Preview-Dataset (Luo et al., 2025). We employ a dynamic filtering strategy that aligns the dataset with each model's initial capabilities. Based on preliminary evaluation, we classify problems as "Too Easy" (discarded), "Too Hard" (retained), or "Potentially Solvable" (50% subsampled). This curates a challenging yet tractable training set focused on the frontier of the model's abilities. For this dataset, we generate our three-tiered hints by prompting the DeepSeek-R1 model (Guo et al., 2025) with ground-truth solution steps. Further details on our data filtering strategy and the hint generation process are provided in Appendix B.1 and Appendix B.2, respectively.

**Models.** To demonstrate the general applicability of Scaf-GRPO, we conduct experiments across a diverse set of models, including: math-specialized models (Qwen2.5-Math-7B & 1.5B) to test in-domain performance; a general-purpose base model (Qwen2.5-7B) to assess skill acquisition; a different architecture (Llama-3.2-3B-Instruct) to verify model-agnosticism; and a Long-Chain-of-Thought model (DeepSeek-R1-Distill-Qwen-1.5B) to evaluate applicability to extended reasoning.

**Baseline methods.** We benchmark Scaf-GRPO against three distinct classes of baselines: (1) Vanilla GRPO (Shao et al., 2024), the standard on-policy algorithm without guidance. This serves as our baseline to quantify the gains from our scaffolding mechanism. (2) Leading GRPO implementations, including Simple-RL (Zeng et al., 2025) and Oat-Zero (Liu et al., 2025a), to contextualize our performance against highly-optimized public benchmarks. (3) LUFFY (Yan et al., 2025), a representative of RLVR with off-policy guidance. This provides a direct comparison between the dominant prefix-continuation strategy and our in-prompt scaffolding approach.

**Evaluation details.** We evaluate on diverse mathematics benchmarks, including GaoKao2023en (Chinese GaoKao Community, 2024), AIME24 (AIME, 2024), AIME25 (AIME, 2025), AMC (AMC, 2023), MATH-500 (Hendrycks et al., 2021), and OlympiadBench (He et al.,

2024). To assess out-of-distribution (OOD) generalization, we evaluate on the scientific reasoning benchmark, GPQA-Diamond (Rein et al., 2024). For all benchmarks, we report pass@1 accuracy via greedy decoding. Vanilla GRPO is trained with our data and hyperparameters, and LUFFY on our data with its original parameters. For Simple-RL and Oat-Zero, we evaluate their publicly available weights.

**Implementation details.** We train all models for 10 epochs using the verl framework (Sheng et al., 2025), reporting results from the best-performing checkpoint. The maximum response length is 2048 tokens (8192 for the LongCoT model). Consistent with recent studies (Liu et al., 2025b; Yu et al., 2025; Yan et al., 2025), we set the KL divergence penalty to zero to maximize policy exploration. A comprehensive list of hyperparameters is detailed in Appendix C.

Table 2: Ablation study on Scaf-GRPO's key components using Qwen2.5-Math-7B model. The best performance is highlighted in bold. The "No Guidance" row serves as the vanilla GRPO baseline.

| Hint Strategy | AIME24 | AIME25 | AMC23 | Minerva | MATH-500 | Olympiad | Gaokao2023en | Avg. |
|---|---|---|---|---|---|---|---|---|
| **Scaf-GRPO (Full K →P →S)** | **43.3** | **20.0** | **70.0** | **36.4** | **80.0** | 43.3 | 63.4 | **50.9** |
| w/o Progressive (Solution-Only) | 40.0 | 13.3 | 65.0 | 36.2 | 78.6 | **43.7** | 62.3 | 48.4 |
| w/o Knowledge Hint (P →S) | 43.3 | 13.3 | 70.0 | 34.2 | 77.8 | 42.4 | 63.1 | 49.2 |
| w/o Planning Hint (K →S) | 43.3 | 16.7 | 62.5 | 35.0 | 79.4 | 40.0 | 63.6 | 48.6 |
| w/o Solution Hint (K →P) | 40.0 | 10.0 | 67.5 | 34.2 | 78.6 | 42.2 | 63.4 | 48.0 |
| w/o Incremental Chunking | 43.3 | 10.0 | 62.5 | 36.0 | 76.0 | 41.6 | **64.2** | 47.7 |
| No Guidance (Vanilla GRPO) | 30.0 | 13.3 | 60.0 | 33.4 | 75.8 | 41.3 | 62.6 | 45.2 |

## 4.2 MAIN RESULTS

In this section, we present the primary evaluation of Scaf-GRPO, focusing on pass@1 performance across diverse model architectures. The results highlight the method's performance advantage over Vanilla GRPO and competing prefix-based baselines. Comprehensive supplementary analyses are detailed in Appendix I, covering robustness checks using the avg@16 metric, comparisons against an expanded suite of methods such as DAPO Yu et al. (2025) and DeepScaleR Luo et al. (2025).

**Comparison with GRPO.** As shown in Table 1, compared to the vanilla GRPO baseline, Scaf-GRPO achieves comprehensive and significant performance gains across all tested models. On the Qwen2.5-Math-7B model, Scaf-GRPO boosts the pass@1 score from 0.300 to 0.433 on AIME24, a relative improvement of 44.3%. These results provide strong evidence that our scaffolding mechanism effectively helps the model overcome the "learning cliff," enabling it to tackle problems that were previously beyond its independent capabilities.

**Comparison with other methods.** To contextualize Scaf-GRPO within the broader research landscape, we compare it against other leading methods in Table 1. Scaf-GRPO on Qwen2.5-Math-7B demonstrates a marked superiority, achieving an average score of 0.509. This performance represents a substantial improvement of 19.5% over Simple-RL and 9.5% over Oat-Zero. More importantly, Scaf-GRPO establishes a clear advantage over the prefix-continuation paradigm, outperforming LUFFY by 9.2%. This significant outperformance suggests that our in-prompt scaffolding strategy offers a more effective training alternative to prefix-continuation methods.

**Generalization to non-Qwen architectures.** To verify that the benefits of Scaf-GRPO are not confined to a single model family, we extend our evaluation to the Llama-3.2-3B-Instruct model (Dubey et al., 2024). As detailed in Table 1, our framework demonstrates strong generalization. While vanilla GRPO provides a significant uplift over the base model, Scaf-GRPO achieves a further relative improvement of 10.3% in average performance. This confirms Scaf-GRPO is a model-agnostic method, capable of enhancing reasoning abilities beyond the Qwen series.

**Applicability to LongCoT models.** We further investigate the efficacy of Scaf-GRPO on models optimized for Long Chain-of-Thought (LongCoT) reasoning, using the specialized DeepSeek-R1-Distill-Qwen-1.5B model. The results in Table 1 show that Scaf-GRPO effectively enhances this already capable baseline, delivering a 5.9% relative performance gain over vanilla GRPO. This demonstrates our framework's versatility in scaffolding not only standard-length solutions but also the extensive derivations characteristic of LongCoT models.

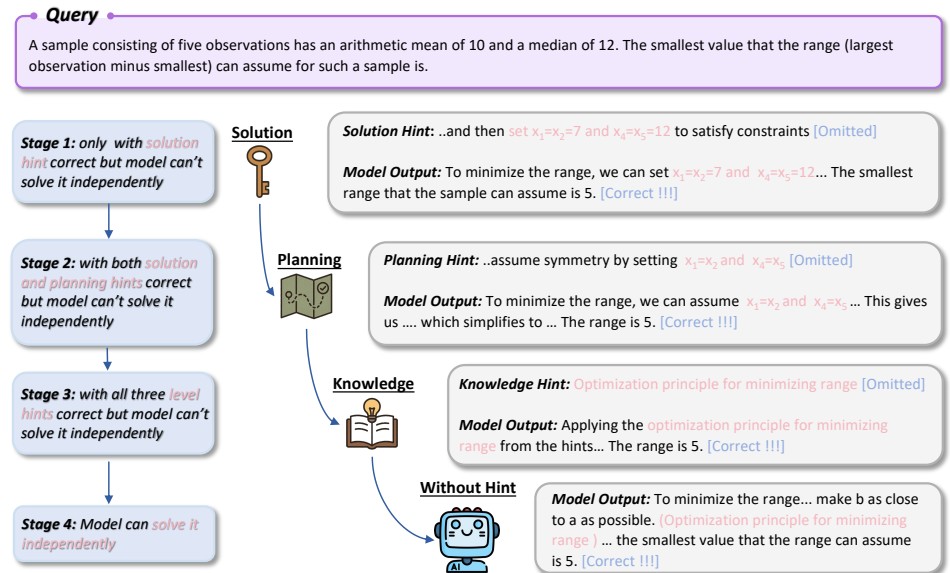

Figure 5: Evolution of reasoning from guidance to autonomy. The model progresses from imitating a concrete hint (a) to applying abstract knowledge (b), ultimately achieving (c) autonomous problem-solving by internalizing the guided skills.

## 4.3 ABLATION STUDY

We conduct a series of ablation studies on the Qwen2.5-Math-7B model (see Table 2). Detailed investigations into the guidance exemption period and data filtering strategies are presented in Appendix G.1 and Appendix G.2, respectively.

**Necessity and robustness of the guidance exemption period.** To validate the necessity and robustness of the guidance exemption period (Phase 1), we conduct detailed ablation studies in Appendix G.1. Our experiments show that applying scaffolding from the very beginning leads to a 9.2% relative performance drop on Qwen2.5-Math-7B compared to the full framework, confirming that an initial phase of autonomous exploration is crucial to prevent hint dependency. Furthermore, sensitivity analysis reveals that the method is highly stable across exemption durations ranging from 10% to 40%, where the model maintains a high-performance plateau between 49.5% and 50.9%. This validates our selection of 15% as an optimal and robust configuration.

**Efficacy of progressive & hierarchical guidance.** Our methodology is founded on the hypothesis that progressive guidance, from abstract concepts to concrete steps, is superior to simply providing a direct solution. To test this, we evaluate a "Solution-Only" variant that bypasses the hierarchy and immediately provides the most concrete hint, which is $H_{\text{solution}}$. This results in a significant performance degradation of 4.9% compared to the full model. This confirms our hypothesis: compelling the model to first engage with higher-level reasoning fosters more generalizable skills.

**Justifying the completeness of the hint hierarchy.** We design a three-tiered hint structure (K→P→S) assuming each layer serves a unique function. To verify this, we systematically removed one layer at a time. As shown in Table 2, every removal degrades performance. The most severe degradation, a 5.7% drop, occurs when the final "Solution" hint is removed (the K→P variant). This highlights the dual role of the hierarchy: abstract hints encourage high-level reasoning, while concrete hints serve as an essential fallback. The superior performance of the full K→P→S model demonstrates that the layers are complementary, not redundant.

**Efficacy of incremental guidance.** A core principle of Scaf-GRPO is to provide the minimal necessary support by delivering hints incrementally. We test this against a "Full Hint" variant, which provides the entire content of a hint level at once. This non-incremental approach collapses performance by 6.3% compared to the incremental one. This decline validates our strategy: minimal, incremental intervention is critical for preserving model autonomy and preventing over-reliance.

**Impact of hint quality.** We investigate the correlation between hint quality and student performance using a multi-faceted rubric (accuracy, minimality, clarity, and structural coherence). Em-

Table 3: Impact of data filtering on Scaf-GRPO vs. Vanilla GRPO. Both methods were trained on the full dataset (Original) and a harder subset (Filtered). The best performance is highlighted in bold. Scores are pass@1 (%).

| Data | Method | AIME24 | AIME25 | AMC23 | MATH-500 | Olympiad | Avg. |
|------|--------|--------|--------|-------|----------|----------|------|
| | | | *Qwen2.5-Math-1.5B* | | | | |
| Original | Vanilla GRPO | 13.3 | 6.7 | 52.5 | 68.6 | 31.4 | 34.5 |
| Original | Scaf-GRPO | 20.0 | 10.0 | 55.0 | 73.2 | 36.4 | 38.9 |
| Filtered | Vanilla GRPO | 13.3 | 10.0 | 47.5 | 72.2 | 34.8 | 35.6 |
| Filtered | Scaf-GRPO | **20.0** | **13.3** | **60.0** | **73.4** | **36.6** | **40.7** |
| | | | *Qwen2.5-Math-7B* | | | | |
| Original | Vanilla GRPO | 30.0 | 16.7 | 60.0 | 74.4 | 38.5 | 43.9 |
| Original | Scaf-GRPO | 33.3 | 16.7 | 70.0 | 79.0 | 43.0 | 48.4 |
| Filtered | Vanilla GRPO | 30.0 | 13.3 | 60.0 | 75.8 | 41.3 | 44.1 |
| Filtered | Scaf-GRPO | **43.3** | **20.0** | **70.0** | **80.0** | **43.3** | **51.3** |

ploying an LLM-as-a-Judge, we observe that higher-quality hints lead to superior downstream outcomes. Notably, DeepSeek-R1 achieved a higher hint quality score, outperforming Qwen2.5-72B-Instruct. This translated to significant performance gains for the student model (Qwen2.5-Math-7B), yielding a relative accuracy improvement of 4%. We defer the complete evaluation rubric, the specific prompt used for the LLM judge, and detailed experimental results to Appendix H.

## 4.4 FURTHER ANALYSIS

**Confronting the learning cliff.** Figure 2 visualizes Scaf-GRPO's advantage. In Figure 2(a), we plot the number of "zero-reward" problems per batch. The count for both methods drops sharply at the start of training. However, the vanilla GRPO curve quickly flattens, defining the learning cliff: a point where the baseline can no longer extract a learning signal from a persistent set of "true-hard" problems. In contrast, Scaf-GRPO's scaffolding activates, enabling the model to consistently learn from these problems and continue reducing the zero-reward count. This directly impacts validation performance (Figure 2(b)). By turning intractable problems into learning opportunities, Scaf-GRPO achieves a higher, steadily improving validation score while the baseline stagnates.

**Internalizing skills beyond imitation.** Scaf-GRPO succeeds by fostering skill acquisition rather than simple imitation. Figure 5 illustrates this trajectory on a challenging problem: the model evolves from utilizing concrete "Solution Hints" to abstract "Knowledge Hints," and finally to solving the problem without any support. To quantify this transition from dependency to autonomy, we track "skill graduation" events over the first 300 training steps. We define this metric to capture the learning breakthrough specific to each method: for Scaf-GRPO, we count problems transitioning from hint-dependent to autonomous success, while for the Vanilla baseline, we track the shift from total failure to success. As shown in Table 4, Scaf-GRPO consistently yields a significantly higher volume of graduations (e.g., +137.8% on Qwen2.5-Math-1.5B). This confirms that our method effectively converts temporary guidance into lasting, independent reasoning capabilities. By enabling such robust skill-building on hard problems, Scaf-GRPO effectively overcomes the learning cliff.

Table 4: Comparison of total "graduations" across different model backbones.

| Model | Vanilla | Scaf-GRPO (Ours) | Relative Increase |
|-------|---------|------------------|-------------------|
| Qwen2.5-Math-1.5B | 1,123 | 2,670 | +137.8% |
| Qwen2.5-Math-7B | 434 | 483 | +11.3% |
| Qwen2.5-7B | 464 | 723 | +55.8% |
| DeepSeek-R1-Distill-1.5B | 453 | 694 | +53.2% |
| Llama-3.2-3B-Instruct | 577 | 986 | +70.9% |

**Aligning data difficulty with model capacity.** To validate our data filtering strategy, we train models on both the complete dataset and our filtered subset. As detailed in Table 3, the harder, filtered data yields a marginal 0.5% relative gain for vanilla GRPO but a substantial 6.0% boost for Scaf-GRPO on Qwen2.5-Math-7B. This disparity underscores that exposing a model to difficult problems is insufficient. A challenging curriculum is most effective when paired with a robust learning framework like Scaf-GRPO, which converts these challenges into learning opportunities. We also observe performance is sensitive to the density of solvable problems. Specifically, discarding "Too Easy" samples and subsampling the "Potentially Solvable" category at 50% achieves

the optimal balance, yielding a 10.2% relative gain over the full dataset (Table 12). This empirical evidence justifies our training data selection. All experimental details are provided in Appendix G.2.

**Generalization to out-of-distribution tasks.** To verify that Scaf-GRPO cultivates robust reasoning skills beyond in-domain pattern matching, we evaluate its generalization on the expert-level OOD benchmark, GPQA-Diamond. As shown in Table 6, Scaf-GRPO significantly enhances the Qwen2.5-Math-7B model, achieving a 15.5% relative improvement over Vanilla GRPO and matching the strong LUFFY baseline. Furthermore, this gain effectively transfers to the general-purpose Qwen2.5-7B-Base, where our method yields a 7.5% relative increase over Vanilla GRPO and outperforms LUFFY. These results demonstrate that the problem-solving abilities fostered by Scaf-GRPO are fundamental and agnostic to the model backbone.

**Computational efficiency.** Scaf-GRPO optimizes training efficiency by applying guidance selectively and accelerating convergence. Empirically, during the training of Qwen2.5-Math-7B, the hint-guided exploration is triggered for only 17.4% of the samples, ensuring that the majority of the computational throughput remains dedicated to standard generation. More importantly, by converting zero-reward signals into high-value learning signals, Scaf-GRPO significantly reduces the total training duration required to reach optimal performance. As shown in Table 5, Scaf-GRPO reaches its best-performing checkpoint (50.9% avg.) in approximately 12 hours. This represents a clear efficiency gain over the Vanilla GRPO baseline, which requires 13 hours to reach a lower peak performance (45.2%), demonstrating that Scaf-GRPO achieves better results with a smaller total time budget.

Table 5: Training efficiency comparison on Qwen2.5-Math-7B. Best results are in **bold**.

| Method | Best Avg. (%) | Time to Best Ckpt. | Trigger Rate | Peak Memory |
|---|---|---|---|---|
| *Qwen2.5-Math-7B* | | | | |
| Vanilla GRPO | 45.2 | ~13 hours | N/A | ~72 GB |
| Scaf-GRPO (Ours) | **50.9** | **~12 hours** | 17.4% | ~73 GB |

## 5 DISCUSSION AND CONCLUSION

**Limitations.** The practical deployment of Scaf-GRPO is subject to two main considerations. First, its efficacy currently relies on the availability of a high-quality, tiered hint hierarchy. Generating these structured hints requires a non-trivial data preparation effort. Second, the framework is principally designed for tasks with verifiable solutions and structured reasoning paths, such as mathematics. Its applicability to more open-ended, subjective domains like creative writing is less direct.

Table 6: OOD performance (pass@1,%) of Qwen2.5-Math-7B and Qwen2.5-7B-Base on the GPQA-Diamond benchmark.

| Model | GPQA-Diamond |
|---|---|
| *Qwen2.5-Math-7B* | |
| Base Model | 24.7 |
| Vanilla GRPO | 32.3 |
| SimpleRL-Zero Zeng et al. (2025) | 33.3 |
| Oat-Zero Liu et al. (2025a) | 33.3 |
| LUFFY Yan et al. (2025) | 37.3 |
| **Scaf-GRPO (Ours)** | **37.3** |
| *Qwen2.5-7B-Base* | |
| Vanilla GRPO | 33.3 |
| LUFFY Yan et al. (2025) | 34.4 |
| **Scaf-GRPO (Ours)** | **35.8** |

**Future work.** Future research could focus on automating hint generation to enhance the framework's scalability. We also plan to explore adaptive scaffolding mechanisms where guidance dynamically adjusts to the model's improving proficiency, thereby personalizing the learning process.

**Conclusion.** In this work, we introduce Scaf-GRPO, a training framework that overcomes the "learning cliff" in reinforcement learning for large language models. By providing hierarchical hints in the prompt, Scaf-GRPO offers scaffolding for models to solve problems beyond their reach. This on-policy guidance preserves exploratory autonomy and mitigates the distributional consistency issues inherent in prefix-continuation methods. Our experiments show Scaf-GRPO significantly outperforms vanilla GRPO and strong prefix-based baselines across challenging mathematics benchmarks. This framework enables models to learn from previously intractable problems, establishing a more effective path toward autonomous reasoning.

ETHICS STATEMENT

This research adheres to the principles outlined in the ICLR Code of Ethics. Our primary objective is to contribute to society by advancing the frontiers of machine reasoning, upholding the highest standards of scientific excellence through transparent and reproducible methods. We acknowledge the potential for dual-use applications and the risks associated with the misuse of advanced AI reasoning systems. We also recognize the environmental impact of training large models and have striven for computational efficiency. Our work is based on publicly available datasets devoid of personal identifiable information. We are committed to fostering responsible innovation and encourage continued investigation into the societal impacts of increasingly capable models.

REPRODUCIBILITY STATEMENT

The supplementary material contains the complete source code to ensure full reproducibility. This includes the implementation of Scaf-GRPO and all scripts for data filtering and the training pipeline.

ACKNOWLEDGEMENTS

This work was supported in part by the Research Grants Council under the Areas of Excellence scheme grant AoE/E-601/22-R.

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

## A  THE USE OF LARGE LANGUAGE MODELS (LLMS)

During the preparation of this manuscript, the authors employed a large language model (LLM) to aid in refining the language and correcting grammatical errors. The role of the LLM was strictly limited to that of a writing-enhancement tool. The authors take full responsibility for the substantive content, arguments, and final phrasing presented in this paper.

## B  DATASET AND BENCHMARK DETAILS

### B.1  TRAINING DATA SOURCE AND FILTERING

Our training data is derived from the DeepScaleR-Preview-Dataset (Luo et al., 2025), a comprehensive collection of 40k mathematical problems. Its contents are sourced from AIME, AMC, MATH (Hendrycks et al., 2021), Still (Min et al., 2024), and Omni-MATH (Gao et al., 2024a). To maximize training efficiency and target problems most conducive to learning, we implement a dynamic data filtering strategy tailored to each model's capabilities. This strategy categorizes problems based on the model's initial performance, assessed through 8 independent sampling attempts for each problem. These samples are generated using nucleus sampling with a temperature of 1.0 (top-p=1.0, top-k=-1) and a maximum length of 2048 tokens for both the prompt and the response. For our LongCoT model, this response token was increased to 8192 tokens to accommodate its generation style. Based on the outcomes, problems are categorized as follows:

- **Too Easy:** Problems solved correctly in all 8 attempts are excluded, as they offer minimal learning value.
- **Potentially Solvable:** Problems solved in 1 to 7 of the attempts are considered to be within the model's learning-rich sweet spot. We randomly sample 50% of these for inclusion.
- **Too Hard:** Problems that fail in all 8 attempts are retained, as they are the primary candidates for our scaffolding mechanism.

This filtering process results in a final training dataset where approximately 50% of the problems are from the "Potentially Solvable" category and the remaining 50% are "Too Hard". This curated dataset is challenging yet tractable, maximizing the efficiency of the training process for both the baseline GRPO and our Scaf-GRPO framework.

### B.2  HINT GENERATION FOR TRAINING DATA

The hierarchical hints ($H_{knowledge}$, $H_{planning}$, $H_{solution}$) are the cornerstone of Scaf-GRPO's guidance mechanism. To create them, we perform a one-time, offline preprocessing step using the powerful DeepSeek-R1 model (Guo et al., 2025). For each problem in our curated training set, we provide the model with both the problem statement and its ground-truth solution trace.

We then employ a highly structured prompt, detailed in Appendix E, which is engineered not only to decompose the solution into our three-tiered hierarchy but also to enforce a crucial internal structure. Specifically, the prompt compels the model to generate exactly four numbered, progressive items for each category. These items are designed to build upon one another, creating four distinct levels of guidance. For instance, the four items for an $H_{planning}$ hint might represent: (1) the first step of the plan, (2) the second step, (3) the third step, and (4) the fourth step. This structured, multi-level design within each hint category enables the fine-grained, progressive exploration central to our method. The entire process ensures a consistent and high-quality set of structured hints, and the scripts will be included in our code release.

### B.3  EVALUATION BENCHMARKS

We evaluate all models on a diverse suite of seven challenging mathematics benchmarks and the GPQA-Diamond benchmark to ensure a robust and comprehensive assessment of their reasoning abilities. Table 7 provides details for each benchmark used. The mathematics datasets span various difficulty levels and mathematical domains, from high-school competition problems to Olympiad-level challenges, providing a rigorous testbed for advanced reasoning. GPQA-Diamond serves as a crucial Out-of-Distribution (OOD) benchmark, testing generalization to expert-level scientific questions outside the training domain. All benchmarks are publicly available.

### B.4 MOTIVATION FOR DATASET SELECTION

Our choice of the DeepScaleR-Preview-Dataset for training is deliberate. This decision is motivated by two key factors. First, the dataset's broad scope, encompassing problems of varying difficulty, provides the rich and diverse material necessary for our filtering strategy to be effective. Second, its successful application in prior research (Min et al., 2024) establishes it as a robust and relevant foundation for training advanced reasoning models.

For evaluation, our selection of benchmarks is designed for a rigorous and multifaceted assessment. The suite primarily consists of challenging competition-level benchmarks (AIME24/25, AMC, MATH, OlympiadBench, Minerva) and a standardized national exam (GaoKao2023), covering a broad spectrum of mathematical reasoning. Crucially, to measure out-of-distribution (OOD) generalization, we include the GPQA-Diamond benchmark (Rein et al., 2024). As GPQA consists of graduate-level questions whose style and domain are distinct from our training data, strong performance on this benchmark indicates that Scaf-GRPO fosters genuine reasoning skills rather than mere pattern memorization of the training distribution.

| Benchmark | Description | Citation | # Problems |
|---|---|---|---|
| AIME24 | American Invitational Mathematics Examination 2024 | (AIME, 2024) | 30 |
| AIME25 | American Invitational Mathematics Examination 2025 | (AIME, 2025) | 30 |
| AMC | American Mathematics Competitions 2023 | (AMC, 2023) | 25 |
| MATH-500 | A subset from the MATH test set | (Hendrycks et al., 2021) | 500 |
| GaoKao2023en | Chinese National College Entrance Exam 2023 | (Chinese GaoKao Community, 2024) | 385 |
| OlympiadBench | Math Olympiad-level problems | (He et al., 2024) | 675 |
| Minerva | A specialized dataset to evaluate quantitative and scientific reasoning abilities | (Lewkowycz et al., 2022) | 272 |
| GPQA-Diamond | Expert-level questions across biology, physics, and chemistry | (Rein et al., 2024) | 198 |

Table 7: Details of evaluation benchmarks used in our experiments.

## C EXPERIMENTAL SETUP DETAILS

### C.1 COMPUTING INFRASTRUCTURE

All experiments are conducted on a high-performance computing cluster. The specific hardware and software configurations are as follows:

- **Hardware:** All models are trained and evaluated on servers equipped with 8 NVIDIA A100 (80GB) GPUs.

- **Software:** The operating system is Ubuntu 22.04. Key software libraries and their versions include PyTorch 2.6.0, Transformers 4.51.1, and CUDA 12.4.

- **Framework:** Our implementation is built upon the verl (0.4.1) framework (Sheng et al., 2025), a robust and efficient library designed for large-scale reinforcement learning with LLMs.

### C.2 HYPERPARAMETER DETAILS

Our experimental setup is carefully configured for performance and reproducibility. The final hyperparameter configuration is detailed comprehensively in Table 8. These settings were applied consistently across all experiments to ensure a fair comparison. However, to accommodate models specialized in Long Chain-of-Thought (LongCoT) reasoning, we increased the maximum response length to 8192 tokens for those specific tasks.

| Hyperparameter | Value |
|---|---|
| *Optimization & Training* | |
| Learning Rate (LR) | $1 \times 10^{-6}$ |
| Optimizer | AdamW |
| Weight Decay | 0.0 |
| *Batching Strategy* | |
| Global Batch Size | 256 |
| PPO Mini-batch Size | 64 |
| Micro-batch Size (per GPU) | 16 |
| Validation Batch Size | 512 |
| *RL Algorithm (GRPO)* | |
| Rollouts per Query ($N$) | 8 |
| GRPO Clip Epsilon ($\epsilon$) | 0.2 |
| KL Divergence Penalty ($\beta$) | 0.0 |
| Entropy Coefficient | 0.0 |
| *Generation & Tokenization* | |
| Rollout Temperature | 1.0 |
| Max Response Tokens | 2048 |
| *Infrastructure & Scheduling* | |
| Nodes | 1 |
| GPUs per Node | 8 |
| VLLM GPU Memory Utilization | 0.8 |

Table 8: Comprehensive list of key hyperparameters for training and generation.

## C.3 Implementation of Baseline Methods

To ensure a rigorous and fair comparison, we carefully implement or utilize baselines as follows:

**Vanilla GRPO.** This is our primary control. We train a vanilla GRPO model using our verl framework. It is configured with the exact same filtered dataset and hyperparameters as Scaf-GRPO, allowing us to cleanly isolate the performance contribution of our scaffolding mechanism.

**LUFFY.** To compare against the dominant prefix-continuation paradigm, we train LUFFY (Yan et al., 2025) using its official public implementation and its original, recommended hyperparameter settings. To ensure a fair comparison of methodological effectiveness, we train it on our high-quality filtered dataset. This directly contrasts their off-policy guidance with our in-prompt scaffolding on identical data.

**Simple-RL and Oat-Zero.** For other leading methods like Simple-RL (Zeng et al., 2025) and Oat-Zero (Liu et al., 2025a), we do not perform any implementation or retraining. Instead, we evaluate their officially released, publicly available model checkpoints directly. All reported results for these models are obtained by running them through our unified evaluation pipeline, ensuring a consistent and fair comparison against established state-of-the-art work.

## C.4 Evaluation Metrics Details

Our primary evaluation metric is pass@1, which measures the percentage of problems for which a model generates a correct solution in a single attempt. This metric is chosen for its straightforwardness and its status as a standard for evaluating definitive problem-solving capabilities. For all evaluations, we use greedy decoding to generate one complete solution trace for each problem.

The verification process is tailored to the benchmark type to ensure maximum rigor and fairness.

- **For all mathematical reasoning benchmarks,** we employ the "symeval" library (Tong, 2024), specifically its EvaluatorMathBatch module, to determine correctness. This approach moves beyond simple string comparison by using a sophisticated pipeline that combines regular expressions for robust answer extraction with SymPy for symbolic mathematical evaluation. This allows for the accurate verification of complex answers, including matrices, intervals, and symbolic expressions.

- **For the out-of-distribution GPQA-Diamond benchmark,** we utilize the EleutherAI's lm-evaluation-harness (Gao et al., 2024b) to ensure a fair and standardized assessment. This widely adopted framework provides a consistent testing environment for generative models. We use its implementation of the gpqa-diamond task to compute the pass@1 score, thereby maintaining metric consistency while leveraging a community-standardized evaluation harness for OOD generalization.

# D Methodology Details

This section provides a granular description of the Progressive Exploration and Replacement Algorithm, which is central to how Scaf-GRPO overcomes the learning cliff by strategically providing minimal guidance during training.

## D.1 The Progressive Exploration and Replacement Algorithm

When a "true-hard" problem triggers the hierarchical hint-guided exploration phase, Scaf-GRPO executes a deterministic, multi-level search algorithm to find the minimal effective hint. The algorithm's goal is to provide just enough information for the model to succeed, thereby maximizing its independent reasoning.

The algorithm leverages the pre-generated, four-level progressive hint structure, detailed in Appendix B.2. It systematically searches through the hint categories in order of decreasing abstraction ($H_{\text{knowledge}} \rightarrow H_{\text{planning}} \rightarrow H_{\text{solution}}$) and, within each category, through the four levels of increasing detail.

Let $h_c^i$ denote the $i$-th hint item for a category $c \in \{$knowledge, planning, solution$\}$. The cumulative hint provided to the model at level $l \in \{1, 2, 3, 4\}$ is the union of the first $l$ items, denoted as $C_c^l = \bigcup_{i=1}^{l} \{h_c^i\}$. The search process for a single problem is as follows:

(1) Iterate through Hint Categories: For each category $c$ in the sequence ("knowledge", "planning", "solution"):
   (1) Iterate through Hint Levels: For each level $l$ from 1 to 4:
      (1) Construct an augmented prompt by injecting the cumulative hint $C_c^l$.
      (2) Generate a new solution on-policy using this augmented prompt.
      (3) If the generated solution is correct, the search successfully terminates. The trajectory produced with hint $C_c^l$ replaces one of the failed trajectories in the batch. The algorithm then concludes for this problem.

(2) Handle Intractable Case: If the nested loops complete without finding a correct solution (i.e., even the most detailed hint $C_{\text{solution}}^4$ fails), the problem is deemed intractable for the current training step. No replacement occurs, and the algorithm concludes for this problem, leaving the original all-failure group in the batch.

This structured and exhaustive search ensures that if a solution is reachable with any level of guidance, the framework will find it using the most abstract and minimal hint possible, thereby preserving the on-policy learning signal for "true-hard" problems.

## E  PROMPT DESIGN IN SCAF-GRPO

The effectiveness of Scaf-GRPO relies on two distinct but complementary types of structured prompts: those for generating the hierarchical hints, and those for injecting these hints during training.

### E.1  HINT GENERATION PROMPT

To systematically create our tiered hints, we provide a powerful teacher model (DeepSeek-R1) with a structured prompt. For each problem-solution pair in our dataset, this prompt instructs the teacher model to decompose the solution into our three-tiered hierarchy ($H_{\text{knowledge}}, H_{\text{planning}}, H_{\text{solution}}$). This semi-automated process is a critical preprocessing step that ensures a consistent and high-quality hint dataset. The exact prompt template used is below.

---

**Prompt for hint injection**

```
**[ROLE & GOAL]**
You are an expert AI assistant specializing in problem-solving methodology and
knowledge engineering. Your task is to analyze a given problem and its ground-truth
solution, and then generate a structured breakdown of the reasoning process with a high
degree of granularity.

**[INPUT]**
I will provide you with a "Problem" and its "Ground-Truth Solution".

**[INSTRUCTIONS]**
Based on the provided input, you must generate exactly THREE components. For each
component, you MUST generate **a minimum of 4 numbered items**.

- If a natural breakdown results in fewer than 4 items, you must **subdivide the
existing steps into more detailed, finer-grained sub-steps** to meet the requirement.
For example, a single calculation step can be broken down into 'substituting values',
'performing the operation', and 'stating the result'.

1.  **Planning Skeleton**: Extract a high-level planning skeleton. This should be a
concise, ordered list of the key reasoning steps and the overall strategy used to reach
the solution. Do not include detailed calculations, just the logical flow. Break it
down into at least 4 detailed steps.
2.  **Knowledge Components**: Identify at least 4 essential knowledge components (like
facts, definitions, theorems, lemmas, or formulas) required to solve the problem. List
each component clearly in a numbered list.
3.  **Solution Breakdown**: Divide the original Ground-Truth Solution into a numbered
list of semantically coherent steps or chunks.There should be at least 4 steps or
chunks. Each item in the list should be a direct quote or a faithful summary of a part
of the original solution text.
```

---

```
**[OUTPUT FORMAT]**
You MUST provide your response in the following structured format. Ensure each section
contains at least 4 items.

<PLANNING_SKELETON>
1. [Item 1]
...
4. [Item 4]
... (and more if applicable)
</PLANNING_SKELETON>

<KNOWLEDGE_COMPONENTS>
1. [Item 1]
...
4. [Item 4]
... (and more if applicable)
</KNOWLEDGE_COMPONENTS>

<SOLUTION_BREAKDOWN>
1. [Item 1]
...
4. [Item 4]
... (and more if applicable)
</SOLUTION_BREAKDOWN>

**[EXAMPLE]**

--- BEGIN EXAMPLE ---
# ... (Example Problem and Solution are the same)
--- END EXAMPLE ---

**[EXPECTED OUTPUT FOR THE EXAMPLE]**
(This example now demonstrates the required granularity with >= 4 items per section)

<PLANNING_SKELETON>
1. Identify the geometric shape (right-angled triangle) and the goal (find the
hypotenuse).
2. Recall the relevant mathematical theorem that connects the sides of a right-angled
triangle.
3. Formulate the equation by substituting the given side lengths (base and height) into
the theorem.
4. Execute the arithmetic calculation to find the square of the hypotenuse.
5. Perform the final step of taking the square root to isolate the length of the
hypotenuse.
</PLANNING_SKELETON>

<KNOWLEDGE_COMPONENTS>
1. Theorem: Pythagorean Theorem (a² + b² = c² for a right-angled triangle).
2. Definition: Hypotenuse (The longest side of a right-angled triangle, opposite the
right angle).
3. Concept: Right-angled Triangle (A triangle with one angle measuring 90 degrees).
4. Mathematical Operation: Square Root (The inverse operation of squaring a number).
</KNOWLEDGE_COMPONENTS>

<SOLUTION_BREAKDOWN>
1. To find the hypotenuse of a right-angled triangle, we can use the Pythagorean
theorem, which states that a² + b² = c², where a and b are the lengths of the two
shorter sides (legs) and c is the length of the hypotenuse.
2. The given values are a = 4 cm and b = 3 cm.
3. Substituting these into the formula gives: 4² + 3² = 16 + 9 = 25.
4. This means c² = 25. Taking the square root of both sides results in c = 5 cm.
</SOLUTION_BREAKDOWN>

---

**[TASK]**
Now, please process the following problem and solution, strictly following all
instructions.

**Problem**:
{question}

**Ground-Truth Solution**:
{solution}
```

### E.2 INJECT HINTS PROMPT

During the Hierarchical Hint-Guided Exploration phase, when the model fails to solve a "true-hard" problem, Scaf-GRPO injects a hint directly into the input prompt. This approach is fundamental to our on-policy methodology, as it reframes the problem for the model rather than forcing it to continue a partial, off-policy trajectory. The design of this prompt is crucial: it explicitly informs the model that it is receiving guidance, ensuring that the model processes both the problem and the hint under a single, unified policy, thereby avoiding the distributional shifts common in prefix-continuation methods.

The exact prompt template used for hint injection is shown below.

> **Prompt for hint injection**
>
> ```
> ## System message
> Please reason step by step, and put your final
> answer within \boxed{}.
>
> ## User query
> Qusetion: {{question}}
> Solution/Planning/Knowledge Hints: {{hint}}
> ```

## F  FORMAL ALGORITHMIC AND MATHEMATICAL DESCRIPTION OF SCAF-GRPO

This section provides a formal mathematical and algorithmic description of the Scaffolded Group Relative Policy Optimization (Scaf-GRPO) framework. We formalize the two-phase training process and detail the construction of the loss function, particularly for the hierarchical hint-guided exploration phase.

### F.1 PRELIMINARIES: THE STANDARD GRPO OBJECTIVE

We begin by restating the core Group Relative Policy Optimization (GRPO) objective. For a given prompt $q$, the policy $\pi_\theta$ generates a group of $N$ trajectories, $\mathcal{G} = \{o_1, \ldots, o_N\}$, where each trajectory is sampled from the policy, $o_i \sim \pi_\theta(\cdot|q)$. Each trajectory receives a terminal reward $R(o_i)$ from an external verifier.

The normalized advantage for each trajectory $o_i$ in the group $\mathcal{G}$ is calculated as:

$$\hat{A}_i = \frac{R(o_i) - \mu_\mathcal{G}}{\sigma_\mathcal{G} + \epsilon_{\text{std}}} \tag{6}$$

where $\mu_\mathcal{G}$ and $\sigma_\mathcal{G}$ are the mean and standard deviation of rewards in $\mathcal{G}$, and $\epsilon_{\text{std}}$ is a small constant for numerical stability.

The standard GRPO objective, which is maximized during training, is a clipped surrogate objective defined as the empirical expectation over trajectories and timesteps:

$$J_{\text{GRPO}}(\theta) = \hat{\mathbb{E}}_{i,t} \left[ \min \left( r_{i,t}(\theta)\hat{A}_i, \, \text{clip}(r_{i,t}(\theta), 1 - \epsilon, 1 + \epsilon)\hat{A}_i \right) \right] \tag{7}$$

where $r_{i,t}(\theta) = \frac{\pi_\theta(o_{i,t}|o_{i,<t},q)}{\pi_{\theta_{\text{old}}}(o_{i,t}|o_{i,<t},q)}$ is the probability ratio for the token at timestep $t$ of trajectory $i$ between the current and old policies, and $\epsilon$ is the clipping hyperparameter. The "learning cliff" phenomenon, a key challenge addressed by our work, occurs when $R(o_i) = 0$ for all $i \in \{1, \ldots, N\}$. In this scenario, $\mu_\mathcal{G}$ and $\sigma_\mathcal{G}$ become zero, causing the advantage $\hat{A}_i$ to collapse to zero for the entire group and stall the learning process.

### F.2 THE SCAF-GRPO TRAINING PROCESS: A TWO-PHASE FORMULATION

Scaf-GRPO modifies the training process by strategically augmenting the trajectory group $\mathcal{G}$ when a learning cliff is detected. The process consists of a conditional batch construction procedure followed by the application of the standard GRPO loss.

---

**Algorithm 1** Scaf-GRPO Batch Construction and Update

---

**Require:** Policy $\pi_\theta$; Prompt $q$; Current training step $t$; Guidance exemption end-step $T_{\text{exempt}}$; Verifier $\mathcal{V}$; Number of rollouts per prompt $N$.

**Ensure:** Final trajectory group $\mathcal{G}_{\text{final}}$ for loss computation.

    *Step 1: Standard On-Policy Generation*
1: $\mathcal{G} \leftarrow \emptyset$
2: **for** $i = 1$ to $N$ **do**
3:     $o_i \sim \pi_\theta(\cdot|q)$                                ▷ Generate N trajectories for the same prompt
4:     $\mathcal{G} \leftarrow \mathcal{G} \cup \{o_i\}$
5: **end for**
6: $\{\mathcal{R}(o_1), \ldots, \mathcal{R}(o_N)\} \leftarrow \mathcal{V}(\mathcal{G})$                    ▷ Evaluate rewards for the group
    *Step 2: Learning Cliff Monitor*
7: $\mathcal{C}_{\text{cliff}} \leftarrow (\sum_{i=1}^{N} \mathcal{R}(o_i) = 0)$
8: **if** $t > T_{\text{exempt}}$ **and** $\mathcal{C}_{\text{cliff}}$ **then**                 ▷ Guidance is active and needed
        *Step 3: Hierarchical Hint-Guided Exploration*
9:     $(o_h^*, h^*) \leftarrow \text{SearchHierarchicalHints}(q, \pi_\theta)$       ▷ Search for a minimal effective hint
10:     **if** $h^* \neq \text{null}$ **then**             ▷ A successful guided trajectory was found
        *Step 4: Batch Augmentation*
11:         Randomly select an index $j \in \{1, \ldots, N\}$ of a failed trajectory.
12:         $\mathcal{G}_{\text{final}} \leftarrow (\mathcal{G} \setminus \{o_j\}) \cup \{o_h^*\}$       ▷ Replace one failure with the success
13:         **return** $\mathcal{G}_{\text{final}}$
14:     **end if**
15: **end if**
    *Default case: No intervention*
16: $\mathcal{G}_{\text{final}} \leftarrow \mathcal{G}$             ▷ Use original batch if no cliff or guidance failed
17: **return** $\mathcal{G}_{\text{final}}$

---

Figure 6: Overview of the Scaffolded Group Relative Policy Optimization (Scaf-GRPO) Algorithm.

Let $t$ denote the current training step and $T_{\text{exempt}}$ be the step at which the guidance exemption period ends. The core logic is detailed in Figure 6.

The function $\text{SearchHierarchicalHints}(q, \pi_\theta)$ represents the deterministic, multi-level search described in Section 3.3 and Appendix D. It iterates through the pre-defined hint hierarchy $\mathcal{H} = \{\mathcal{H}_{\text{knowledge}}, \mathcal{H}_{\text{planning}}, \mathcal{H}_{\text{solution}}\}$ to find the first hint $h^*$ that enables the policy $\pi_\theta$ to generate a successful trajectory $o_h^* \sim \pi_\theta(\cdot|q \oplus h^*)$, where $\oplus$ denotes the concatenation of the hint into the prompt. If no hint leads to a solution, it returns $(\text{null}, \text{null})$.

### F.3   The Unified Scaf-GRPO Loss Function

The core insight of Scaf-GRPO is that it does not alter the mathematical form of the GRPO loss function. Instead, it modifies the data distribution used for the loss computation by conditionally augmenting the batch.

Let $\mathcal{G}_{\text{final}}$ denote the group of trajectories returned by Figure 6 for a given prompt $q$ at step $t$. This group is composed of trajectories sampled under one of two conditions:

(1) **Standard Generation:** All $N$ trajectories are from $\pi_\theta(\cdot|q)$. This occurs if the learning cliff is not triggered or if the training is within the exemption period.

(2) **Augmented Generation:** $N - 1$ trajectories are from $\pi_\theta(\cdot|q)$ (with zero reward), and one trajectory, $o_h^*$, is from $\pi_\theta(\cdot|q \oplus h^*)$ (with positive reward). This occurs only when the learning cliff is triggered post-exemption and the hint search is successful.

The Scaf-GRPO loss function is therefore defined by applying the standard GRPO objective to this conditionally constructed batch. First, the advantage is computed on the final group $\mathcal{G}_{\text{final}}$:

$$\hat{A}_i' = \frac{R(o_i') - \mu_{\mathcal{G}_{\text{final}}}}{\sigma_{\mathcal{G}_{\text{final}}} + \epsilon_{\text{std}}} \quad \text{for } o_i' \in \mathcal{G}_{\text{final}} \tag{8}$$

The overall objective is then:

$$J_{\text{Scaf-GRPO}}(\theta) = \hat{\mathbb{E}}_{i,t}\left[\min\left(r_{i,t}'(\theta)\hat{A}_i', \text{clip}(r_{i,t}'(\theta), 1 - \epsilon, 1 + \epsilon)\hat{A}_i'\right)\right] \tag{9}$$

where the probability ratio $r'_{i,t}(\theta)$ for each trajectory $o'_i \in \mathcal{G}_{\text{final}}$ is critically computed with respect to its specific originating prompt:

$$r'_{i,t}(\theta) = \begin{cases} \frac{\pi_\theta(o'_{i,t}|o'_{i,<t},q)}{\pi_{\theta_{\text{old}}}(o'_{i,t}|o'_{i,<t},q)} & \text{if } o'_i \in \mathcal{G}_{\text{final}} \text{ and } o'_i \neq o^*_h \\ \frac{\pi_\theta(o'_{i,t}|o'_{i,<t},q \oplus h^*)}{\pi_{\theta_{\text{old}}}(o'_{i,t}|o'_{i,<t},q \oplus h^*)} & \text{if } o'_i = o^*_h. \end{cases} \tag{10}$$

By reformulating the batch rather than the loss, Scaf-GRPO ensures that when a learning signal is absent ($\hat{A}_i = 0$), a new signal is injected by providing a single successful, minimally-guided trajectory. This intervention re-establishes a non-zero reward variance within the group, reactivates the advantage calculation, and enables learning to resume on previously intractable problems, thereby directly overcoming the learning cliff.

### F.4 CONSERVATIVE NATURE AND PRESERVATION OF THE ON-POLICY OBJECTIVE

A crucial property of Scaf-GRPO is that it does not alter the fundamental GRPO optimization objective. Instead, it operates as a conservative data augmentation strategy that activates only under the specific condition of a learning cliff. We can formalize the framework's impact on the policy gradient by analyzing two distinct cases based on the initial on-policy sampling results for a given prompt $q$.

**Case 1: At least one successful trajectory** ($\exists o_i \in \mathcal{G}$ **such that** $R(o_i) > 0$)**.** In this scenario, the initial group of trajectories $\mathcal{G}$ already contains a non-uniform reward signal, meaning $\mu_\mathcal{G} > 0$ and $\sigma_\mathcal{G} \geq 0$. The condition for triggering the hierarchical hint-guided exploration is not met. Consequently, the final batch used for the update is the original batch, $\mathcal{G}_{\text{final}} = \mathcal{G}$. The Scaf-GRPO objective function is therefore mathematically identical to the standard GRPO objective:

$$J_{\text{Scaf-GRPO}}(\theta) \equiv J_{\text{GRPO}}(\theta) \tag{11}$$

In the most frequent training scenarios where the model has some capacity to solve the problem, our framework makes no modifications and is equivalent to vanilla GRPO.

**Case 2: All trajectories fail** ($\forall o_i \in \mathcal{G}, R(o_i) = 0$)**.** This is the learning cliff scenario. In standard GRPO, the rewards are uniform and zero, causing the advantage calculation to collapse: $\mu_\mathcal{G} = 0$ and $\sigma_\mathcal{G} = 0$, leading to $\hat{A}_i = 0$ for all trajectories. The resulting policy gradient for this prompt is zero, and no learning occurs.

Scaf-GRPO intervenes by constructing an augmented batch $\mathcal{G}_{\text{final}}$. This batch consists of $N - 1$ of the original failed trajectories and one new, successful trajectory $o^*_h \sim \pi_\theta(\cdot|q \oplus h^*)$. The crucial insight is that this new trajectory is generated *on-policy* by the current policy $\pi_\theta$, conditioned on the hint-augmented prompt.

The key benefits of this intervention are:

(1) **Gradient Restoration.** The augmented batch $\mathcal{G}_{\text{final}}$ now contains at least one trajectory with a positive reward. This ensures that $\mu_{\mathcal{G}_{\text{final}}} > 0$ and $\sigma_{\mathcal{G}_{\text{final}}} > 0$, which in turn guarantees a non-zero advantage signal $\hat{A}'_i$ for the trajectories in the group. Learning is effectively restored where it would have stalled.

(2) **Preservation of the On-Policy Principle.** Unlike off-policy methods that mix trajectories from a different policy $\pi_\phi$ and require importance sampling corrections (e.g., $\frac{\pi_\theta}{\pi_\phi}$) to account for the distributional shift, Scaf-GRPO's guided trajectory $o^*_h$ is sampled directly from the current policy $\pi_\theta$. Therefore, the probability ratio $r'_{i,t}(\theta)$ is a standard on-policy ratio computed at each timestep. This avoids the high variance and potential instability associated with off-policy corrections, ensuring that the learning signal remains stable and directly attributable to the current policy's capabilities.

In summary, Scaf-GRPO does not introduce any harmful bias or modification to the GRPO objective. It is a targeted intervention that is inactive when a valid learning signal already exists. When the learning signal vanishes, it provides a constructive, on-policy gradient by minimally augmenting the task, thereby transforming an unproductive training sample into a valuable learning opportunity without compromising the integrity of the on-policy optimization process.

### F.5 COMPARATIVE ANALYSIS OF ON-POLICY STABILITY AND DISTRIBUTIONAL MISMATCH

To elucidate the design rationale behind Scaf-GRPO, we provide a comparative analysis against an alternative off-policy formulation. A natural question arises regarding whether the trajectory generated via the hint-augmented prompt ($q \oplus h^*$) could be used to directly update the policy conditioned on the original prompt ($q$). We analyze this alternative from three perspectives: theoretical stability, distributional mismatch, and empirical performance.

**Theoretical rationale for on-policy formulation.** The core mechanism of GRPO relies on the probability ratio $r_t(\theta)$ to measure the divergence between the current and old policies. This ratio is constrained by a clipping mechanism, $\text{clip}(r_t(\theta), 1 - \epsilon, 1 + \epsilon)$, to prevent destructive updates. Stability requires that the ratio remains close to unity, implying the numerator and denominator distributions must be well-aligned.

In Scaf-GRPO, when a learning cliff necessitates the use of a hint $h^*$, we augment the input context for the current policy. Consequently, the probability ratio for a guided trajectory is computed as:

$$r_t(\theta) = \frac{\pi_\theta(\cdot | q \oplus h^*)}{\pi_{\theta_{\text{old}}}(\cdot | q \oplus h^*)} \tag{12}$$

This formulation maintains a strict on-policy property. The numerator and denominator are conditioned on the identical context ($q \oplus h^*$). As a result, the distributions remain closely aligned, maximizing the likelihood that $r_t(\theta)$ falls within the stable trust region $[1 - \epsilon, 1 + \epsilon]$. This ensures smooth gradient updates even when the model operates in a guided state.

**Instability of off-policy alternatives.** An alternative approach involves using the hint-guided trajectory to optimize the policy for the original prompt directly. This implies an off-policy ratio formulation:

$$r_t^{\text{off}}(\theta) = \frac{\pi_\theta(\cdot | q)}{\pi_{\theta_{\text{old}}}(\cdot | q \oplus h^*)} \tag{13}$$

We argue that this formulation introduces a fundamental distributional mismatch. The numerator is conditioned on $q$, while the denominator is conditioned on $q \oplus h^*$. Since $h^*$ is selected specifically to alter the probability landscape and guide the model toward a correct solution that was previously inaccessible under $q$, the distributions $\pi(\cdot | q)$ and $\pi(\cdot | q \oplus h^*)$ are inherently divergent. This discrepancy causes the ratio $r_t^{\text{off}}(\theta)$ to fluctuate significantly, frequently violating the $[1 - \epsilon, 1 + \epsilon]$ bounds. Such behavior forces the optimization algorithm to rely heavily on clipping, thereby truncating the learning signal and introducing high variance into the training process.

**Empirical validation of training stability.** To validate this theoretical analysis, we conduct a comparative experiment implementing the off-policy formulation described above. We monitor the clip ratio, defined as the fraction of tokens where the probability ratio falls outside the stable interval $[1 - \epsilon, 1 + \epsilon]$. A high clip ratio indicates that the policy update is being constrained due to excessive divergence.

Figure 7 illustrates the training dynamics. The proposed Scaf-GRPO maintains a consistently low clip ratio, confirming that the on-policy formulation preserves distributional alignment. In contrast, the off-policy alternative exhibits a significantly higher and more volatile clip ratio. This empirical evidence supports the hypothesis that the distributional mismatch in the off-policy approach destabilizes the optimization landscape.

**Impact on downstream performance.** The instability observed during training directly correlates with inferior final performance. Table 9 presents a performance comparison across mathematical reasoning benchmarks. The model trained with the off-policy ratio suffers from performance degradation compared to Scaf-GRPO. Specifically, the inability to maintain stable updates limits the effective transfer of knowledge, whereas our on-policy approach effectively leverages the scaffolded trajectories to improve the model's reasoning capabilities.

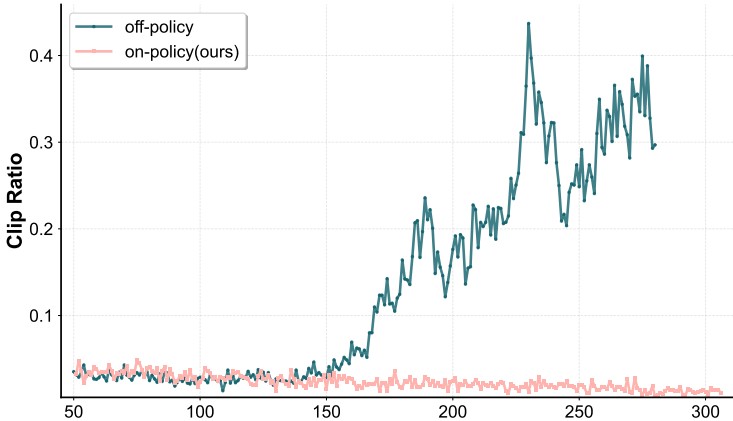

Figure 7: Comparison of the clip ratio during training. The proposed Scaf-GRPO (red) maintains a low and stable clip ratio, indicating effective on-policy learning. The off-policy alternative (blue) exhibits high volatility and frequent clipping, indicative of distributional mismatch and training instability.

Table 9: Performance comparison (Pass@1, %) on Qwen-2.5-Math-7B. We compare Vanilla GRPO, the Off-Policy alternative, and our proposed **Scaf-GRPO**. By maintaining optimization stability, Scaf-GRPO achieves superior performance across all datasets. Best results are highlighted in **bold** with background color.

| Method | AIME 24 | AIME 25 | AMC | Minerva | MATH-500 | Olympiad | Gaokao2023en | Avg. |
|---|---|---|---|---|---|---|---|---|
| *Qwen2.5-Math-7B* | | | | | | | | |
| Vanilla GRPO | 30.0 | 13.3 | 60.0 | 33.4 | 75.8 | 41.3 | 62.6 | 45.2 |
| Off-Policy Alt. | 36.7 | 13.3 | 65.0 | 34.2 | 78.2 | 38.5 | 65.2 | 47.3 |
| **Scaf-GRPO (Ours)** | **43.3** | **20.0** | **70.0** | **36.4** | **80.0** | **43.3** | 63.4 | **50.9** |

## G    ADDITIONAL ABLATION STUDY AND ANALYSIS

### G.1    ABLATION ON THE GUIDANCE EXEMPTION PERIOD

To validate the design of our guidance exemption period (Phase 1), we conduct a two-fold analysis: first establishing the necessity of this phase, and then analyzing the method's sensitivity to its duration.

**Necessity of the Exemption Phase.**    We first investigate whether an initial autonomous learning phase is necessary by comparing our full Scaf-GRPO framework against a variant where scaffolding is activated from the very beginning ("Scaf-GRPO w/o Phase 1"). We posit that Phase 1 is crucial for distinguishing between "true-hard" problems (genuine capability gaps) and "pseudo-hard" problems (superficial errors like formatting). Applying scaffolding prematurely to pseudo-hard cases fosters dependency, preventing the model from developing robust, independent problem-solving habits.

The results in Table 10 confirm this hypothesis. While activating scaffolding from the start yields improvement over the vanilla GRPO baseline, it is notably inferior to the complete Scaf-GRPO framework. This performance gap underscores that allowing an initial phase of unguided learning is critical to prevent over-reliance on hints.

**Sensitivity to Exemption Duration.**    To further investigate the optimal duration of Phase 1 and validate the robustness of our method, we evaluated a wide range of exemption ratios: 0%, 5%, 10%, 15%, 20%, 40%, and 100% of the total training steps. Here, 0% represents immediate scaffolding, while 100% is equivalent to the vanilla GRPO baseline. The detailed results are presented in Table 11.

Our analysis reveals three key dynamics:

Table 10: Ablation study on the necessity of the guidance exemption period (Phase 1). We compare the full Scaf-GRPO framework against vanilla GRPO and a Scaf-GRPO variant without the initial exemption phase. Scores: pass@1 (%). Best results are in **bold**.

| Model | AIME 24 | AIME 25 | AMC | Minerva | MATH-500 | Olympiad | Gaokao2023en | Avg. |
|---|---|---|---|---|---|---|---|---|
| *Qwen2.5-Math-1.5B* | | | | | | | | |
| Vanilla GRPO | 13.3 | 10.0 | 47.5 | 28.3 | 72.2 | 34.8 | 57.4 | 37.6 |
| Scaf-GRPO (w/o Phase 1) | 10.0 | 10.0 | 57.5 | 27.5 | 71.4 | 36.3 | 57.9 | 38.7 |
| Scaf-GRPO | **20.0** | **13.3** | **60.0** | **29.1** | **73.4** | **36.6** | 57.9 | **41.5** |
| *Qwen2.5-Math-7B* | | | | | | | | |
| Vanilla GRPO | 30.0 | 13.3 | 60.0 | 33.4 | 75.8 | 41.3 | 62.6 | 45.2 |
| Scaf-GRPO (w/o Phase 1) | 23.3 | 13.3 | 70.0 | 34.2 | 78.4 | 41.2 | 63.1 | 46.2 |
| Scaf-GRPO | **43.3** | **20.0** | **70.0** | **36.4** | **80.0** | **43.3** | **63.4** | **50.9** |

- **Premature guidance is detrimental:** Short exemption periods (0% and 5%) yield the low scores among scaffolded variants (46.2% and 47.6%, respectively). This confirms that applying guidance too early prevents the model from learning to resolve minor errors independently.

- **Autonomy alone is insufficient:** The 100% setting (Vanilla GRPO) results in the worst overall performance (45.2%). This demonstrates the "learning cliff" effect: without scaffolding intervention, the model remains stuck on true-hard problems due to persistent zero-reward signals.

- **Robustness across a wide effective range:** Crucially, once the exemption period is sufficient (over 10%), Scaf-GRPO consistently achieves a high-performance plateau. As shown in the table, average scores for periods between 10% and 40% remain stable and high (ranging from 49.5% to the peak of 50.9%). This demonstrates that our framework is not overly sensitive to this hyperparameter; as long as the model has an initial phase to stabilize its capabilities, activating scaffolding effectively unlocks learning for the remaining intractable problems.

Table 11: Ablation study on the Guidance Exemption Period duration. We investigate the impact of varying the initial autonomous learning phase (from 0% to 100%) on pass@1 performance (%). The **15%** setting (Our Method) achieves the optimal trade-off. Best chosen configuration is highlighted in with background color. Best results are in **bold**.

| Exemption Period | AIME 24 | AIME 25 | AMC | Minerva | MATH-500 | Olympiad | Gaokao2023en | Avg. |
|---|---|---|---|---|---|---|---|---|
| *Qwen2.5-Math-7B* | | | | | | | | |
| 0% (w/o Phase 1) | 23.3 | 13.3 | 70.0 | 34.2 | 78.4 | 41.2 | 63.1 | 46.2 |
| 5% | 33.3 | 20.0 | 65.0 | 33.8 | 76.2 | 41.1 | 63.9 | 47.6 |
| 10% | 40.0 | 16.7 | 70.0 | 35.8 | 79.2 | 42.5 | 63.5 | 50.0 |
| 15% (Our Method) | 43.3 | **20.0** | **70.0** | **36.4** | 80.0 | 43.3 | 63.4 | **50.9** |
| 20% | **46.7** | 13.3 | 65.0 | 36.0 | **80.4** | **43.5** | **65.4** | 50.0 |
| 40% | 43.3 | 13.3 | 70.0 | 34.9 | 78.2 | 41.9 | 64.6 | 49.5 |
| 100% (Vanilla GRPO) | 30.0 | 13.3 | 60.0 | 33.4 | 75.8 | 41.3 | 62.6 | 45.2 |

## G.2   ABLATION ON DATA FILTERING

We determine our dataset composition by classifying problems based on the model's initial pass rate over $N = 8$ sampling attempts. The training samples are divided into three categories: (1) "Too Easy" (solved 8/8 times), which are discarded; (2) "Potentially Solvable" (solved 1-7/8 times); and (3) "Too Hard" (solved 0/8 times), which are fully retained.

To identify the optimal data mixture, we evaluate Qwen2.5-Math-7B under four configurations ranging from using the full dataset to training exclusively on hard samples. Table 12 presents the results. We observe that removing "Too Easy" samples improves performance over the full dataset baseline. Furthermore, randomly subsampling the "Potentially Solvable" category at a 50% ratio yields the highest Scaf-GRPO score of 50.9%. Based on these empirical results, we adopt this configuration (w/o "Too Easy" & w/o 50% Solvable) for our main experiments.

Table 12: Ablation study on dataset filtering strategies. We compare the Scaf-GRPO performance under different dataset settings. The selected configuration is highlighted.

| Dataset Setting | Scaf-GRPO |
|---|---|
| Full Dataset | 46.2 |
| w/o "Too Easy" | 48.8 |
| w/o "Too Easy" & w/o 50% Solvable (Ours) | **50.9** |
| w/o "Too Easy" & w/o 100% Solvable | 45.2 |

# H  HINT QUALITY ASSURANCE AND IMPACT ANALYSIS

To ensure the reliability of our scaffolding mechanism, we rigorously assess the quality of the generated hints and analyze their impact on model performance. We address this from two perspectives: (1) assessment using a multi-faceted rubric, and (2) empirical validation linking hint quality to downstream model accuracy.

**Assessment Methodology: A Multi-Faceted Rubric.**  While our final hints are generated using the DeepSeek-R1 model, we selected this generator after comparing it with other capable models, such as Qwen2.5-72B-Instruct. To objectively compare the quality of hint sets, we developed a rubric designed to assess hints not only for correctness but for their pedagogical value—specifically their ability to provide minimal, clear, and structured guidance. Our rubric evaluates hints across four key dimensions: (1) **Accuracy:** Factual and logical correctness of the mathematical content; (2) **Minimality:** The degree to which the hint provides the least necessary support, thereby preserving student autonomy; (3) **Clarity:** The use of unambiguous and easy-to-parse language; and (4) **Structural Coherence:** The logical progression and distinctness between abstract (knowledge/planning) and concrete (solution) tiers.

We employed an LLM-as-a-Judge (Gemini-2.5-Pro) to automatically score hint sets based on this rubric. The comparative results are presented in Table 13. The DeepSeek-R1 generated hints achieved a superior aggregate score, particularly excelling in structural coherence and minimality.

Table 13: LLM Judge scores for hint quality from different generator models (Scale 1–5).

| Hint Generator Model | LLM Judge Score |
|---|---|
| Qwen2.5-72B-Instruct | 4.38 |
| DeepSeek-R1 (Selected) | **4.72** |

**Empirical Validation: Hint Quality vs. Model Performance.**  To validate the hypothesis that higher-quality hints lead to better training outcomes, we trained the Qwen2.5-Math-7B model using both sets of hints (Qwen-generated vs. DeepSeek-generated) under identical training configurations.

The results, detailed in Table 14, demonstrate a clear positive correlation between the rubric score and final model performance. The model trained with the higher-rated DeepSeek-R1 hints achieved an average pass@1 score of 50.9%, compared to 49.0% for the model trained with Qwen hints. Notably, performance gains were most pronounced on difficult benchmarks like AIME24 (+6.6%) and Minerva (+2.6%), suggesting that high-quality, minimal scaffolding is particularly critical for mastering complex reasoning tasks. This validation confirms that our LLM-based evaluation rubric serves as a reliable proxy for hint utility.

---

**Prompt for hint evaluation**

```
**[ROLE & GOAL]**
You are an expert pedagogical evaluator and mathematics content specialist. Your task
is to objectively assess the quality of "Hint Sets" generated for a specific math
problem. You will evaluate how well these hints guide a student towards the solution
without giving away the answer directly, based on a strict multi-faceted rubric.

**[INPUT]**
```

```
I will provide you with:
1.  **Problem**: The original math question.
2.  **Ground-Truth Solution**: The correct step-by-step answer.
3.  **Generated Hints**: A set of hints (which may include a Planning Skeleton,
Knowledge Components, or Step-wise Breakdown) produced by an AI model.

**[SCORING RUBRIC]**
You must evaluate the hints on a scale of 1 to 5 for each of the following four
dimensions.

1.  **Accuracy (1-5)**: Factual and logical correctness.
    *   *5*: All information is mathematically precise, correct, and strictly aligns
    with the Ground-Truth Solution.
    *   *1*: Contains major mathematical errors, hallucinations, or leads the student
    down an incorrect path.

2.  **Minimality (1-5)**: Providing the least necessary support to preserve autonomy.
    *   *5*: Hints are concise and scaffolding-oriented. They guide the *reasoning
    process* (the "how" and "why") rather than just stating intermediate calculation
    results. They do not "spoon-feed" the answer.
    *   *1*: Hints are overly verbose, irrelevant, or give away the answer/key numbers
    immediately, destroying the student's opportunity to think.

3.  **Clarity (1-5)**: Unambiguous and easy-to-parse language.
    *   *5*: Language is simple, direct, and grammatically perfect. Mathematical
    notation is standard and consistent.
    *   *1*: Language is confusing, ambiguous, overly complex, or contains formatting
    errors that make it hard to read.

4.  **Structural Coherence (1-5)**: A logical flow from abstract to concrete.
    *   *5*: The hints follow a logical progression (e.g., Plan -> Concepts -> Steps).
    The order makes sense for a learner attempting to solve the problem.
    *   *1*: The hints are disorganized, random, or jump between unconnected ideas
    without a clear thread.

**[INSTRUCTIONS]**
1.  **Analyze the Input**: Read the Problem and Ground-Truth Solution to understand the
logic.
2.  **Verify the Hints**: Check the Generated Hints against the Solution for Accuracy.
3.  **Assess Pedagogy**: Evaluate if the hints explain *principles* (high
Minimality/Coherence) or just *calculate numbers* (low Minimality).
4.  **Draft Rationale**: For each dimension, write a brief justification for your
score.
5.  **Assign Scores**: Give a final integer score (1-5).

**[OUTPUT FORMAT]**
You MUST provide your response in the following structured format:

<EVALUATION_RATIONALE>
**Accuracy**: [Analysis...]
**Minimality**: [Analysis...]
**Clarity**: [Analysis...]
**Structural Coherence**: [Analysis...]
</EVALUATION_RATIONALE>

<SCORES>
Accuracy: [1-5]
Minimality: [1-5]
Clarity: [1-5]
Structural Coherence: [1-5]
</SCORES>

**[EXAMPLE]**

--- BEGIN EXAMPLE ---
**Problem**: Find the value of x if 2x + 3 = 9.
**Ground-Truth Solution**: Subtract 3 from both sides: 2x = 6. Divide by 2: x = 3.
**Generated Hints**:
1.  First, you need to isolate the variable x.
2.  Subtract 3 from 9 to get 6.
3.  Then divide 6 by 2 to get the final answer 3.
--- END EXAMPLE ---

**[EXPECTED OUTPUT FOR THE EXAMPLE]**

<EVALUATION_RATIONALE>
**Accuracy**: The math is correct. The operations lead to the right answer. (High)
**Minimality**: The hints are not minimal. Step 2 and 3 explicitly perform the
calculation ("get 6", "answer 3") instead of just suggesting the operation ("Subtract 3
from both sides"). This removes student autonomy. (Low)
```

```
**Clarity**: The language is very clear and easy to understand. (High)
**Structural Coherence**: The flow is logical (Method -> Calculation -> Result). (High)
</EVALUATION_RATIONALE>

<SCORES>
Accuracy: 5
Minimality: 2
Clarity: 5
Structural Coherence: 5
</SCORES>

---

**[TASK]**
Now, please evaluate the following input strictly following all instructions.

**Problem**:
{question}

**Ground-Truth Solution**:
{solution}

**Generated Hints**:
{hints}
```

Table 14: Impact of hint source quality on model performance. We compare the final pass@1 (%) of Qwen2.5-Math-7B trained using hints generated by Qwen2.5-72B-Instruct versus our selected DeepSeek-R1. The best results are in **bold** and the selected configuration is highlighted.

| Hint Source | AIME 24 | AIME 25 | AMC | Minerva | MATH-500 | Olympiad | Gaokao2023en | Avg. |
|---|---|---|---|---|---|---|---|---|
| *Qwen2.5-Math-7B* | | | | | | | | |
| w/ Qwen2.5-72B-Instruct Hints | 36.7 | 20.0 | 67.5 | 33.8 | 79.4 | 42.2 | 63.1 | 49.0 |
| w/ DeepSeek-R1 Hints (Ours) | **43.3** | **20.0** | **70.0** | **36.4** | **80.0** | **43.3** | **63.4** | **50.9** |

# I   ADDITIONAL EVALUATION AND ANALYSIS

This section provides extensive supplementary evidence to corroborate the effectiveness of Scaf-GRPO. To demonstrate that our method's improvements are robust and not limited to specific metrics, we report performance across varying sampling configurations and compare against an expanded suite of competitive baselines. We also delve deeper into the model's generalization capabilities on challenging OOD tasks and analyze its behavior when integrated with or compared against other method (e.g., DeepScaleR Luo et al. (2025), DAPO Yu et al. (2025)). The results consistently highlight the stability of Scaf-GRPO and its ability to establish a well performance on complex reasoning tasks.

**Robustness Analysis: From Pass@1 to Avg@16.**   While the pass@1 metric via greedy decoding is a standard benchmark, it can exhibit fluctuations due to the inherent stochasticity of LLM generation. To ensure the reliability of our results and demonstrate that our improvements are not artifacts of variance, we conducted a comprehensive re-evaluation using the **avg@16** metric. For this analysis, we generated 16 distinct samples for each problem with a temperature of 0.6 and top-p of 0.95, averaging the binary success scores. As presented in Table 15, Scaf-GRPO consistently maintains a significant performance advantage over the Vanilla GRPO baseline and other methods across all model architectures. Notably, on Qwen2.5-Math-7B, our method achieves an average score of 48.13%, surpassing strong baselines like Oat-Zero Liu et al. (2025a) and LUFFY Yan et al. (2025). These results confirm that the capability gains fostered by Scaf-GRPO are robust and stable under rigorous multi-sample evaluation.

**Comparison with Expanded Baselines.**   To rigorously contextualize the performance of Scaf-GRPO within the broader landscape of mathematical reasoning, we expand our comparative analysis to include a diverse set of representative 7B-parameter models. We classify these baselines into two categories: (1) Short Chain-of-Thought (Short CoT) baselines, represented by standard base models prompting with concise reasoning steps; and (2) Long Chain-of-Thought (Long CoT) baselines, comprising recent state-of-the-art models optimized for extended reasoning via advanced instruction

Table 15: Overall performance on seven benchmarks using **avg@16** metric. We compare our method, SCAF-GRPO, against vanilla GRPO baselines across diverse architectures. Best results are in **bold**. The background color of Scaf-GRPO cells indicates performance change vs. Vanilla GRPO (**green** for improvement, **red** for decline).

| Model | AIME 24 | AIME 25 | AMC | Minerva | MATH-500 | Olympiad | Gaokao2023en | Avg. |
|---|---|---|---|---|---|---|---|---|
| *Qwen2.5-Math-1.5B* | | | | | | | | |
| Vanilla GRPO | 11.8 | 8.2 | 42.2 | 28.5 | 72.4 | 35.0 | 57.4 | 36.5 |
| Scaf-GRPO | **14.7** | **10.8** | **50.9** | **31.4** | **74.8** | **37.4** | **58.8** | **39.8** |
| *Qwen2.5-Math-7B* | | | | | | | | |
| Vanilla GRPO | 30.7 | 10.8 | 61.1 | 33.4 | 74.2 | 41.1 | 62.6 | 44.8 |
| SimpleRL-Zero Zeng et al. (2025) | 22.5 | 11.0 | 49.4 | 31.8 | 76.2 | 38.2 | 61.0 | 41.5 |
| Oat-Zero Liu et al. (2025a) | 30.7 | 11.5 | 62.7 | 35.2 | 78.2 | 41.4 | 63.0 | 46.1 |
| LUFFY Yan et al. (2025) | 31.9 | 12.0 | 60.4 | 33.0 | 74.3 | 40.6 | 61.9 | 44.9 |
| Scaf-GRPO | **35.6** | **14.6** | **63.8** | **36.8** | **79.1** | **42.1** | **64.9** | **48.1** |
| *Qwen2.5-7B* | | | | | | | | |
| Vanilla GRPO | 11.0 | 9.5 | 53.0 | 36.7 | 76.8 | 40.4 | 64.5 | 41.7 |
| Scaf-GRPO | **15.6** | **11.0** | **54.2** | **37.0** | **77.6** | **41.2** | **64.8** | **43.1** |
| *Llama-3.2-3B-Instruct* | | | | | | | | |
| Vanilla GRPO | 7.6 | 0.0 | 28.7 | 18.7 | 50.5 | 19.2 | 46.1 | 24.4 |
| Scaf-GRPO | **9.6** | **0.3** | **32.8** | **19.1** | **54.7** | **21.3** | **46.4** | **26.3** |
| *DeepSeek-R1-Distill-Qwen-1.5B* | | | | | | | | |
| Vanilla GRPO | 27.1 | 22.9 | 65.4 | 32.5 | 83.8 | 54.0 | 75.3 | 51.7 |
| Scaf-GRPO | **28.5** | **25.3** | **75.1** | **35.0** | **85.7** | **54.5** | **76.3** | **54.3** |

tuning or reinforcement learning, including Eurus-2-PRIME Cui et al. (2025), rStar-Math Guan et al. (2025), and OpenReasoner-Zero Hu et al. (2025). As detailed in Table 16, Scaf-GRPO consistently outperforms both categories. It not only surpasses base models by a wide margin but also exceeds other 7B RL-tuned models, particularly on high-difficulty benchmarks like AIME 24 (43.3%) and AMC 23 (70.0%).

Table 16: Comparison with Short CoT and Long CoT baselines on 7B models. We report the pass@1 accuracy (%) on various mathematical benchmarks. **Scaf-GRPO (Ours)** achieves the best average performance among all 7B models. The best results in each column are highlighted in **bold**.

| Category | Model | AIME 24 | AIME 25 | AMC | Minerva | MATH-500 | Olympiad | Gaokao | Avg. |
|---|---|---|---|---|---|---|---|---|---|
| Short CoT | Qwen-2.5-7B-Base | 10.0 | 6.7 | 37.5 | 26.4 | 61.8 | 34.4 | 42.6 | 31.3 |
| | Qwen-2.5-Math-7B-Base | 13.3 | 13.3 | 42.5 | 16.5 | 53.6 | 18.2 | 35.1 | 27.5 |
| Long CoT | Eurus-2-7B-PRIME Cui et al. (2025) | 26.7 | 16.7 | 55.0 | **38.6** | 74.4 | 34.7 | 59.3 | 43.6 |
| | rStar-Math-7B Guan et al. (2025) | 26.7 | 13.9 | 47.5 | 32.2 | 78.4 | **47.1** | 59.2 | 43.6 |
| | SimpleRL-Zero Zeng et al. (2025) | 23.3 | 13.3 | 55.0 | 31.6 | 76.8 | 37.2 | 60.8 | 42.6 |
| | LUFFY Yan et al. (2025) | 33.3 | 16.7 | 62.5 | 33.8 | 75.2 | 41.7 | 62.7 | 46.6 |
| | Oat-Zero Liu et al. (2025a) | 30.0 | 16.7 | 62.5 | 34.6 | 78.0 | 41.0 | 62.9 | 46.5 |
| | OpenReasoner-Zero Hu et al. (2025) | 26.7 | 16.7 | 50.0 | 35.6 | 77.4 | 41.6 | **67.2** | 45.0 |
| | **Scaf-GRPO (Ours)** | **43.3** | **20.0** | **70.0** | 36.4 | **80.0** | 43.3 | 63.4 | **50.9** |

**Integration with Advanced Models: DeepScaleR.** To demonstrate the compatibility of Scaf-GRPO with other training paradigms and its ability to enhance data efficiency, we conducted an experiment using the DeepScaleR-1.5B-Preview Luo et al. (2025) model. Since this model has already converged using standard GRPO on the DeepScaleR-Preview-Dataset, simply continuing training with Vanilla GRPO on the same data results in performance regression, as indicated by the drop in the average score (from 55.1% to 54.0%) in Table 17. In contrast, Scaf-GRPO successfully extracts further value from the identical dataset. By leveraging the hint-guided mechanism to unlock previously inaccessible reasoning paths, our method achieves a clear positive gain, raising the average score to 59.1% and showing significant improvements on challenging benchmarks like AMC 23 and OlympiadBench. This confirms that Scaf-GRPO can effectively complement existing advanced checkpoints to maximize data utilization.

**Comparison with DAPO.** Both Scaf-GRPO and DAPO (Yu et al., 2025) aim to mitigate the vanishing gradient problem caused by consistent failure on difficult queries. However, the two methods

Table 17: Performance comparison of Scaf-GRPO versus Vanilla GRPO when continuing training the DeepScaleR-1.5B-Preview model. We report the pass@1 (%) accuracy on various benchmarks. The best results are in **bold**.

| Method | AIME 24 | AIME 25 | AMC | Minerva | MATH-500 | Olympiad | Gaokao2023en | Avg. |
|---|---|---|---|---|---|---|---|---|
| *DeepScaleR-1.5B-Preview Luo et al. (2025)* | | | | | | | | |
| Base Model | **43.1** | 30.0 | 73.6 | 30.2 | **87.8** | 50.5 | 70.4 | 55.1 |
| + Vanilla GRPO | 40.0 | 30.0 | 72.0 | 30.5 | 86.9 | 49.1 | 69.2 | 54.0 |
| + Scaf-GRPO (Ours) | 40.0 | **33.3** | **85.0** | **33.5** | **87.8** | **55.0** | **79.0** | **59.1** |

diverge fundamentally in their strategy. DAPO employs a filtering mechanism (Dynamic Sampling) that effectively omits these zero-gradient samples to maintain training stability. In contrast, Scaf-GRPO adopts an intervention strategy: rather than discarding hard problems, we utilize hierarchical hints to guide the model toward a correct solution, transforming these failures into valuable learning signals. As shown in Table 18, this active approach yields superior results. On the Qwen2.5-Math-7B model, Scaf-GRPO outperforms DAPO across all benchmarks, achieving a higher average accuracy (50.9% vs. 48.5%). This suggests that scaffolding the model to conquer its hardest challenges is more effective for capability advancement than simply stabilizing the training distribution.

Table 18: Performance comparison on Qwen2.5-Math-7B (pass@1, %). We compare our Scaf-GRPO against Vanilla GRPO and DAPO baselines. The best results are in **bold**.

| Method | AIME 24 | AIME 25 | AMC | Minerva | MATH-500 | Olympiad | Gaokao23 | Avg. |
|---|---|---|---|---|---|---|---|---|
| *Qwen2.5-Math-7B* | | | | | | | | |
| Vanilla GRPO | 30.0 | 13.3 | 60.0 | 33.4 | 75.8 | 41.3 | 62.6 | 45.2 |
| DAPO Yu et al. (2025) | 36.7 | **20.0** | 62.5 | 35.6 | 79.6 | 42.3 | 63.0 | 48.5 |
| Scaf-GRPO (Ours) | **43.3** | **20.0** | **70.0** | **36.4** | **80.0** | **43.3** | **63.4** | **50.9** |

