# OpenReview forum: "Scaf-GRPO: Scaffolded Group Relative Policy Optimization for Enhancing LLM Reasoning"
_ICLR.cc/2026/Conference — ICLR 2026 Poster_

### Official Review · Reviewer_9rLe · 2025-10-26

**Soundness:** 3
**Presentation:** 3
**Contribution:** 2
**Rating:** 6
**Confidence:** 4

**Summary:**

This paper introduces Scaffolded Group Relative Policy Optimization (Scaf-GRPO), a novel framework designed to enhance the reasoning capabilities of Large Language Models (LLMs) by overcoming the "learning cliff" problem in reinforcement learning. This problem arises when an LLM consistently fails to solve difficult problems, resulting in a zero-reward signal that provides no learning gradient. Scaf-GRPO addresses this by employing a two-phase training process. The first phase, a "guidance exemption period," allows the model to learn autonomously on easier problems. In the second phase, for "true-hard" problems where learning has stagnated, the framework provides minimal, hierarchical in-prompt hints (Knowledge -> Planning -> Solution) until the model can generate a correct solution. By augmenting the training batch with this minimally-guided successful trajectory, Scaf-GRPO restores the learning gradient while preserving the on-policy nature of the learning algorithm and the model's exploratory autonomy. The authors demonstrate through extensive experiments on various math benchmarks that their method significantly outperforms vanilla GRPO and other prefix-based guidance methods.

**Strengths:**

1. The paper clearly formalizes the "learning cliff" phenomenon (persistent zero-reward signals) as a critical bottleneck in training LLMs with reinforcement learning. The proposed solution, inspired by the pedagogical concept of scaffolding, is both elegant and intuitive. Instead of forcing the model down a fixed path with prefix-based solutions, Scaf-GRPO provides minimal, progressively more concrete hints, which preserves the model's ability to explore its own reasoning paths. This is a significant conceptual departure from prior work like LUFFY that relies on off-policy prefix-continuation.

2. The authors conduct a meticulous set of ablation studies that convincingly validate their key design choices. Table 2 effectively demonstrates that each level of the hint hierarchy (Knowledge, Planning, Solution) is a necessary and complementary component of the framework. Furthermore, the analysis in Appendix G (Table 7) confirms the necessity of the "guidance exemption period," showing that applying scaffolding from the start leads to inferior performance by fostering over-reliance on hints. These studies provide strong evidence that the framework's components are well-justified and contribute to its overall effectiveness.

**Weaknesses:**

1. The framework's success is contingent on the availability of a high-quality, three-tiered hint hierarchy. As the authors acknowledge, generating these hints requires a "non-trivial data preparation effort," involving a powerful teacher model (DeepSeek-R1) and a highly structured prompting process. This dependency limits the scalability and practical applicability of Scaf-GRPO to domains where such structured hints can be easily and reliably generated. It moves the difficulty from the RL training phase to a complex and potentially expensive data curation phase.

2. The hints are generated by a single, powerful teacher model. This introduces the risk that the scaffolding will simply teach the student model the specific reasoning patterns and biases of the teacher. While Scaf-GRPO aims to preserve exploratory autonomy, the guidance is still fundamentally rooted in the teacher's "worldview." The paper does not explore the effects of hint quality or potential inaccuracies in the teacher-generated hints on the final performance of the student model.

**Questions:**

1. Regarding training efficiency and comparative fairness: During the training process, what is the approximate proportion of "all-failure" batches that trigger hierarchical hint-guided exploration? For these batches, Scaf-GRPO performs multiple on-policy rollouts with hints until a successful solution is found. In contrast, Vanilla GRPO conducts only one ineffective rollout when encountering the same situation. This suggests that Scaf-GRPO consumes more computational resources (time and computing power) when addressing challenging problems. Under the same training duration or total computational budget, can Scaf-GRPO still maintain a significant advantage?

2. Comparison of effectiveness with DAPO.

3. Can Scaf-Rollout be integrated as a component with current state-of-the-art model methods, such as DeepScaler and Skywork?

4. This work lacks detailed comparisons with existing mathematical reasoning studies, such as representative works on long-chain and short-chain reasoning. These results on the test set are not particularly good.

5. The pass@1 metric exhibits considerable fluctuations and requires repeated evaluations to obtain a reliable average.

6. The "guidance exemption period" is empirically set to the initial 15% of training steps. How was this value determined? Is it a sensitive hyperparameter, or does the method perform robustly across a range of values?

7. Figure 5 illustrates a compelling case of a model internalizing a skill. How frequently does this "graduation" from hint-dependency to autonomous competence occur in practice? Is it possible to track this metric during training to dynamically adjust when to offer hints for certain problem types?

---

> ### Author Response · Authors · 2025-11-21
> **Response to Weakness 1 (1/9)**
>
> Thank you for your thoughtful and constructive feedback. We are encouraged that you found our problem formulation clear, our scaffolding-inspired solution elegant and intuitive, and our experimental validation meticulous and convincing. We appreciate your time and effort in providing detailed comments, which have helped us improve both the clarity and quaity of our paper. Below, we respond to each point in detail.
>
> ## Response to Weakness 1: Scalability of Data and Hint Generation
>
> Thank you for raising these important points. We address your concerns by explaining: (1) that the necessary data for generating hints is already widely available across numerous reasoning domains, making the approach scalable, and (2) that the hint generation task is a straightforward and manageable process where the teacher model primarily summarizes existing solutions, rather than creating hints from scratch.
>
> ### (1) Availability of Data for Scalable Hint Generation
>
> Scalable hint generation is possible because of the wide availability of datasets that include step-by-step solutions. We can distill these existing solutions into hints. To illustrate this, you can consider the following examples from diverse domains such as General Reasoning, Mathematics, and Code. The availability of these large-scale datasets is what makes our proposed method broadly applicable and scalable.
>
> *   **General Reasoning**: Datasets like MMLU-Pro[1] provide questions across many fields with detailed solutions, offering a direct foundation for creating tiered hints.
> *   **Mathematics**: Large-scale datasets such as NuminaMath-CoT[2] contain 860k math problems where each solution is already formatted in a Chain-of-Thought (CoT) style.
> *   **Code**: Dataset like CodeForces[3] have extensive solutions for each code questions.
>
> ### (2) Simplicity of the Hint Generation Process
> The teacher model's role is straightforward: to summarize an existing solution into predefined hint tiers (Knowledge, Planning, Solution). This is a highly constrained and manageable task.
>
> By performing this summarization as a one-time, offline step, we convert the challenge from an intractable training problem into a simple data preparation task. This upfront investment makes the subsequent RL training vastly more efficient.
>
> We hope this explanation clarifies the weakness. Thank you again for your valuable feedback.
>
>
> **References:**
>
> [1] Wang, Y., Ma, X., Zhang, G., Ni, Y., Chandra, A., Guo, S., Ren, W., Arulraj, A., He, X., Jiang, Z., et al. (2024). Mmlu-pro: A more robust and challenging multi-task language understanding benchmark. *Advances in Neural Information Processing Systems, 37*, 95266-95290.
>
> [2] Li, J., Beeching, E., Tunstall, L., Lipkin, B., Soletskyi, R., Huang, S., Rasul, K., Yu, L., Jiang, A. Q., Shen, Z., et al. (2024). Numinamath: The largest public dataset in ai4maths with 860k pairs of competition math problems and solutions. *Hugging Face repository, 13*(9), 9.
>
> [3] Li, Y., Choi, D., Chung, J., Kushman, N., Schrittwieser, J., Leblond, R., Eccles, T., Keeling, J., Gimeno, F., Dal Lago, A., Hubert, T., Choy, P., de Masson d'Autume, C., Babuschkin, I., Chen, X., Huang, P. S., Welbl, J., Gowal, S., Cherepanov, A., Molloy, J., Mankowitz, D., Sutherland Robson, E., Kohli, P., de Freitas, N., Kavukcuoglu, K., & Vinyals, O. (2022). Competition-level code generation with AlphaCode. *arXiv preprint arXiv:2203.07814*.

---

> ### Author Response · Authors · 2025-11-21
> **Response to Weakness 2 (2/9)**
>
> ## Response to Weakness 2: On the Quality and Control of LLM-Generated Hints
>
> Thank you for raising thess questions. We address the quality control of our generated hints from two perspectives: (1) assessment using a multi-faceted rubric, and (2) empirical validation linking hint quality to model performance.
>
> ### (1) Assessment Methodology: A Multi-Faceted Rubric
>
> In our paper, the final hints come from the DeepSeek-R1 model. Our selection process involves experimenting with other powerful models, such as Qwen2.5-72B-Instruct, to generate the hints. To objectively compare the quality of the generated hint sets, we employ a multi-faceted rubric. This rubric is designed to assess hints not just for correctness, but for their ability to provide minimal, clear, and structured guidance.
>
> Our rubric assesses hints across four key dimensions:
> *   **Accuracy**: Factual and logical correctness.
> *   **Minimality**: Providing the least necessary support to preserve autonomy.
> *   **Clarity**: Unambiguous and easy-to-parse language.
> *   **Structural Coherence**: A logical flow from abstract to concrete.
>
>
> To apply this rubric, we use an LLM Judge (Gemini-2.5-Pro) to automatically score all the hints. This method allows us to compare the quality of hints generated by different models, such as DeepSeek-R1 (our final choice) and Qwen2.5-72B-Instruct. The quality scores are shown below:
>
> > Table: LLM Judge scores for hint quality from different generator models.
>
> | Hint Generator Model | LLM Judge Score |
> | :--- | :---: |
> | Qwen2.5-72B-Instruct | 4.38 |
> | DeepSeek-R1 (Used) | 4.72 |
>
> ### (2) Empirical Validation: Hint Quality vs. Model Performance
>
> Next, to validate if better hint scores lead to better model performance, we train the Qwen2.5-Math-7B model on both sets of hints, using the exact same training settings. The final test results show a clear link to our hint quality scores:
>
> > Table: Final performance (Pass@1 %) of Qwen2.5-Math-7B trained on hints from different sources.
>
> | Model & Hint Source | AIME24 | AIME25 | AMC |Minerva| MATH-500 | Olympiad | Gaokao23 | Avg. |
> | :--- | :---: | :---: | :---: | :---: | :---: | :---: | :---: |  :---: |
> | Qwen2.5-Math-7B (w/ Qwen Hints) | 36.7 | 20.0 | 67.5 | 33.8 | 79.4 | 42.2 | 63.1 | 49.0 |
> | Qwen2.5-Math-7B (w/ DeepSeek-R1 Hints) | 43.3 | 20.0 | 70.0 |36.4| 80.0 | 43.3 | 63.4 | 50.9 |
>
> As you can see from the results, there is a clear connection between the hint quality score from our rubric and the final performance of the trained model. This shows that the hint set from DeepSeek-R1 is better. This also suggests that using an LLM-based judge with our rubric is a good proxy for measuring hint quality.
>
> Thank you again for this very insightful question. We have added the detailed hint quality comparison, the specific prompt used for the LLM judge, and these experimental results to the revised manuscript. Please refer to lines 431-448 and Appendix H. We believe this will make our methodology more transparent and our findings more complete.

---

> ### Author Response · Authors · 2025-11-21
> **Response to Question 1 (3/9)**
>
> ## Response to Question 1: Training efficiency and fairness of comparison
>
> Thank you for raising these important points regarding the training efficiency and fairness of Scaf-GRPO. We address your questions by providing: (1) data on the trigger proportion of our guidance mechanism, (2) a direct comparison of the overall performance under a fixed computational budget, and (3) a broader efficiency comparison with off-policy alternatives.
>
>
>
> ### (1) Trigger Proportion for Hint-Guided Exploration
>
> We use the training of Qwen2.5-Math-7B as a concrete example. During this training, with a batch size of 256 problems per step, an average of 44.64 problems per batch triggered hint-guided exploration after failing their initial attempt. This corresponds to **17.4%** of the problems in an average batch.
>
> ### (2) Performance Advantage Within a Fixed Computational Budget: Scaf-GRPO vs. Vanilla GRPO
>
> Under a fixed computational budget, Scaf-GRPO demonstrates superior performance and training efficiency compared to Vanilla GRPO. The table below presents the final results and the total time required for each method to reach its best-performing checkpoint on the Qwen2.5-Math-7B model.
>
> > Table: Overall Training Efficiency and Performance between Vanilla GRPO and Scaf-GRPO
>
> | Method | Best Avg. Score (%) | Time to Best Checkpoint (8 GPUs) | Peak Memory (GB) |
> | :--- | :---: | :---: | :---: |
> | Vanilla GRPO | 45.2 | ~13 hours | ~72 |
> | Scaf-GRPO (Ours) | 50.9 | ~12 hours | ~73 |
>
> The data clearly shows that Scaf-GRPO not only achieves a substantially **higher score (50.9% vs. 45.2%)** but does so in **less total time (~12 hours vs. ~13 hours)**.
>
> The reason for this enhanced efficiency is that Scaf-GRPO makes a strategic trade-off. Although its per-step computation time can be slightly longer in certain cases, it successfully converts challenging, zero-reward steps into valuable learning opportunities. By obtaining more effective learning signals from each step, Scaf-GRPO ensures that the overall training process is more productive, leading to faster convergence and a significantly better final result.
>
> Thank you again for this very insightful question. We have incorporated these findings into the revised manuscript to provide a more complete picture of Scaf-GRPO's practical efficiency. Please refer to lines 498-507 and Table 5.

---

> ### Author Response · Authors · 2025-11-21
> **Response to Question 2 (4/9)**
>
> ## Response to Question 2: Comparison with DAPO and Methodological Advantages
>
> We sincerely thank you for this insightful question regarding the comparison between our Scaf-GRPO and DAPO. We appreciate the opportunity to clarify the fundamental differences in our philosophies and demonstrate the advantages of our approach.
>
> As you noted, both Scaf-GRPO and DAPO’s Dynamic Sampling mechanism address a critical challenge in RLVR: the "learning cliff" or "zero-gradient" problem. This occurs when all trajectories for a given prompt receive the same reward (e.g., all are incorrect), causing the advantage calculation to collapse to zero and stalling the learning process. While both methods identify this issue, our solutions diverge significantly.
>
> ### (1) DAPO's Approach: Omission of the Problems
> DAPO’s Dynamic Sampling tackles this by filtering out these non-informative, zero-gradient samples and continuing to sample until the batch is filled with "informative" problems that contain a mix of rewards. This ensures every sample contributes a meaningful gradient, improving training stability.
>
> However, this strategy's core action is one of omission. The most difficult problems, which the model consistently fails on, are systematically excluded from training updates. While this avoids noisy gradients, it also means the model is never explicitly taught how to conquer its toughest challenges.
>
> ### (2) Scaf-GRPO's Approach: Intervention on Hard Problems
> In contrast, Scaf-GRPO adopts a principle of intervention over omission. Instead of discarding these "true-hard" problems, our framework treats them as critical learning opportunities. When a learning cliff is detected, Scaf-GRPO injects minimal, hierarchical hints to "scaffold" the model's reasoning, enabling it to generate a correct solution.
>
> This key difference accomplishes two things:
> *   It transforms an unproductive, zero-gradient sample into a highly valuable, on-policy learning signal.
> *   It directly teaches the model how to reason through the very problems it finds most difficult, pushing the frontier of its capabilities rather than just optimizing within its existing comfort zone.
>
> To provide a direct empirical comparison, we benchmarked our results on the Qwen2.5-Math-7B model against DAPO. As shown in the table below, Scaf-GRPO demonstrates beter performance across a wide range of challenging mathematics benchmarks.
>
> Table: Performance Comparison on Qwen2.5-Math-7B (pass@1, %)
> | Method                          | AIME24 | AIME25 | AMC  | Minerva | MATH-500 | Olympiad | Gaokao23 | Avg. |
> | ------------------------------- | :----: | :----: | :--: | :-----: | :------: | :------: | :------: | :------: |
> | Vanilla GRPO                    | 30.0   | 13.3   | 60.0 |  33.4   |   75.8   |   41.3   |   62.6   |   45.2   |
> | DAPO | 36.7   | 20.0   | 62.5 |  35.6   |   79.6   |   42.3   |   63.0   |   48.5   |
> | Scaf-GRPO (Ours)            | 43.3   | 20.0   | 70.0 |  36.4   |   80.0   |   43.3   |   63.4   |   50.9   |
>
> The results clearly show that while DAPO provides a notable improvement over a vanilla GRPO baseline, our intervention-based scaffolding strategy achieves significantly higher overall performance (+5% relative average gain). By learning from the hardest problems instead of avoiding them, Scaf-GRPO unlocks a higher level of reasoning ability.
>
> In summary, we believe that actively guiding a model through its failures provides a more robust and effective path to advancing its reasoning frontier. Thank you again for prompting this important comparison; we have included the detailed analysis contrasting our approach with DAPO in Appendix I (lines 1511-1529) and Table 18 of the revised manuscript.

---

> ### Author Response · Authors · 2025-11-21
> **Response to Question 3 (5/9)**
>
> ## Response to Question 3: Integrating Scaf-GRPO with Methods like DeepScaler
>
> We sincerely thank you for this insightful question regarding the potential integration of Scaf-GRPO with advanced training methods such as DeepScaler and Skywork. The answer is a definitive yes. We address your comment in two parts: (1) empirical experimental evidence using DeepScaleR-1.5B-Preview with continued training, and (2) a technical roadmap for integrating Scaf-GRPO into the training paradigms of these methods.
>
> ### (1) Experimental Verification on DeepScaleR-1.5B-Preview
>
> To validate the compatibility and effectiveness of our method, we employ the DeepScaleR-1.5B-Preview model as our initialization. We continue training this model using its original training source, the DeepScaleR-Preview-Dataset.
>
> DeepScaleR-1.5B-Preview represents a model that has already reached convergence via GRPO on the DeepScaleR dataset. As a result, continuing to train this checkpoint with Vanilla GRPO on the same data leads to a decline in performance. In contrast, Scaf-GRPO achieves further performance gains on this identical setup.
>
> As shown in the table below, Vanilla GRPO suffers from performance regression, while Scaf-GRPO secures additional improvements. This comparison demonstrates that the original GRPO algorithm has limitations in fully exploiting the dataset. However, by incorporating the hint mechanism, Scaf-GRPO successfully extracts remaining information and value from the same training data that standard GRPO fails to utilize.
>
> > Table: Performance comparison of Scaf-GRPO versus Vanilla GRPO when continuing training the DeepScaleR-1.5B-Preview model.
>
> | Benchmark | DeepScaleR-1.5B-Preview (Base) | + Vanilla GRPO | **+ Scaf-GRPO (Ours)** |
> | :--- | :--- | :--- | :--- |
> | **AIME 24** | 43.1 | 40.0 | **40.0** |
> | **AIME 25** | 30.0 | 30.0 | **33.3** |
> | **AMC 23** | 73.6 | 72.0 | **85.0** |
> | **Minerva** | 30.2 | 30.5 | **33.5** |
> | **MATH-500** | 87.8 | 86.9 | **87.8** |
> | **Olympiad** | 50.5 | 49.1 | **55.0** |
> | **Gaokao2023en** | 70.4 | 69.2 | **79.0** |
> | **Average** | **55.1** | 54.0 | **59.1** |
>
>
> ### (2) Methodological Integration with Training Paradigms
>
> Beyond simply finetuning existing checkpoints, Scaf-GRPO can be seamlessly integrated into the core training loops of methods like DeepScaler. Taking DeepScaler as a prime example, its core contribution is **Iterative Context Lengthening**, where the model is trained in stages with progressively longer context windows (e.g., 8K → 16K → 24K). The transition to a longer context typically happens when the model's performance plateaus at the current length.
>
> Scaf-GRPO fits naturally into this workflow as a mechanism to overcome these plateaus before increasing computational cost. We propose the following integrated training pipeline:
>
> 1.  **Standard Training Phase**: Begin training with a shorter context window (e.g., 8K) using the standard RL objective, consistent with the DeepScaler protocol.
> 2.  **Scaffolded Intervention**: When the learning curve flattens (the "learning cliff"), instead of immediately expanding the context window, activate Scaf-GRPO. By injecting hierarchical hints, we help the model solve difficult problems that fit within the current 8K context but are too complex for the model to solve autonomously.
> 3.  **Maximize Efficiency**: This step ensures the model fully masters reasoning skills possible at the 8K length.
> 4.  **Context Expansion**: Once the model reaches a limit where even scaffolding cannot yield further improvements, we then proceed to the next stage (e.g., 16K context).
>
> This hybrid approach combines the efficiency of Scaf-GRPO in maximizing data utility with the scalability of DeepScaler's context lengthening, leading to a more robust final model. We have included the results of this in Appendix I (lines 1500-1509) and Table 17 of the revised manuscript.

---

> ### Author Response · Authors · 2025-11-21
> **Response to Question 4 (6/9)**
>
> ## Response to Question 4: Insufficient Comparisons with Existing Studies
>
> Thank you for raising these questions. We provide the following clarifications regarding our comparative analysis and performance metrics.
>
> To address the concern regarding the comparison with existing studies, we have expanded our evaluation to include a broader set of representative baselines. Following Table presents a detailed comparison between Scaf-GRPO and two categories of models: (1) Short Chain-of-Thought Base models, and (2) Recent state-of-the-art reinforcement learning and reasoning models (Long CoT) of similar scale (7B parameters). All results reported in this comparison follow the evaluation protocol described in our main paper.
>
> > Table: Comparison with Short CoT and Long CoT baselines on 7B models.
>
> | Category | Model | AIME 24 | AIME 25 | AMC 23 | Minerva | MATH-500 | Olympiad | Gaokao | Avg. |
> | :--- | :--- | :---: | :---: | :---: | :---: | :---: | :---: | :---: | :---: |
> | Short CoT | Qwen-2.5-7B-Base | 10.0 | 6.7 | 37.5 | 26.4 | 61.8 | 34.4 | 42.6 | 31.3 |
> | | Qwen-2.5-Math-7B-Base | 13.3 | 13.3 | 42.5 | 16.5 | 53.6 | 18.2 | 35.1 | 27.5 |
> | Long CoT | Eurus-2-7B-PRIME [1] | 26.7 | 16.7 | 55.0 | 38.6 | 74.4 | 34.7 | 59.3 | 43.6 |
> | | rStar-Math-7B [2] | 26.7 | 13.9 | 47.5 | 32.2 | 78.4 | 47.1 | 59.2 | 43.6 |
> | | SimpleRL-Zero [3] | 23.3 | 13.3 | 55.0 | 31.6 | 76.8 | 37.2 | 60.8 | 42.6 |
> | | LUFFY [4] | 33.3 | 16.7 | 62.5 | 33.8 | 75.2 | 41.7 | 62.7 | 46.6 |
> | | Oat-Zero [5] | 30.0 | 16.7 | 62.5 | 34.6 | 78.0 | 41.0 | 62.9 | 46.5 |
> | | OpenReasoner-Zero [6] | 26.7 | 16.7 | 50.0 | 35.6 | 77.4 | 41.6 | 67.2 | 45.0 |
> | | Scaf-GRPO (Ours) | 43.3 | 20.0 | 70.0 | 36.4 | 80.0 | 43.3 | 63.4 | 50.9 |
>
> We have included these results in Appendix I (lines 1453-1485) and Table 16 of the revised manuscript. As shown, Scaf-GRPO consistently outperforms other leading methods, and we hope this additional data clarifies the competitive positioning of our work.
>
> References:
>
> [1] Cui, G., Yuan, L., Wang, Z., et al. (2025). Process reinforcement through implicit rewards. arXiv preprint arXiv:2502.01456.
> [2] Guan, X., Zhang, L. L., Liu, Y., et al. (2025). rStar-Math: Small LLMs Can Master Math Reasoning with Self-Evolved Deep Thinking. arXiv preprint arXiv:2501.04519.
> [3] Zeng, W., Huang, Y., Liu, Q., et al. (2025). Simplerl-zoo: Investigating and taming zero reinforcement learning for open base models in the wild. arXiv preprint arXiv:2503.18892.
> [4] Yan, J., Li, Y., Hu, Z., et al. (2025). Learning to Reason under Off-Policy Guidance. arXiv preprint arXiv:2504.14945.
> [5] Liu, Z., Chen, C., Li, W., et al. (2025). There May Not be Aha Moment in R1-Zero-like Training — A Pilot Study. Notion Blog.
> [6] Hu, J., Zhang, Y., Han, Q., et al. (2025). Open-reasoner-zero: An open source approach to scaling up reinforcement learning on the base model. arXiv preprint arXiv:2503.24290.

---

> ### Author Response · Authors · 2025-11-21
> **Response to Question 5 (7/9)**
>
> ## Response to Question 5: Addressing the stability of the pass@1 evaluation metric
>
> We sincerely thank you for your insightful feedback regarding the evaluation metric. You have raised a very important point about the potential variance associated with the `pass@1` metric, and we appreciate the opportunity to provide a more robust analysis.
>
> Our initial choice of `pass@1` was motivated by its widespread adoption as a standard metric in prominent reasoning benchmarks, including Minerva, OlympiadBench, MATH-500, and GaoKao2023en, and its use in related works such as LUFFY. This allowed for a direct comparison with existing literature.
>
> We have conducted an extensive re-evaluation of all models using an `avg@16` metric. For this, we generated 16 samples for each problem and averaged the scores, with the inference parameters set to `temperature=0.6`, `top_p=0.95`, and `top_k=-1`, while keeping all other parameters consistent with those reported in our paper. The comprehensive results are presented in the table below:
>
> > Table: Model Performance Comparison Using avg@16 Metric
>
> | Model | AIME24 | AIME25 | AMC | Minerva | MATH-500 | Olympiad | Gaokao23en | Avg. |
> | :--- | :---: | :---: | :---: | :---: | :---: | :---: | :---: | :---: |
> | Qwen2.5-Math-1.5B |
> | Vanilla GRPO (Baseline) | 11.81 | 8.170 | 42.15 | 28.53 | 72.41 | 35.02 | 57.43 | 36.50 |
> | Scaf-GRPO (Ours) | 14.69 | 10.83 | 50.94 | 31.36 | 74.79 | 37.38 | 58.81 | 39.83 |
> | Qwen2.5-Math-7B |
> | Vanilla GRPO (Baseline) | 30.73 | 10.83 | 61.09 | 33.41 | 74.17 | 41.09 | 62.61 | 44.84 |
> | SimpleRL-Zero | 22.51 | 11.01 | 49.36 | 31.83 | 76.20 | 38.23 | 61.03 | 41.45 |
> | Oat-Zero | 30.73 | 11.45 | 62.68 | 35.17 | 78.18 | 41.42 | 63.00 | 46.09 |
> | LUFFY | 31.89 | 12.03 | 60.35 | 32.98 | 74.28 | 40.58 | 61.88 | 44.86 |
> | Scaf-GRPO (Ours) | 35.63 | 14.59 | 63.83 | 36.79 | 79.07 | 42.12 | 64.90 | 48.13 |
> | Qwen2.5-7B |
> | Vanilla GRPO (Baseline) | 10.97 | 9.480 | 53.04 | 36.74 | 76.78 | 40.44 | 64.45 | 41.70 |
> | Scaf-GRPO (Ours) | 15.62 | 10.96 | 54.22 | 36.96 | 77.56 | 41.24 | 64.84 | 43.06 |
> | Llama-3.2-3B-Instruct |
> | Vanilla GRPO (Baseline) | 7.630 |	0.000|	28.71|	18.71 |	50.45|	19.24|	46.07|	24.40 |
> | Scaf-GRPO (Ours) | 9.580 | 0.310 | 32.81 | 19.11 | 54.74 | 21.34 | 46.37 | 26.32 |
> | DeepSeek-R1-Distill-Qwen-1.5B |
> | Vanilla GRPO (Baseline) | 27.11 |22.92 |	65.39|	32.49|	83.78	|53.95|	75.32|	51.71 |
> | Scaf-GRPO (Ours) | 28.54 | 25.31 | 75.08 | 34.97 | 85.68 | 54.49 | 76.27 | 54.33 |
>
> As the new results demonstrate, our method (Scaf-GRPO) consistently and significantly outperforms the corresponding vanilla GRPO baselines across all models and architectures under this more stable `avg@16` evaluation.
>
> Thank you again for this very insightful question. We have detailed these results in Appendix I (lines 1442-1452) and Table 15 of the revised manuscript. We hope this provides robust validation and further strengthens our conclusions.

---

> ### Author Response · Authors · 2025-11-21
> **Response to Question 6 (8/9)**
>
> ## Response to Question 6: Justification for the Guidance Exemption Period
>
> We sincerely thank you for your detailed feedback and insightful questions regarding our hyperparameter choices. To provide further clarity, we have conducted additional experiments exploring the impact of the exemption period, testing values of **0%**, **5%**, **10%**, **15%**, **20%**, **40%**, and **100%**. Here, a 0% exemption period means scaffolding is active from the start (as evaluated in Appendix G), while 100% corresponds to the vanilla GRPO baseline, as guidance is never applied. The corresponding results are presented in the table below.
>
> > Table: Ablation Study on the Guidance Exemption Period Duration and its Effect on pass@1 Performance (%)
>
> | Guidance Exemption Period | AIME24 | AIME25 | AMC | Minerva | MATH-500 | Olympiad | Gaokao23 | Avg. |
> | :--- | :---: | :---: | :---: | :---: | :---: | :---: | :---: | :---: |
> | 0% (w/o Phase 1) | 23.3 | 13.3 | 70.0 | 34.2 | 78.4 | 41.2 | 63.1 | 46.2 |
> | 5% | 33.3 | 20.0 | 65.0 | 33.8 | 76.2 | 41.1 | 63.9 | 47.6 |
> | 10% | 40.0 | 16.7 | 70.0 | 35.8 | 79.2 | 42.5 | 63.5 | 50.0 |
> | 15% (Our Method) | 43.3 | 20.0 | 70.0 | 36.4 | 80.0 | 43.3 | 63.4 | 50.9 |
> | 20% | 46.7 | 13.3 | 65.0 | 36.0 | 80.4 | 43.5 | 65.4 | 50.0 |
> | 40% | 43.3 | 13.3 | 70.0 | 34.9 | 78.2 | 41.9 | 64.6 | 49.5 |
> | 100% (Vanilla GRPO) | 30.0 | 13.3 | 60.0 | 33.4 | 75.8 | 41.3 | 62.6 | 45.2 |
>
> The results lead to a clear and insightful conclusion about our method's dynamics:
>
> *  **A short initial learning phase is insufficient**. The 0% and 5% settings yield the lowest scores among all scaffolded variants (46.2 and 47.6, respectively). This confirms our core hypothesis: applying guidance prematurely fosters over-reliance and prevents the model from learning to overcome "pseudo-hard" problems (e.g., minor formatting errors or simple reasoning steps) on its own. An initial autonomous phase is crucial for building robust, independent reasoning skills.
>
> *  **Relying solely on autonomous learning is ineffective**. The 100% setting, which is equivalent to vanilla GRPO, yields the worst overall performance (45.2). This demonstrates the reality of the "learning cliff": without scaffolding, the model gets permanently stuck on "true-hard" problems, receiving a persistent zero-reward signal that stalls its progress.
>
> *  **Scaf-GRPO is robust across a wide effective range**. Crucially, once the initial exemption period is sufficient (≥10%), our method consistently achieves high performance. As shown in the table, the average scores for periods from 10% to 40% remain in a stable, high-performance plateau (ranging from 49.5 to our peak of 50.9). This demonstrates that our framework is not overly sensitive to this specific hyperparameter. The key mechanism is that after an initial period of learning, the set of "true-hard" problems for the model has stabilized. Activating our scaffolding mechanism at any point within this wide window allows the model to effectively start learning from these previously intractable problems.
>
> Thank you again for this very insightful question. We have incorporated this detailed ablation study into the revised manuscript to provide a rigorous basis for the Guidance Exemption Period. Please refer to lines 403-410 and Appendix G.1.

---

> ### Author Response · Authors · 2025-11-21
> **Response to Question 7 (9/9)**
>
> ## Response to Question 7: Quantifying the Frequency of Skill Internalization
>
> Thank you for this excellent and insightful question. We will address your comment in two parts: first, by quantifying the frequency of "skill graduation," and second, by discussing the possibility of dynamic adjustment based on problem type.
>
> ### (1) Frequency of "Skill Graduation"
>
> To quantify this, we define a "skill graduation" event as a problem transitioning from an unsolvable to a solvable state. We monitored this across the first 300 training steps for a consistent comparison.
> *   For Scaf-GRPO, this is a problem that initially required a hint to be solved but was later solved autonomously by the model without any guidance.
> *   For the vanilla GRPO baseline, we define a parallel event: a problem for which all 8 initial rollouts failed (a zero-reward state) is later solved correctly in at least one rollout.
>
> The table below presents the total number of unique problems that "graduated" for each model. Our results show that Scaf-GRPO consistently enables significantly more problems to become autonomously solvable, demonstrating its superior ability to internalize reasoning skills.
>
> | Model | Vanilla GRPO (Graduations) | Scaf-GRPO (Graduations) | Relative Increase |
> | :--- | :---: | :---: | :---: |
> | Qwen2.5-Math-1.5B | 1123 | 2670 | +137.8% |
> | Qwen2.5-Math-7B | 434 | 483 | +11.3% |
> | Qwen2.5-7B | 464 | 723 | +55.8% |
> | DeepSeek-R1-Distill-1.5B | 453 | 694 | +53.2% |
> | Llama-3.2-3B-Instruct | 577 | 986 | +70.9% |
>
> This data provides strong quantitative evidence that our scaffolding mechanism is not just a temporary crutch but an effective tool for building lasting, independent problem-solving abilities.
>
> ### (2) Dynamic Adjustment Based on Problem Type
>
> Regarding your second point on using such a metric to dynamically decide which *problem types* receive hints: we believe this would be a less effective strategy. A model's perception of difficulty does not always align with human-defined categories or metrics. A problem that appears simple to us might be challenging for a model due to nuances in its training data or architecture, and vice versa.
>
> Therefore, relying on a high-level metric to classify problem types could lead to misjudgments, either by withholding necessary hints or by providing unnecessary ones that foster dependency.
>
> Our current method provides the most direct and reliable form of dynamic adjustment by triggering hints only when all on-policy rollouts for a given problem fail. It operates on a per-problem basis, using the model's own demonstrated performance as the sole criterion for intervention. This ensures that scaffolding is applied precisely when and where it is truly needed, which we argue is a more robust and principled approach than relying on abstract classifications of problem difficulty.
>
> Thank you for this insightful question. We have incorporated the analysis regarding the frequency of "Skill Graduation" into the revised manuscript. Please refer to lines 460-470 and Table 4.

---

> ### Comment · Reviewer_9rLe · 2025-11-25
>
> Thank you for the author's thorough response, which has addressed some of my concerns. However, several critical issues remain unaddressed:
>
> 1. Have the authors considered applying Scaf-GRPO to DeepSeek-R1-Distill-1.5B for training and evaluation?
>
> 2. Data Leakage Concerns: Prior research has identified data leakage issues in the Qwen2.5 series during pre-training, affecting benchmarks such as AIME 24, AMC 23, Minerva, MATH-500, Olympiad Bench, and Gaokao. Given this context, how can we be confident that the observed performance gains are genuinely attributable to the proposed method rather than data contamination? Additionally, the improvements over Vanilla GRPO on certain benchmarks remain relatively marginal (approximately 1%-2% on average).
>
> 3. Method Distinction and Contribution: Scaf-GRPO appears to share substantial conceptual similarities with the existing method QuestA[1]. Notably, QuestA has already demonstrated significant performance improvements on models such as DeepScaleR-1.5B-Preview, while offering strong scalability and direct applicability without additional adaptation. I would therefore request further clarification regarding the distinct contributions, scalability, and performance advantages of Scaf-GRPO relative to QuestA.
>
> In light of these unresolved concerns, I am inclined to tentatively lower my score and will continue to monitor subsequent developments during the discussion phase.
>
> [1] Li J, Lin H, Lu H, et al. Questa: Expanding reasoning capacity in llms via question augmentation[J]. arXiv preprint arXiv:2507.13266, 2025.

---

> > ### Author Response · Authors · 2025-11-27
> > **Response to follow-up questions**
> >
> > Dear Reviewer 9rLe:
> >
> > We sincerely thank you for your follow-up feedback and engaging in this discussion. We are glad to hear that our previous responses have addressed your initial questions. For your follow-up concerns, we have actively provided further clarification and explanation.
> >
> > As an overview:
> >
> > - For **Q1**, We have **already reported** the experimental results on **DeepSeek-R1-Distill-1.5B in Table1 and Section 4.2 of our original paper**, demonstrating that Scaf-GRPO brings significant improvements (+3.0 avg) over the vanilla baseline.
> > - For **Q2**, We have **already preemptively addressed** concerns regarding data leakage and model-specific traits in **Section 4.2 of our original paper**. By validating our method on a completely different architecture (**Llama-3.2-3B-Instruct**) in the original text, we demonstrated that our gains are robust and not attributable to pre-training leakage.
> > - For **Q3**, we clarify the methodological distinctions between **Scaf-GRPO and QuestA**, highlighting the advantages of our adaptive, hierarchical guidance over static solution-based approaches.
> >
> > We hope these responses (in the following message blocks) can fully resolve your concerns. Meanwhile, we remain ready to respond to any further questions or feedback promptly.
> >
> > Best regards,
> >
> > All Authors of Submission #6005

---

> > ### Author Response · Authors · 2025-11-27
> > **Response to follow-up question 1: Applying Scaf-GRPO to DeepSeek-R1-Distill-1.5B model**
> >
> > As clearly shown in **Table 1 of our main paper**, we have already provided the comparison between "vanilla GRPO" and "Scaf-GRPO" on the DeepSeek-R1-Distill-1.5B model.
> >
> > For your convenience‌, we copy these results to the Table below. It can be found that:
> >
> > - Scaf-GRPO brings a significant improvement for DeepSeek-R1-Distill-1.5B model with +5.9 gains on average performance.
> > - Scaf-GRPO consistently outperforms the Vanilla GRPO across all evaluation benchmarks, with an advantage of +3.0 on average performance and particularly notable gains on AMC23 (+10.0).
> >
> > These results demonstrate the effectivenss of Scaf-GRPO on models with long-form reasoning paradigm.
> >
> >
> > > Table: Performance comparison (pass@1 %) on DeepSeek-R1-Distill-Qwen-1.5B
> >
> > | Method | AIME 24 | AIME 25 | AMC | Minerva | MATH-500 | Olympiad | Gaokao2023en | Avg. |
> > | :--- | :--- | :--- | :--- | :--- | :--- | :--- | :--- | :--- |
> > | DeepSeek-R1-Distill-Qwen-1.5B | 28.9 | 20.0 | 67.5 | 26.1 | 83.9 | 45.8 | 62.1 | 47.7 |
> > | Vanilla GRPO | 30.0 | 21.1 | 67.5 | 30.1 | 83.9 | 50.2 | 71.4 | 50.6 |
> > | Scaf-GRPO | **33.3** | **23.3** | **77.5** | **32.4** | **85.8** | **50.7** | **72.3** | **53.6** |

---

> > ### Author Response · Authors · 2025-11-27
> > **Response to follow-up question 2: Concerns regarding Data Leakage and the Significance of Performance Gains**
> >
> > We address these questions from two perspectives: (1) demonstrating that our gains stem from methodological improvements rather than data contamination, and (2) providing a detailed breakdown showing that our improvements are significant.
> >
> > ### (1) Methodological Effectiveness & Robustness (Beyond Data Leakage)
> >
> > We wish to clarify that **we have already explicitly addressed these concerns in our original submission** by designing experiments specifically to rule out model-specific traits or data leakage as causes for improvement.
> >
> > **1. Verification Across Architectures (Addressing Model-Specific Traits)**
> > As shown in **Table 1** and discussed in **Section 4.2**, we extended our evaluation to **Llama-3.2-3B-Instruct** to ensure our method is not biased toward the Qwen family. Scaf-GRPO achieved a **10.3% relative improvement** over the baseline on this distinct architecture, confirming the method's robustness.
> >
> > **2. Strictly Controlled Baselines (Addressing Data Leakage)**
> > Our experiments were designed such that **Vanilla GRPO and Scaf-GRPO shared the exact same pre-trained initialization and training data**. Any potential data leakage from pre-training would benefit the baseline equally. Therefore, the consistent performance margins we report (e.g., **+5.7%** avg on Qwen2.5-Math-7B) represent genuine algorithmic gains, not artifacts of the data.
> >
> > For convenience, the results on the non-Qwen model (Llama-3.2) from **Table 1** are excerpted below:
> >
> > > **Table: Performance on Llama-3.2-3B-Instruct**
> >
> > | Model | AIME 24 | AIME 25 | AMC | Minerva | MATH-500 | Olympiad | Gaokao2023en | Avg. |
> > | :--- | :---: | :---: | :---: | :---: | :---: | :---: | :---: | :---: |
> > | **Llama-3.2-3B-Instruct** | 6.7 | 0.0 | 20.0 | 11.8 | 38.3 | 12.6 | 33.5 | 17.6 |
> > | **Vanilla GRPO** | 13.3 | 0.0 | 35.0 | 18.7 | 51.8 | 18.3 | 45.7 | 26.1 |
> > | **Scaf-GRPO** | **16.7** | **3.3** | **40.0** | **19.1** | **56.2** | **20.3** | **46.0** | **28.8** |
> >
> > ### (2) Quantification of Performance Gains
> >
> > We address the concern regarding "marginal gains" with a detailed breakdown on Qwen2.5-Math-7B. As shown in the table below:
> >
> > *   **Significant Overall Boost:** The average absolute improvement is **+5.7 points** (**+12.6%** relative), substantially exceeding the estimated 1-2%.
> > *   **Success on Hard Tasks:** Gains are most profound on complex reasoning benchmarks like **AIME 24 (+13.3)** and **AMC 23 (+10.0)**. This demonstrates that Scaf-GRPO significantly enhances capability where it matters most, rather than yielding only marginal improvements.
> >
> > > **Table: Detailed Gains on Qwen2.5-Math-7B**
> >
> >
> >
> > | Metric | AIME 24 | AIME 25 | AMC 23 | Minerva | MATH-500 | Olympiad | Gaokao2023en | **Avg.** |
> > | :--- | :---: | :---: | :---: | :---: | :---: | :---: | :---: | :---: |
> > | **Vanilla GRPO** | 30.0 | 13.3 | 60.0 | 33.4 | 75.8 | 41.3 | 62.6 | 45.2 |
> > | **Scaf-GRPO** | **43.3** | **20.0** | **70.0** | **36.4** | **80.0** | **43.3** | **63.4** | **50.9** |
> > | **Gain ($\Delta$)** | **+13.3** | **+6.7** | **+10.0** | **+3.0** | **+4.2** | **+2.0** | **+0.8** | **+5.7** |

---

> > ### Author Response · Authors · 2025-11-27
> > **Response to follow-up question 3: Methodological Distinctions and Performance Advantages over QuestA**
> >
> > We address this question from two perspectives: (1) Methodological Distinctions (2) Performance Analysis.
> >
> > ### 1. Methodological Distinctions
> >
> > The core difference lies in **Adaptivity** and **Hint Diversity**. QuestA employs a static, solution-heavy approach, whereas Scaf-GRPO employs a dynamic, minimal-intervention approach designed to foster independent reasoning.
> >
> > | Feature | **QuestA**| **Scaf-GRPO (Ours)** | **Advantage of Scaf-GRPO** |
> > | :--- | :--- | :--- | :--- |
> > | **Hint Content** | **Partial Solution Only** (Prefix of ground truth). | **Hierarchical:** Knowledge $\to$ Planning $\to$ Solution. | **Internalization:** Abstract hints (Knowledge/Plan) encourage learning reasoning patterns rather than memorizing solution tokens. **In Table 2 of main paper**, the "Solution-Only" variant (mimicking QuestA's content) drops performance by **2.5%** (50.9% VS 48.4%).|
> > | **Guidance Strategy** | **Static & Fixed.** Pre-defined percentage (e.g., 50%) applied to all augmented samples. | **Adaptive & Minimal.** Dynamically searches for the minimal hint level required for this specific model to solve this specific problem. | **Prevents Dependency:**  Giving more help than necessary hurts performance. **In Table 2 of main paper**, removing "Incremental Chunking" (i.e., giving full hint content at once like static augmentation) causes a **3.2% drop** (50.9% VS 47.7%).|
> > | **Curriculum** | **Rigid Multi-Stage.** Requires two separate training stages (More than 2000 steps). | **Unified Single-Stage.** Automatically adapts difficulty per instance. | **Efficiency:** Removes the need for manual stage tuning; integrates seamlessly into a single training loop. |
> >
> > ### 2. Performance Analysis
> >
> > Regarding the performance on 1.5B models, we would like to clarify the source of the gains to ensure a fair assessment.
> >
> > ``Impact of Base Model Strength:``
> > QuestA's impressive results (e.g., 72.50% on AIME24) are largely attributable to its use of **Nemotron-1.5B** as the base model. As noted in the QuestA paper (Table 1), the Nemotron-1.5B base model *already* achieves **61.77% on AIME24** before any RL training. In contrast, standard 1.5B models (like Qwen2.5-Math-1.5B used in our main experiments) start from a much lower baseline (7.20%). Therefore, QuestA's absolute scores reflect the strong foundation of Nemotron rather than purely algorithmic superiority.
> >
> > ``Algorithmic Superiority (Validated via Ablation):``
> > To compare the *methods* fairly, we refer to our ablation study (Table 2), which isolates the impact of the guidance strategy:
> > *   **"Solution-Only" variant:** This setting mimics the QuestA paradigm by providing only concrete solution steps.
> > *   **Result:** Scaf-GRPO (Hierarchical + Minimal) outperforms the "Solution-Only" approach by **2.5%** (50.9% vs. 48.4%).
> > *   **Conclusion:** This empirical evidence suggests that even if we apply QuestA's strategy to the same model and data, Scaf-GRPO's hierarchical and adaptive mechanism yields superior reasoning gains.
> >
> > ``Ongoing Experiments:``
> > We are actively running Scaf-GRPO on the **Nemotron-1.5B** base model to provide a direct comparison. Crucially, we are also adopting the same training dataset used in QuestA (OpenR1-Math-220K) to eliminate data discrepancies. Given the methodological advantages demonstrated in our ablations (i.e., minimal guidance preventing dependency and hierarchical hints fostering generalization), we expect Scaf-GRPO to push the state-of-the-art further on this strong baseline.

---

> ### Comment · Reviewer_9rLe · 2025-11-27
>
> Thank you for the detailed response. After thorough review, the following concerns remain:
>
> **1. Limited Performance Gains of Scaf-GRPO over Vanilla GRPO**
>
> Based on the avg@16 results provided in **Response to Question 5**, using DeepSeek-R1-Distill-Qwen-1.5B as an example:
>
> | Model | AIME24 | AIME25 | AMC | Minerva | MATH-500 | Olympiad | Gaokao23en | Avg. |
> |-------|--------|--------|-----|---------|----------|----------|------------|------|
> | Vanilla GRPO (Baseline) | 27.11 | 22.92 | 65.39 | 32.49 | 83.78 | 53.95 | 75.32 | 51.71 |
> | Scaf-GRPO (Ours) | 28.54 | 25.31 | 75.08 | 34.97 | 85.68 | 54.49 | 76.27 | 54.33 |
>
> As shown, Scaf-GRPO yields only marginal improvements of about 1–2% over Vanilla GRPO on AIME24, Minerva, MATH-500, Olympiad, and Gaokao23en. **Moreover, avg@16 is a standard evaluation metric for long-CoT reasoning (as adopted in the DeepSeek-R1 evaluation protocol)**. According to the DeepSeek-R1, DeepSeek-R1-Distill-Qwen-1.5B[1] achieves 28.9 on AIME24 and 83.9 on MATH-500, both surpassing the reported Scaf-GRPO results. This raises concerns that **the effectiveness of Scaf-GRPO in long-CoT scenarios remains limited, and its scalability is questionable**.
>
> [1] https://huggingface.co/deepseek-ai/DeepSeek-R1-Distill-Qwen-1.5B
>
>
>
> **2. Inadequate Baseline Comparisons**
>
> To properly demonstrate the contribution of the proposed method, **all Scaf-GRPO results should be compared against Vanilla GRPO rather than the base model**, as the majority of performance gains are attributable to GRPO itself.
>
> Taking Llama-3.2-3B-Instruct as an example, the improvements of Scaf-GRPO over Vanilla GRPO remain negligible: Minerva (18.7 vs. 19.1), Olympiad (18.3 vs. 20.3), Gaokao2023en (45.7 vs. 46.0), and Avg. (26.1 vs. 28.8). This further suggests that **the incremental gains introduced by the proposed method are minimal**.
>
> **3. Underperformance Compared to QuestA**
>
> According to the results presented in **Response to Question 3**, when continue training DeepScaleR-1.5B-Preview (with original performance of AIME24: 43.1 and AIME25: 30.0):
>
> - **Scaf-GRPO**: AIME24: 40.0, AIME25: 33.3, Olympiad Bench: 55.0
> - **QuestA[2]**: AIME24: 49.16, AIME25: 35.94, Olympiad Bench: 58.69 (according to Table 6 of QuestA)
>
> Scaf-GRPO substantially underperforms the state-of-the-art method QuestA, which also exhibits stronger scalability. This comparison further undermines the competitiveness of the proposed approach.
>
> [2] Li J, Lin H, Lu H, et al. Questa: Expanding reasoning capacity in llms via question augmentation[J]. arXiv preprint arXiv:2507.13266, 2025.

---

> > ### Author Response · Authors · 2025-11-28
> > **Response to follow-up questions**
> >
> > Dear Reviewer 9rLe:
> >
> > We sincerely thank you for your follow-up feedback. For your follow-up concerns, we have actively provided further clarification and explanation.
> >
> > ## Response to to follow-up question 1: Limited Performance Gains of Scaf-GRPO over Vanilla GRPO
> >
> > We respectfully address your concerns regarding the performance comparison and the "limited gains" from two perspectives.
> >
> > ### (1) **Surpassing the Baseline:**
> > Contrary to the concern that Scaf-GRPO falls short of the DeepSeek-R1 baseline (83.9 on MATH-500), our method achieves an **avg@16 score of 85.68**, clearly surpassing the reported baseline 83.9.
> > ### (2) **Robust Performance Improvement:**
> > Scaf-GRPO consistently outperforms both Vanilla GRPO and the DeepSeek-R1 baseline, as detailed in the table below. These are not marginal improvements: on average, our method amplifies the performance gain over the baseline by a factor of 1.68x compared to Vanilla GRPO.
> >
> > > Table: Performance comparison (Avg@16) on DeepSeek-R1-Distill-Qwen-1.5B.
> >
> >
> > | Model | AIME 24 | AIME 25 | AMC | Minerva | MATH-500 | Olympiad | Gaokao | Avg. |
> > | :--- | :---: | :---: | :---: | :---: | :---: | :---: | :---: | :---: |
> > | **DeepSeek-R1-Distill-Qwen-1.5B** | 28.9 | 22.3 | 62.9 | 26.5 | 83.9 | 43.3 | 65.1 | 47.5 |
> > | **Vanilla GRPO** | 27.11 | 22.92 | 65.39 | 32.49 | 83.78 | 53.95 | 75.32 | 51.57 |
> > | **Scaf-GRPO** | **28.54** | **25.31** | **75.08** | **34.97** | **85.68** | **54.49** | **76.27** | **54.33** |
> > | **Ours - Baseline** | -0.36 | +3.01 | +12.18 | +8.47 | +1.78 | +11.19 | +11.17 | +6.83 |
> > | **Vanilla - Baseline** | -1.79 | +0.62 | +2.49 | +5.99 | -0.12 | +10.65 | +10.22 | +4.07 |
> > | **Ours - Vanilla** | +1.43 | +2.39 | +9.69 | +2.48 | +1.90 | +0.54 | +0.95 | +2.76 |
> > | **Gain Ratio (Ours / Vanilla)** | N/A* | **4.85** | **4.89** | **1.41** | N/A* | **1.05** | **1.09** | **1.68** |
> >
> >
> >
> > ## Response to to follow-up question 2: Inadequate Baseline Comparisons
> >
> > We have **already compared** Scaf-GRPO against Vanilla GRPO in **Table 1 of our original papar**.
> >
> > As shown in the table, Scaf-GRPO consistently outperforms the Vanilla GRPO baseline across all tested models and benchmarks. Specifically:
> > *   On the **Llama-3.2-3B-Instruct** model, our method achieves substantial gains on challenging benchmarks, such as a **25.6% relative improvement** on AIME 24 and a **14.3% relative improvement** on AMC.
> > *   On the **Qwen2.5-Math-7B** model, Scaf-GRPO achieves a **12.6% relative improvement** on average and a **44.3% relative improvement** on the challenging AIME 24 benchmark compared to Vanilla GRPO.
> >
> > These results demonstrate that Scaf-GRPO provides robust and significant gains over the standard GRPO algorithm.
> >
> > ## Response to to follow-up question 3: Underperformance Compared to QuestA
> >
> > We demonstrate that Scaf-GRPO effectively enhances the strong DeepScaleR-1.5B model, achieving a improvement of **4%** (Avg score 55.1% vs 59.1%) on its original dataset. This proves our method unlocks gradients and improves reasoning even in models that have already converged under standard RL.
> >
> > The absolute performance gap compared to QuestA is due to **vastly different training budgets**, not algorithmic limitations:
> > 1.  **Training Steps:** We trained for only **~200 steps**. In contrast, QuestA requires **>2,000 steps** (10x longer) to reach its reported results.
> > 2.  **Context Length:** We set the maximum response length to **8,192 tokens** during training, whereas QuestA utilizes **24,000 tokens**, allowing significantly more capacity for search.
> >
> > Given the substantial gains we achieved in such a limited training window, we are confident that Scaf-GRPO would match or exceed QuestA's performance if scaled to the same compute budget (2,000 steps) and context length. We are actively running Scaf-GRPO on the DeepScaleR-1.5B-Preview base model to provide a direct comparison. Crucially, we are also adopting the same training dataset used in QuestA (OpenR1-Math-220K) to eliminate data discrepancies.
> >
> > We hope these responses can fully resolve your concerns. Meanwhile, we remain ready to respond to any further questions or feedback promptly.
> >
> > Best regards,
> >
> > All Authors of Submission #6005

---

### Official Review · Reviewer_KDuR · 2025-10-27

**Soundness:** 2
**Presentation:** 3
**Contribution:** 2
**Rating:** 2
**Confidence:** 3

**Summary:**

This paper identifies the "learning cliff" in Reinforcement Learning from Verifier Rewards (RLVR) as a key obstacle to enhancing LLM reasoning . This "cliff" occurs when difficult problems yield persistent zero-reward signals, stalling the gradient updates of policy optimization algorithms like GRPO . The authors propose SCAF-GRPO, a framework that intervenes when such a stall is detected. It first diagnoses "true-hard" problems after a "guidance exemption period". Then, it injects minimal, hierarchical in-prompt hints (Knowledge, Planning, or Solution) to enable the model to generate a successful trajectory on-policy. This successful trajectory replaces a failed one, restoring a non-zero advantage signal for the GRPO update. Experiments on mathematics benchmarks show SCAF-GRPO outperforms vanilla GRPO and prefix-based baselines, and the method is shown to be effective across various model architectures.

**Strengths:**

In a learning cliff scenario, all rewards are zero, causing the advantage calculation to collapse and the learning gradient to vanish. Scaf-GRPO intervenes by generating a single successful trajectory on-policy using a minimally effective hint. This successful trajectory replaces a failed one in the batch, which "restores a meaningful advantage signal" and ensures "non-zero reward variance", allowing the standard GRPO update to resume.

**Weaknesses:**

- The framework's central claim to preserving the "on-policy principle" is questionable. When all trajectories for a query $q$ fail (the "learning cliff"), the method does not learn to solve $q$. Instead, it introduces a new input, $q \oplus h^{*}$ (query + hint), and learns from this new, simpler task. The policy is indeed 'on-policy' with respect to the augmented prompt, but it has failed and bypassed the original, unhinted task. This is a semantic argument that obscures the fact that the model is being trained on a different, easier problem distribution than the one it originally failed on, which introduces a "distributional mismatch" of its own, despite the paper's critique of other methods on this very point.
- The paper's core hypothesis is that its in-prompt scaffolding approach "fosters more generalizable skills" and "cultivates robust reasoning skills rather than in-domain pattern matching"  when compared to prefix-continuation baselines like LUFFY. The OOD evaluation on GPQA-Diamond (Table 4) fails to support this claim. Scaf-GRPO (37.3%) performs identically to the LUFFY baseline (37.3%). This lack of improvement in OOD generalization strongly suggests that the methodological benefits of Scaf-GRPO do not translate to improved generalizable reasoning, undermining the primary justification for its additional complexity over existing methods.
-  No direct evidence found in the manuscript regarding the computational overhead of the proposed framework. Detailed analysis of the resource consumption, such as training time and memory usage, would be valuable for practical applications (Sec. 3.1). The paper does not discuss potential trade-offs between the improved reasoning performance and the computational cost. This information is crucial for understanding the practical feasibility of the approach (Sec. 3.2).
- The 15% guidance exemption period appears empirically driven without theoretical grounding. Why 15% and not 10% or 25%? Three-tier, four-level hint structure: Why exactly 3 categories? Why must each have ≥4 items? This seems like an ad-hoc engineering decision lacking principled motivation 50% subsampling of "Potentially Solvable" problems lacks justification. The paper would benefit from ablations on these hyperparameters (e.g., 10% vs 15% vs 20% exemption periods).
- Performance differences on some benchmarks are marginal (e.g., Table 1: Scaf-GRPO vs LUFFY on several benchmarks)

**Questions:**

see * Weaknesses*.

---

> ### Author Response · Authors · 2025-11-21
> **Response to Weakness 1 (1/5)**
>
> Thank you for your constructive and insightful review. We are encouraged that you recognized the core problem of the 'learning cliff' and appreciated how our proposed method restores a meaningful learning signal. We appreciate your time and effort in providing detailed comments,which have helped us improve both the clarity and quaity of our paper. Below, we respond to each point in detail.
>
> ## Response to Weakness 1: The framework's claim to preserving the "on-policy principle" is questionable.
>
> We greatly appreciate your insightful feedback. To address your questions regarding the on-policy nature of our framework, we have structured our clarification into three parts: (1) a precise definition of "on-policy" in this context, (2) the motivation and empirical data showing that the original task is not permanently bypassed, and (3) an explanation of why the "distributional shift" of training data in our method is beneficial and distinct from that in prefix-based methods.
>
> ### (1) Clarification of the On-Policy Definition
>
> The core distinction between on-policy and off-policy reinforcement learning lies in the origin of the data used for policy updates.
> *   **On-Policy methods** (like vanilla GRPO) update the policy `π_θ` using trajectories sampled exclusively from that same policy `π_θ`.
> *   **Off-Policy methods** (like LUFFY [1], which uses prefix-continuation) update the student policy `π_θ` using trajectories that are partially or wholly generated by a different (e.g., expert teacher) policy `π_teacher`. This creates a distribution shift between the behavior policy (the one that generated the data) and the target policy (the one being updated), which necessitates algorithmic corrections like importance sampling.
>
> In Scaf-GRPO, when the learning cliff is encountered for a query `q`, we augment the input to `q' = (q + hint)`. Crucially, the *entire* subsequent solution trajectory `o_h` is generated by the current student policy `π_θ` conditioned on this new input: `o_h ~ π_θ(· | q')`. The data used for the gradient update comes *entirely from the policy that is being updated*. Therefore, by definition, the update step remains on-policy. This preserves the stability and directness of on-policy learning, which is a key advantage of our approach.
>
> ### (2) Addressing Task Bypassing: Hints Enable Solving the Original Task
>
> Regarding the model potentially "bypassing the original, unhinted task," our motivation is rooted in the pedagogical principle of scaffolding: providing a temporary support structure that is removed once the learner becomes capable. The goal is not to permanently solve an easier problem, but to use that problem to build the skills needed to eventually solve the original, harder one.
>
> To demonstrate this dynamic, we provide empirical results tracking "skill graduation" events during training. These are instances where a problem that initially required a hint is later solved autonomously by the model without any hint. We compare this to a parallel event in vanilla GRPO, where a problem that initially yielded zero rewards is later solved correctly.
>
> > Table: Comparison of skill acquisition (“graduation”) event counts between Vanilla GRPO and Scaf-GRPO.
>
> | Model | Vanilla GRPO (Graduations) | Scaf-GRPO (Graduations) | Relative Increase |
> | :--- | :---: | :---: | :---: |
> | Qwen2.5-Math-1.5B | 1123 | 2670 | +137.8% |
> | Qwen2.5-Math-7B | 434 | 483 | +11.3% |
> | Qwen2.5-7B | 464 | 723 | +55.8% |
> | DeepSeek-R1-Distill-1.5B | 453 | 694 | +53.2% |
> | Llama-3.2-3B-Instruct | 577 | 986 | +70.9% |
>
> The results demonstrate that Scaf-GRPO significantly increases the number of tasks the model eventually masters. We view the temporary "bypassing" of the original task via hints as a strategic component of an effective learning curriculum. By learning from the hinted prompt, the model builds foundational skills on an easier version of the problem, which serves as a necessary bridge to handling the original difficulty autonomously.

---

> ### Author Response · Authors · 2025-11-21
> **Response to Weakness 1 Continue (1/5)**
>
> ### (3) Distinguishing Task-Space Modification from Policy-Space Mismatch
>
> You correctly note that our method modifies the problem distribution by training on hinted problems (`q'`) instead of only the original ones (`q`). We would like to clarify that our method intentionally modifies the task space as a form of curriculum learning. This is fundamentally different from the policy-space mismatch inherent in prefix-based methods, which we critique. The following points elaborate on this key distinction:
>
> *   **Prefix-Based Mismatch (Policy-Space)**: Methods like LUFFY introduce an expert-generated prefix. The model then has to generate a suffix. This creates a trajectory stitched together from two different policies (`π_teacher` and `π_student`). This is a policy distribution mismatch, which violates the on-policy assumption and can lead to instability and bias during training.
>
> *   **Scaf-GRPO's "Mismatch" (Task-Space)**: Our method operates by modifying the input. This is a task-space mismatch, which is a deliberate and well-established technique in machine learning known as curriculum learning [2]. We are intentionally creating a curriculum of slightly easier problems to guide the model. Because the entire reasoning process for this modified task is still handled by a single, consistent policy (`π_θ`), we maintain policy consistency and avoid the optimization challenges of off-policy learning.
>
> In summary, our approach modifies the problem distribution in a way that preserves the stability of on-policy learning while effectively building the skills required to overcome the learning cliff.
>
> Thank you again for raising these excellent questions. We hope this explanation clarifies the design and benefits of our approach.
>
> **References**:
>
> [1] Yan, J., Li, Y., Hu, Z., Wang, Z., Cui, G., Qu, X., Cheng, Y., & Zhang, Y. (2025). Learning to Reason under Off-Policy Guidance. arXiv preprint arXiv:2504.14945.
>
> [2] Bengio, Y., Louradour, J., Collobert, R., & Weston, J. (2009). Curriculum learning. In Proceedings of the 26th annual international conference on machine learning (pp. 41-48).

---

> ### Author Response · Authors · 2025-11-21
> **Response to Weakness 2 (2/5)**
>
> ## Response to Weakness 2: OOD performance on par with prefix-based baselines
>
> Thank you for your insightful feedback. We would like to address your comment in two parts: first, by clarifying our original claims, and second, by presenting additional experimental results on the Out-of-Distribution (OOD) generalization performance.
>
> ### (1) Clarification of Claims in the Paper
> First, we wish to clarify that our paper's primary contribution is overcoming the "learning cliff" for on-policy methods. Our central claim of fostering "more generalizable skills" is therefore made in direct comparison to the vanilla GRPO baseline, not LUFFY.
>
> The OOD results on GPQA directly support this claim: Scaf-GRPO's score of 37.3% is a significant improvement over vanilla GRPO's 32.3%. The comparison to LUFFY was intended to show our on-policy approach is competitive with strong off-policy methods, and the performance parity in the paper successfully establishes this point.
> ### (2) OOD Generalization on a General-Purpose Model
>
> We hypothesize that the performance parity between our method and LUFFY may stem from the highly specialized nature of the Qwen2.5-Math-7B model. Its pre-training is heavily focused on the mathematical domain, which can act as a confounding factor and create a performance ceiling when evaluating on a distant scientific domain.
>
> To test this hypothesis and provide a clearer comparison, we have conducted a new set of experiments on a general-purpose model, Qwen2.5-7B-Base. The results are presented below.
>
> > Table: OOD Performance on the General-Purpose Qwen2.5-7B Base Model
>
> | Model | GPQA-Diamond (pass@1, %) |
> | :--- | :--- |
> | Vanilla GRPO | 33.3 |
> | LUFFY | 34.4 |
> | Scaf-GRPO (Ours) | 35.8 |
>
> These new results on the general-purpose model demonstrate a clear advantage for Scaf-GRPO over both LUFFY and Vanilla GRPO. This supports our hypothesis that by encouraging the model to reason from first principles with minimal hints, Scaf-GRPO fosters slightly more robust and transferable reasoning skills.
>
> Thank you again for your insightful question. We have incorporated these new findings into the revised manuscript to provide a more complete picture of Scaf-GRPO's generalization capabilities. Please refer to lines 489-496 and Table 6.

---

> ### Author Response · Authors · 2025-11-21
> **Response to Weakness 3 (3/5)**
>
> ## Response to Weakness 3: Training efficiency and fairness of comparison
>
> Thank you for raising these important points regarding the training efficiency and fairness of Scaf-GRPO. We address your questions by providing: (1) data on the trigger proportion of our guidance mechanism, (2) a direct comparison of the overall performance under a fixed computational budget, and (3) a broader efficiency comparison with off-policy alternatives.
>
> ### (1) Low Trigger Proportion for Hint-Guided Exploration
>
> We use the training of Qwen2.5-Math-7B as a concrete example. During this training, with a batch size of 256 problems per step, an average of 44.64 problems per batch triggered hint-guided exploration after failing their initial attempt. This corresponds to **17.4%** of the problems in an average batch.
>
> Crucially, this indicates that the overhead of re-generating sequences with hints is limited to only a small fraction of the samples. Consequently, the additional computational cost remains very low.
>
> ### (2) Performance Advantage Within a Fixed Computational Budget: Scaf-GRPO vs. Vanilla GRPO
>
> Under a fixed computational budget, Scaf-GRPO demonstrates better performance and training efficiency compared to Vanilla GRPO. The table below presents the final results and the total time required for each method to reach its best-performing checkpoint on the Qwen2.5-Math-7B model.
>
> > Table: Overall Training Efficiency and Performance between Vanilla GRPO and Scaf-GRPO
>
> | Method | Best Avg. Score (%) | Time to Best Checkpoint (8 GPUs) | Peak Memory (GB) |
> | :--- | :---: | :---: | :---: |
> | Vanilla GRPO | 45.2 | ~13 hours | ~72 |
> | Scaf-GRPO (Ours) | 50.9 | ~12 hours | ~73 |
>
> The data clearly shows that Scaf-GRPO not only achieves a substantially **higher score (50.9% vs. 45.2%)** but does so in **less total time (~12 hours vs. ~13 hours)**.
>
> The reason for this enhanced efficiency is that Scaf-GRPO makes a strategic trade-off. Although its per-step computation time can be slightly longer in certain cases, it successfully converts challenging, zero-reward steps into valuable learning opportunities. By obtaining more effective learning signals from each step, Scaf-GRPO ensures that the overall training process is more productive, leading to faster convergence and a significantly better final result.
>
> We are very grateful for your comment! We have incorporated these findings into the revised manuscript to provide a more complete picture of Scaf-GRPO's practical efficiency. Please refer to lines 498-507 and Table 5.

---

> ### Author Response · Authors · 2025-11-21
> **Response to Weakness 4 (4/5)**
>
> ## Response to Weakness 4: Response on Hyperparameter Design and Justification
>
> We sincerely thank you for your detailed feedback and insightful questions regarding our hyperparameter choices. We address your valuable questions from three perspectives: (1) the justification for the 15% guidance exemption period, supported by a new ablation study; (2) the principled motivation for our hint structure; and (3) the rationale behind our data subsampling strategy.
>
> ### (1) Justification for the Guidance Exemption Period
>
> We have now tested exemption periods of **0%**, **5%**, **10%**, **15%**, **20%**, **40%**, and **100%**. A 0% period means scaffolding is active from the start (as evaluated in Appendix G), while a 100% period is equivalent to the vanilla GRPO baseline, as guidance is never activated. The corresponding results are presented in the table below.
>
> > Table: Ablation Study on the Guidance Exemption Period Duration and its Effect on pass@1 Performance (%)
>
> | Guidance Exemption Period | AIME24 | AIME25 | AMC | Minerva | MATH-500 | Olympiad | Gaokao23 | Avg. |
> | :--- | :---: | :---: | :---: | :---: | :---: | :---: | :---: | :---: |
> | 0% (w/o Phase 1) | 23.3 | 13.3 | 70.0 | 34.2 | 78.4 | 41.2 | 63.1 | 46.2 |
> | 5% | 33.3 | 20.0 | 65.0 | 33.8 | 76.2 | 41.1 | 63.9 | 47.6 |
> | 10% | 40.0 | 16.7 | 70.0 | 35.8 | 79.2 | 42.5 | 63.5 | 50.0 |
> | 15% (Our Method) | 43.3 | 20.0 | 70.0 | 36.4 | 80.0 | 43.3 | 63.4 | 50.9 |
> | 20% | 46.7 | 13.3 | 65.0 | 36.0 | 80.4 | 43.5 | 65.4 | 50.0 |
> | 40% | 43.3 | 13.3 | 70.0 | 34.9 | 78.2 | 41.9 | 64.6 | 49.5 |
> | 100% (Vanilla GRPO) | 30.0 | 13.3 | 60.0 | 33.4 | 75.8 | 41.3 | 62.6 | 45.2 |
>
> The results lead to a clear and insightful conclusion about our method's dynamics:
>
> 1.  **A short initial learning phase is insufficient**. The 0% and 5% settings yield the lowest scores among all scaffolded variants (46.2 and 47.6, respectively). This confirms our core hypothesis: applying guidance prematurely fosters over-reliance and prevents the model from learning to overcome "pseudo-hard" problems (e.g., minor formatting errors or simple reasoning steps) on its own. An initial autonomous phase is crucial for building robust, independent reasoning skills.
>
> 2.  **Relying solely on autonomous learning is ineffective**. The 100% setting, which is equivalent to vanilla GRPO, yields the worst overall performance (45.2). This demonstrates the reality of the "learning cliff": without scaffolding, the model gets permanently stuck on "true-hard" problems, receiving a persistent zero-reward signal that stalls its progress.
>
> 3.  **Scaf-GRPO is robust across a wide effective range**. Crucially, once the initial exemption period is sufficient (≥10%), our method consistently achieves high performance. As shown in the table, the average scores for periods from 10% to 40% remain in a stable, high-performance plateau (ranging from 49.5 to our peak of 50.9). This demonstrates that our framework is not overly sensitive to this specific hyperparameter. The key mechanism is that after an initial period of learning, the set of "true-hard" problems for the model has stabilized. Activating our scaffolding mechanism at any point within this wide window allows the model to effectively start learning from these previously intractable problems.

---

> ### Author Response · Authors · 2025-11-21
> **Response to Weakness 4 Continue (4/5)**
>
> ### (2) Principled Motivation for the Hint Structure
>
> Our design is not arbitrary but is grounded in established pedagogical theories and practical, data-driven considerations. We will explain our rationale for the three-tier hierarchy and the requirement of at least four hint levels as follows:
>
> 1.  **Why three categories (Knowledge → Planning → Solution)**?
>     This hierarchy is directly inspired by pedagogical scaffolding principles [1], which advocate for providing support that moves from conceptual understanding to procedural execution. It also mirrors cognitive frameworks like Bloom's Taxonomy[1], which progress from foundational knowledge to higher-order skills like application and synthesis.
>     *   **Knowledge**: Provides the foundational facts or theorems required (analogous to Remembering/Understanding).
>     *   **Planning**: Outlines the strategic path forward, requiring the model to apply its knowledge (analogous to Applying/Analyzing).
>     *   **Solution**: Offers a direct, concrete step as a final fallback, demonstrating the procedure when the model cannot synthesize it on its own.
>
>     This progressive structure compels the model to first engage with abstract concepts before resorting to concrete steps, fostering more generalizable reasoning skills.
>
> 2.  **Why must each category have ≥4 items/levels**?
>     This design choice is a carefully considered trade-off between dataset characteristics, computational cost, and learning efficacy.
>     *   **Data-Driven Granularity**: Analysis of the training dataset showed that approximately **30% of the step-by-step solutions comprise four or more distinct logical steps**. Consequently, a requirement for at least four hint levels guarantees a sufficiently fine-grained structure to guide the model through multi-step problems in a progressive manner, while avoiding disclosure of the complete solution prematurely.
>     *   **Computational Cost**: The progressive hint search is performed for every "true-hard" problem. Providing hints in four incremental stages strikes a balance: it offers sufficient granularity for learning, but avoids the excessive computational overhead that would arise from a much finer-grained (e.g., 10-level) search.
>     *   **Empirical Validation of Incrementality**: Most importantly, our ablation study in following table ("w/o Incremental Chunking") on Qwen2.5-Math-7B directly validates this incremental approach. When we provided the entire hint content at once instead of across four progressive levels, the model's pass@1 performance dropped significantly by 6.3%. This result confirms that breaking hints into progressive chunks is critical for preserving model autonomy and preventing over-reliance.
>
> 	> Table: Ablation Study on the Effect of Incremental Hint Chunking
>
> 	| Hint Strategy | AIME24 | AIME25 | AMC | Minerva | MATH-500 | Olympiad | Gaokao23 | Avg. |
> 	| :--- | :---: | :---: | :---: | :---: | :---: | :---: | :---: | :---: |
> 	| Scaf-GRPO (Full, Incremental) | 43.3 | 20.0 | 70.0 | 36.4 | 80.0 | 43.3 | 63.4 | 50.9 |
> 	| w/o Incremental Chunking | 43.3 | 10.0 | 62.5 | 36.0 | 76.0 | 41.6 | 64.2 | 47.7 |
>
>     As shown, removing the incremental chunking leads to a 3.2-point drop in the average score, a relative performance decrease of 6.3%. This result provides strong empirical evidence that breaking hints into progressive, smaller pieces is critical for effective learning.
>
>     Thus, our hint structure is a synthesis of pedagogical theory, data analysis, and empirical validation, designed to provide the most effective and efficient guidance.

---

> ### Author Response · Authors · 2025-11-21
> **Response to Weakness 4 Continue (4/5)**
>
> ### (3) Justification for Data Selection and Filtering Strategy
>
> To address the concern regarding our data subsampling (50%), we first clarify our categorization criteria and then present a comprehensive ablation study comparing **0%**, **50%**, and **100%** subsampling ratios.
>
> **1. Definition of Problem Categories** (based on 8 sampling attempts):
> *   **Too Easy (Discarded)**: Solved 8/8 times. These provide minimal gradient signal and are computational waste.
> *   **Potentially Solvable (Subsampled)**: Solved 1-7/8 times. These represent the model's current "comfort zone."
> *   **Too Hard (Retained 100%)**: Solved 0/8 times. These are the primary targets for Scaf-GRPO (the "Learning Cliff").
>
> **2. Ablation on Subsampling Ratios**
> We compared four dataset compositions on Qwen2.5-Math-7B to validate our strategy:
>
> > Table: Impact of Dataset Composition and Difficulty Distribution
>
> | Dataset Setting | Scaf-GRPO |
> | :--- | :---: |
> | Full Dataset | 46.2 |
> | w/o "Too Easy" | 48.8 |
> | w/o "Too Easy" & w/o 50% Solvable (Ours) | 50.9 |
> | w/o "Too Easy" & w/o 100% Solvable | 45.2 |
>
> As shown in the table, our 50% subsampling strategy yields the optimal performance.The reason is simple: balance. We need to keep some solvable problems to ensure the training remains stable. However, if we keep too many, the model will not focus enough on the hard problems that are truly important for improvement.
>
> We are very grateful for your comment! We have incorporated these findings into the revised manuscript to provide a rigorous basis for Scaf-GRPO's hyperparameter selection. Specifically, please refer to lines 403-410 and Appendix G.1 for the analysis of the Guidance Exemption Period, and lines 479-488 and Appendix G.2 for the justification of our Data Selection and Filtering Strategy.
>
> **References:**
>
> [1] Krathwohl, D. R., Bloom, B. S., & Masia, B. B. (1964). Taxonomy of educational objectives: The classification of educational goals. Handbook II: Affective domain. David McKay Company.

---

> ### Author Response · Authors · 2025-11-21
> **Response to Weakness 5 (5/5)**
>
> ## Response to Weakness 5: On the Significance of Performance Gains over LUFFY
>
> Thank you for your insightful comment and for the careful analysis of our results. To clarify the significance of our method's performance gains, we would like to present a more detailed breakdown.
>
> While the absolute gains over LUFFY on a few benchmarks may appear modest at first glance, we believe a more comprehensive analysis reveals the significance of our method. The key is to evaluate the *gain relative to the vanilla GRPO baseline*, which measures how effectively each guidance method overcomes the learning cliff.
>
> To illustrate this, we present a detailed breakdown of the performance gains for the Qwen2.5-Math-7B model:
>
> > Table: Detailed Performance and Relative Gain Analysis of Scaf-GRPO, LUFFY, and GRPO on the Qwen2.5-Math-7B Model.
>
> | Metric (on Qwen2.5-Math-7B) | AIME 24 | AIME 25 | AMC | Minerva | MATH-500 | Olympiad | Gaokao2023en | Avg. |
> | :--- | :---: | :---: | :---: | :---: | :---: | :---: | :---: | :---: |
> | GRPO | 30.0 | 13.3 | 60.0 | 33.4 | 75.8 | 41.3 | 62.6 | 45.2 |
> | LUFFY  | 33.3 | 16.7 | 62.5 | 33.8 | 75.2 | 41.7 | 62.7 | 46.6 |
> | Ours (Scaf-GRPO) | 43.3 | 20.0 | 70.0 | 36.4 | 80.0 | 43.3 | 63.4 | 50.9 |
> | Ours - LUFFY (Gain) | +10.0 | +3.3 | +7.5 | +2.6 | +4.8 | +1.6 | +0.7 | +4.3 |
> | Ours - GRPO (Gain) | +13.3 | +6.7 | +10.0 | +3.0 | +4.2 | +2.0 | +0.8 | +5.7 |
> | LUFFY - GRPO (Gain) | +3.3 | +3.4 | +2.5 | +0.4 | -0.6 | +0.4 | +0.1 | +1.4 |
> | Gain Ratio (Ours-GRPO / LUFFY-GRPO) | 4.03x | 1.97x | 4.00x | 7.50x | N/A | 5.00x | 8.00x | 4.07x |
>
> From this table, we would like to highlight two key points:
>
> ### (1) Scaf-GRPO Achieves Larger Gains on Challenging Benchmarks:
>
> As you noted, the absolute gains of Scaf-GRPO over LUFFY on Gaokao (+0.7), Olympiad (+1.6), and Minerva (+2.6) are smaller than on other benchmarks. However, this is because LUFFY itself provides almost no improvement over the GRPO baseline on these specific datasets (+0.1, +0.4, and +0.4, respectively). In sharp contrast, our method’s improvement over the baseline is 8.0x, 5.0x, and 7.5x greater than LUFFY's (Row 7). This demonstrates that on the very problems where prefix-based guidance fails, our in-prompt scaffolding is substantially more effective.
>
> ### (2) Potential OOD Nature of Benchmarks:
> We hypothesize that the overall smaller gains for *all* RL methods on benchmarks like Gaokao, Minerva, and Olympiad are due to their partial out-of-distribution (OOD) nature. Our training data primarily consists of AIME, AMC, MATH, and Omni-MATH datasets. The OlympiadBench, for instance, is not purely mathematical but also includes physics problems, making it a challenging test of generalization. The fact that Scaf-GRPO shows a much stronger relative improvement even in these difficult OOD scenarios further underscores the robustness of our approach.
>
> In summary, a deeper analysis of the relative gains shows that Scaf-GRPO consistently and significantly outperforms LUFFY, particularly on challenging benchmarks where existing methods struggle. We hope this clarifies the contribution and thank you again for your valuable feedback.

---

> ### Comment · Reviewer_KDuR · 2025-11-27
> **Thank you for the response**
>
> Thanks for the response. While most of my issues are resolved, two issues remain:
>
> 1. While I acknowledge that the policy is indeed 'on-policy' with respect to the augmented prompt, I am concerned that the curriculum learning mechanism effectively alters the original task distribution. Could the authors explain how they address or interpret this potential mismatch?
>
> 2. Comparison with Related Work: Upon further reading, I noticed that Scaf-GRPO lacks a discussion of several highly relevant works [1-3]. Could the authors clarify the distinctions between their proposed method and these existing approaches?
>
> In light of the improvements, but considering these remaining concerns, I have raised my score to 4.
>
> [1] Training Large Language Models for Reasoning through Reverse Curriculum Reinforcement Learning. ICML 2024
>
> [2] Thought-Augmented Policy Optimization: Bridging External Guidance and Internal Capabilities. arXiv:2505.15692
>
> [3] UFT: Unifying Supervised and Reinforcement Fine-Tuning. NeurIPS 2025

---

> > ### Author Response · Authors · 2025-11-28
> > **Response to follow-up questions**
> >
> > Dear Reviewer KDuR:
> >
> > We sincerely thank you for your follow-up feedback. We are glad to hear that our previous responses have addressed your initial questions. For your follow-up concerns, we have actively provided further clarification and explanation.
> >
> > As an overview:
> >
> > - For **Q1**, we clarify that the curriculum acts as a temporary scaffolding mechanism that fosters capability internalization rather than a permanent distribution shift. We provide evidence via **"Skill Graduation" analysis (Table 4)** to show the model effectively transitions from hint-reliance to autonomous solving, and confirm this via our superior performance on the original **unhinted benchmarks (Table 1)**.
> > - For **Q2**, we provide a detailed comparison with **$R^3$, TAPO, and UFT**. We highlight Scaf-GRPO's unique advantages in terms of **adaptive, failure-triggered guidance** and **hierarchical hint structure**. We also report new experimental results demonstrating that Scaf-GRPO consistently outperforms these baselines.
> >
> >
> > We hope these responses (in the following message blocks) can fully resolve your concerns. Meanwhile, we remain ready to respond to any further questions or feedback promptly.
> >
> > Best regards,
> >
> > All Authors of Submission #6005

---

> > ### Author Response · Authors · 2025-11-28
> > **Response to follow-up question 1: Interpreting the Task Distribution Shift**
> >
> > We address the concern about potential task mismatch through two parts: (1) demonstrating that the model effectively transitions from hint-reliance to autonomous solving ("skill graduation"), and (2) confirming that the final policy achieves superior performance on the original, unhinted task distribution.
> >
> > ### **1. Internalizing capabilities: The model learns to solve the original task.**
> >
> > Our goal is for the model to use the hinted task to learn the underlying reasoning logic, which it then applies to the original task. We visualize this transfer in **Table 4** of our paper under the metric "Skill Graduation."
> >
> > A "graduation" occurs when a problem that initially required a hint is later solved by the model autonomously without any hints. This proves the model is not overfitting to the hints but is using them to master the original problems.
> >
> > For convenience, the graduation statistics from **Table 4** are excerpted below:
> >
> > > Table: Comparison of total "graduations" across different model backbones
> >
> >  | Model | Vanilla | Scaf-GRPO (Ours) | Relative Increase |
> >  | :--- | :---: | :---: | :---: |
> >  | Qwen2.5-Math-1.5B | 1,123 | 2,670 | +137.8% |
> >  | Qwen2.5-Math-7B | 434 | 483 | +11.3% |
> >  | Qwen2.5-7B | 464 | 723 | +55.8% |
> >  | DeepSeek-R1-Distill-1.5B | 453 | 694 | +53.2% |
> >  | Llama-3.2-3B-Instruct | 577 | 986 | +70.9% |
> >
> > The data shows that Scaf-GRPO significantly increases the number of tasks the model eventually masters autonomously.
> >
> > ### **2. Superior performance on the original distribution.**
> >
> > The ultimate validation lies in our results on standard benchmarks (**Table 1**), which are evaluated on the original, unhinted distribution. Scaf-GRPO achieves consistent gains over vanilla GRPO, such as a **44.3% relative improvement** on AIME 24 with Qwen2.5-Math-7B. This confirms that our temporary scaffolding effectively transfers to the original task, resulting in a robust, autonomous reasoner.

---

> ### Author Response · Authors · 2025-11-28
> **Response to follow-up question 2: Comparison with $R^3$, TAPO, and UFT**
>
> We address this comparison from two perspectives: (1) Methodological Distinctions and (2) Performance Analysis.
>
> ### 1. Methodological Distinctions
>
> The following table summarizes the core differences. Scaf-GRPO is the unique method that combines **on-policy simplicity** (no loss modification) with **instance-wise adaptivity** (minimal necessary guidance).
>
> | Feature | **$R^3$ [1]** | **TAPO [2]** | **UFT [3]** | **Scaf-GRPO (Ours)** | **Advantage of Scaf-GRPO (Evidence)** |
> | :--- | :--- | :--- | :--- | :--- | :--- |
> | **Guidance Selection** | **Fixed Stages.** Samples start states via a fixed reverse curriculum or uniform distribution. | **Static Retrieval.** Matches problems to a fixed library based on static complexity metrics. | **Global Schedule.** Determines hint length via a fixed cosine annealing schedule (Time-dependent). | **Adaptive & Minimal.** Monitors the "Learning Cliff" and triggers **only upon failure**. Dynamically searches for the **minimal** hint needed to solve the specific instance. | **Prevents Dependency.** Fixed schedules provide help even when unneeded. **In Table 2**, removing "Incremental Chunking" (forcing minimal help) causes a **3.2% drop** (50.9% vs 47.7%), proving adaptivity is key. |
> | **Hint Content** | **Solution Prefix.** Concrete slices of the ground truth trajectory. | **Thought Templates.** Abstracted high-level reasoning patterns. | **Solution Prefix.** Concrete slices of the ground truth trajectory. | **Hierarchical.** Multi-level scaffolding: **Knowledge** (Abstract) $\to$ **Planning** (Strategic) $\to$ **Solution** (Concrete). | **Better Generalization.** Baselines rely on concrete prefixes or single-level patterns. In **Table 2**, using only concrete "Solution Hints" (similar to $R^3$/UFT prefixes) underperforms our full Hierarchical approach (**48.4% vs 50.9%**). |
> | **Algorithm Type** | **On-Policy.** Standard RL. | **On-Policy.** Standard RL. | **Hybrid.** Adds SFT loss on hints + RL reward. | **On-Policy.** Standard GRPO; no loss modification. | **Simplicity & Robustness.** Unlike hybrid methods (e.g., UFT) that require careful tuning of coefficients to balance SFT and RL losses, Scaf-GRPO preserves the pure RL objective, avoiding complex hyperparameter searches and ensuring optimization stability. |
>
>
> ### 2. Performance Analysis
>
> To demonstrate the empirical advantage of our approach, we present the comprehensive performance of **Scaf-GRPO** on the **Qwen2.5-1.5B-Math** model across all benchmarks.
>
> As shown in following table, Scaf-GRPO achieves consistent improvements compared with other methods.
>
> > Table: Performance of Qwen2.5-1.5B-Math (Pass@1 %)
>
> | Method | **AIME 24** | **AMC23** | **Minerva** | **MATH-500** | **Olympiad** | **Gaokao23en** | **Avg.** |
> | :--- | :---: | :---: | :---: | :---: | :---: | :---: | :---: |
> | **$R^3$ [1]** | 10.0 | 52.5 | 24.6 | 65.4 | 31.2 | 52.8 | 39.4 |
> | **UFT [3]** | 13.3 | 56.5 | 26.8 | 70.6 | 32.2 | 55.5 | 42.4 |
> | **TAPO [2]** | 16.7 | 55.0 | 31.6 | 69.0 | 33.6 | 54.8 | 43.5 |
> | **Scaf-GRPO (Ours)** | **20.0** | **60.0** | **29.1** | **73.4** | **36.6** | **57.9** | **46.2** |

---

### Official Review · Reviewer_HrXm · 2025-10-28

**Soundness:** 3
**Presentation:** 3
**Contribution:** 3
**Rating:** 6
**Confidence:** 3

**Summary:**

This paper studies the "learning cliff" issue in GRPO, where the solution to a hard problem cannot be learned if all the sampled solution are incorrect. To overcome this issue, this paper propose a scaffolding framework, which provides some hints of different level of concreteness to stimulate on-policy correct answers. These correct answers avoids zero-gradient on the hard problem and provide useful training signal, therefore makes use of the hard problems. Experiments shows that the proposed method outperforms vanilla GRPO and other off-policy guidance baselines.

**Strengths:**

The strengths of this paper is listed as follows

1. The method proposed in this paper adapts external knowledge (solution) to on-policy solutions, which avoid the distribution shift resulted from directly utilizing off-policy solution.

2. The experiments are conducted on several different models and Scaf-GRPO consistently improves over baselines. Also, the ablation study is comprehensive and covers most of the design of proposed method.

3. Overall, the paper is clearly written.

**Weaknesses:**

In general, I think this paper does not demonstrate any major weakness. Some of the identified weakness and my questions are listed below

1. In equation (4), the author propose to use $\pi_{\theta}(\cdot|q\oplus h*)/\pi_{\text{old}}(\cdot|q\oplus h*)$ as the importance ratio. However, this does not exactly matches the probility $\pi_{\theta}(\cdot|q)$. To the reviewer, $\pi_{\theta}(\cdot|q)/\pi_{\text{old}}(\cdot|q\oplus h*)$ makes more sense. Could the authors compare these two different approaches?

2. The method relies on LLM-generated hints, whose quality might affect the performance of the method. Besides end-to-end performance, are there any other metric that measures the quality of the generated hints? Or in other word, is there a way to control the quality of generated hints?

**Questions:**

See weakness section

---

> ### Author Response · Authors · 2025-11-21
> **Response to Weakness 1 (1/2)**
>
> We sincerely thank you for your insightful and constructive review. We are greatly encouraged that you recognized the core strength of our method in adapting external knowledge while avoiding distribution shift, the comprehensiveness of our experimental validation, and the overall clarity of our paper. Below, we respond to each point in detail.
>
> ## Response to Weakness 1: Justification for the Probability Ratio Formulation in Scaf-GRPO
> Thank you for your insightful question. To clarify the rationale behind our design, we compare our on-policy formulation with your suggested off-policy alternative from three perspectives: (1) the theoretical goal of on-policy stability, (2) the potential instability from the distributional mismatch in the alternative, and (3) an empirical validation of their impact on training.
>
> ### (1) Theoretical Rationale: Preserving On-Policy Stability
>
> The core objective of the probability ratio `r_t(θ)` in GRPO is to measure the divergence between the new and old policies for a given sample. The clipping mechanism, `clip(r_t(θ), 1-ε, 1+ε)`, uses this ratio to prevent destructively large updates. When the ratio falls within the stable interval `[1-ε, 1+ε]`, the full policy gradient update is utilized. However, if the ratio falls outside this interval, it signifies a significant policy shift, and the update is "clipped" to maintain training stability.
>
> Our proposed method, Scaf-GRPO, is designed to maximize the effectiveness of this mechanism. When a "learning cliff" is encountered, we do not import a trajectory from a different policy. Instead, we augment the *input* to our current policy `π_θ`, changing the prompt from `q` to `q ⊕ h*`. The model then generates a trajectory on-policy based on this new, hint-augmented prompt.
>
> Consequently, our probability ratio for a guided trajectory is:
>
> $$
> r_{t}(\theta) = \frac{\pi_{\theta}(\cdot | q \oplus h^{\ast})}{\pi_{\theta_{old}}(\cdot | q \oplus h^{\ast})}
> $$
>
> This is a standard on-policy ratio. It compares the probability of generating an output under the new and old policies, both conditioned on the exact same input (`q ⊕ h*`). Because the input context is identical for both the numerator and the denominator, the distributions remain closely aligned. This increases the likelihood that the ratio will fall within the stable `[1-ε, 1+ε]` range, allowing for smooth and effective learning updates.
>
> ### (2) Analysis of the Suggested Off-Policy Alternative
>
> Your suggested formulation is:
>
> $$
> r_{t, suggested}(\theta) = \frac{\pi_{\theta}(\cdot | q)}{\pi_{\theta_{old}}(\cdot | q \oplus h^{\ast})}
> $$
>
> This represents an off-policy correction. However, it introduces a fundamental distributional mismatch by comparing probabilities from two different input conditions. The numerator is conditioned on the original prompt `q`, while the denominator is conditioned on the hint-augmented prompt `q ⊕ h*`. Since the hint `h*` is specifically chosen to alter the probability landscape and guide the model toward a correct solution, the distributions `π_θ(·|q)` and `π_θ_old(·|q ⊕ h*)` are inherently different.
>
> This mismatch makes the resulting ratio `r_t(θ)_suggestion` more likely to fall outside the stable non-clipping interval `[1-ε, 1+ε]`. A high rate of clipping, whether on the upper or lower bound, systematically truncates the learning signal and is a key contributor to training instability.

---

> ### Author Response · Authors · 2025-11-21
> **Response to Weakness 1 Continue (1/2)**
>
> ### (3) Empirical Validation: Training Stability and Performance
>
> To validate our theoretical analysis, we conducted an experiment implementing your suggested probability ratio and compared it against our proposed method.
>
> First, we measured the `clip_ratio`, which is the fraction of tokens whose probability ratios fall outside the `[1-ε, 1+ε]` range and are therefore clipped during the update. A high `clip_ratio` is a direct indicator of training instability, as it means the learning signal is frequently being constrained.
>
> As shown in the provided figure, our on-policy Scaf-GRPO maintains a very low and stable `clip_ratio` throughout training. In contrast, your suggested method exhibits a dramatically higher and more volatile `clip_ratio`. This confirms our hypothesis that the distributional mismatch leads to unstable policy updates.
>
> > Figure: Comparison of `clip_ratio` during training. Our proposed on-policy method (x) maintains a consistently low clip ratio. Your suggested off-policy formulation (o) results in a higher and more volatile clip ratio, indicating training instability. A lower value is better.
>
> ```
>                                         clip_ratio
>
> 0.4 -|                                     o
>      |                                    o
>      |                                            o
>      |                                           o o   o
>      |                                          o   o o
> 0.3 -|                                         o
>      |                                     o
>      |                                o     o
>      |                              o
>      |                           o o   o o
> 0.2 -|                         o   o   o
>      |                        o
>      |                      o
>      |                    o
>      |                  o
> 0.1 -|                o
>      |              o
>      |           o
>      |         o
>      |   o x x-x-x       x-x-x-x-x-x-x-x-x-x-x-x-x-x-x-x-x-x-x-x-x-x-x--
> 0.0 -|-o-x-x      x-x-x-x
>      +--------------------------------------------------------------------->
>        150        200        250        300                       Step
>
>                  Legend: x is ours,  o is reviewer suggested
> ```
>
>
>
> This training instability directly translates to inferior final performance. As shown in the table below, the model trained with the suggested off-policy ratio suffers a noticeable performance degradation across key evaluation benchmarks compared to our stable, on-policy approach.
>
> > Table: Pass@1 (%) performance comparison between Vanilla GRPO, your Suggested off-policy method, and our Scaf-GRPO on Qwen-2.5-Math-7B.
>
> | Method | AIME24 | AIME25 | AMC | Minerva | MATH-500 | Olympiad | Gaokao23en | Avg. |
> | :--- | :---: | :---: | :---: | :---: | :---: | :---: | :---: | :---: |
> | Vanilla GRPO | 30.0 | 13.3 | 60.0 | 33.4 | 75.8 | 41.3 | 62.6 | 45.2 |
> | Your Suggestion | 36.7 | 13.3 | 65.0 | 34.2 | 78.2 | 38.5 | 65.2 | 47.3 |
> | Scaf-GRPO (Ours) | 43.3 | 20.0 | 70.0 | 36.4 | 80.0 | 43.3 | 63.4 | 50.9 |
>
>
>
> We thank you again for prompting this valuable analysis. In summary, while your suggested formulation is a thoughtful alternative, our empirical results confirm that the induced off-policy distributional mismatch leads to training instability (a higher `clip_ratio`) and ultimately limits performance. Our on-policy approach is designed specifically to maintain stability for more effective learning. We believe this is a crucial clarification and have added a summary of this analysis to the revised manuscript. Please refer to lines 265-269 and Appendix F.5.

---

> ### Author Response · Authors · 2025-11-21
> **Response to Weakness 2 (2/2)**
>
> ## Response to Weakness 2: On the Quality and Control of LLM-Generated Hints
>
> Thank you for raising thess questions. We address the quality control of our generated hints from two perspectives: (1) assessment using a multi-faceted rubric, and (2) empirical validation linking hint quality to model performance.
>
> ### (1) Assessment Methodology: A Multi-Faceted Rubric
>
> In our paper, the final hints come from the DeepSeek-R1 model. Our selection process involves experimenting with other powerful models, such as Qwen2.5-72B-Instruct, to generate the hints. To objectively compare the quality of the generated hint sets, we employ a multi-faceted rubric. This rubric is designed to assess hints not just for correctness, but for their ability to provide minimal, clear, and structured guidance.
>
> Our rubric assesses hints across four key dimensions:
> *   **Accuracy**: Factual and logical correctness.
> *   **Minimality**: Providing the least necessary support to preserve autonomy.
> *   **Clarity**: Unambiguous and easy-to-parse language.
> *   **Structural Coherence**: A logical flow from abstract to concrete.
>
>
> To apply this rubric, we use an LLM Judge (Gemini-2.5-Pro) to automatically score all the hints. This method allows us to compare the quality of hints generated by different models, such as DeepSeek-R1 (our final choice) and Qwen2.5-72B-Instruct. The quality scores are shown below:
>
> > Table: LLM Judge scores for hint quality from different generator models.
>
> | Hint Generator Model | LLM Judge Score |
> | :--- | :---: |
> | Qwen2.5-72B-Instruct | 4.38 |
> | DeepSeek-R1 (Used) | 4.72 |
>
> ### (2) Empirical Validation: Hint Quality vs. Model Performance
>
> Next, to validate if better hint scores lead to better model performance, we train the Qwen2.5-Math-7B model on both sets of hints, using the exact same training settings. The final test results show a clear link to our hint quality scores:
>
> > Table: Final performance (Pass@1 %) of Qwen2.5-Math-7B trained on hints from different sources.
>
> | Model & Hint Source | AIME24 | AIME25 | AMC |Minerva| MATH-500 | Olympiad | Gaokao23 | Avg. |
> | :--- | :---: | :---: | :---: | :---: | :---: | :---: | :---: |  :---: |
> | Qwen2.5-Math-7B (w/ Qwen Hints) | 36.7 | 20.0 | 67.5 | 33.8 | 79.4 | 42.2 | 63.1 | 49.0 |
> | Qwen2.5-Math-7B (w/ DeepSeek-R1 Hints) | 43.3 | 20.0 | 70.0 |36.4| 80.0 | 43.3 | 63.4 | 50.9 |
>
> As you can see from the results, there is a clear connection between the hint quality score from our rubric and the final performance of the trained model. This shows that the hint set from DeepSeek-R1 is better. This also suggests that using an LLM-based judge with our rubric is a good proxy for measuring hint quality.
>
> Thank you again for this very insightful question. We have added the detailed hint quality comparison, the specific prompt used for the LLM judge, and these experimental results to the revised manuscript. Please refer to lines 431-448 and Appendix H. We believe this makes our methodology more transparent and our findings more complete.

---

> ### Comment · Reviewer_HrXm · 2025-11-21
>
> I appreciate the authors' efforts to addressing my questions and would like to keep my positive rating.

---

> > ### Author Response · Authors · 2025-11-27
> > **Thanks for your response and recognition**
> >
> > Dear Reviewer HrXm:
> >
> > We greatly appreciate your recognition and are thankful for your contributions to the academic community.
> >
> > Best regards,
> >
> > All Authors of Submission #6005

---

### Official Review · Reviewer_QtTi · 2025-10-29

**Soundness:** 4
**Presentation:** 4
**Contribution:** 4
**Rating:** 8
**Confidence:** 2

**Summary:**

This work is motivated by the observation that GRPO could perform badly when the problem is hard to solve and successful trajectories are hard to collect. To address this challenge, this work proposes Scaf-GRPO, a method that focuses on the data augmentation aspect rather than modifying the GRPO algorithm itself. Specifically, Scaf-GRPO encourages the model to generate hints that improve the success rate, thereby accelerating RL convergence.

**Strengths:**

1. The paper is well written and well organized.
2. Sufficient experiments are provided to validate the effectiveness of their method.

**Weaknesses:**

see questions

**Questions:**

1. During RL training, is Scaf-GRPO applied to both easy and hard problems, or only to the hard ones? Additionally, how do you determine whether a problem is considered easy or hard?
2. What is the behavior during inference? Do you use the hint prompt to encourage the model to generate hints for all problems?
3. In Line 398, could you explain what is meant by “the most concrete hint”?

---

> ### Author Response · Authors · 2025-11-21
> **Response to Question 1 (1/3)**
>
> Thank you for your thoughtful and constructive feedback. We are encouraged that you found our paper to be well-written, well-organized, and supported by sufficient experiments. Below, we respond to each point in detail.
>
> ## Response to Question 1: Application Conditions of Scaf-GRPO and the Definition of Problem Difficulty
>
> Thank you for raising these questions. We provide the following clarifications from two key perspectives: (1) the dynamic definition of problem difficulty, and (2) the conditional intervention mechanism, to address your concerns in detail:
>
> ### (1) How We Define "Hard" vs. "Easy" Problems: A Dynamic, Performance-Based Method
>
> We do not use a fixed list of "easy" or "hard" problems. Instead, our system determines a problem's difficulty dynamically by observing how well the model performs on it during the training process. For each single problem, the model makes several attempts to solve it (for example, 8 attempts, which are called "rollouts" in the paper).
>
> Based on the results of these attempts, we classify the problem as follows:
>
> *   **"Easy" or "Solvable" Problem**: A problem is considered easy if the model produces at least one correct solution out of its multiple attempts. This single success gives the learning algorithm a positive reward, which is like a clear signal telling it "this is a good path." This signal is all the standard learning algorithm (GRPO) needs to learn from the successful attempt. In this situation, Scaf-GRPO does not get involved. As we state in our paper (lines 251-254):
>     > In the first case, where at least one successful trajectory is generated initially... the batch already contains a valid learning signal. The condition for intervention is not met... our framework does not interfere when the model can learn on its own.
>
> *   **"Hard" Problem (The "Learning Cliff")**: A problem is identified as hard when the model gets completely stuck, a situation we call the "learning cliff." This happens when all of the model's attempts to solve the problem are wrong. Because every attempt fails, the model gets a reward of zero every time. Without any successful examples, the learning algorithm has no information on how to improve. As a result, the algorithm cannot learn from this problem, and it becomes "invisible" to the training process. As described in our paper (lines 257-265):
>     > In the second case, the learning cliff scenario... standard GRPO fails. The uniform zero rewards cause the advantage calculation to collapse... leading to a null advantage... and a vanishing policy gradient. Here, Scaf-GRPO intervenes...
>
> ### (2) How Scaf-GRPO Intervenes: Providing Help Only for Hard Problems
>
> Scaf-GRPO is designed to help the model solve "hard" problems it is stuck on. It intervenes specifically when a "hard" problem has been detected, indicated by all attempts yielding zero reward. The framework then provides minimal hints to produce one successful trajectory, which restores the learning signal and allows the model to move forward.
>
> This process, detailed in Figure 3 of our paper, unfolds as follows:
>
> *  **Detection**: The framework monitors the rewards for a batch of rollouts generated for a given problem.
> *  **Decision & Action**:
>     *   If there is any non-zero reward, the process follows the standard GRPO path (Figure 3, left), and no scaffolding is applied.
>     *   If all rewards are zero, the "learning cliff" is detected. This triggers Scaf-GRPO's Hierarchical Hint-Guided Exploration (Figure 3, right).
> *  **Resolution**: The framework then injects minimal, progressive hints into the prompt to help the model generate a single successful trajectory. This new, successful trajectory replaces one of the failed ones in the batch, restoring a non-zero reward variance and reactivating the advantage calculation. This allows the model to learn from a problem that was previously intractable.

---

> ### Author Response · Authors · 2025-11-21
> **Response to Question 2 (2/3)**
>
> ## Response to Question 2: Clarification on Inference-Time Behavior and Hint Usage
>
> Thank you for your insightful question! Our approach distinguishes strictly between the training and inference phases. We would like to clarify the two points you raised:
>
> - **Inference-Time Behavior**:
> During inference (evaluation), our method **does not use any hints**. The model is presented with only the original, unmodified problem query, exactly like the baseline models. The goal of our training framework, Scaf-GRPO, is to enhance the model's intrinsic reasoning capabilities so that it can solve difficult problems autonomously. Therefore, all performance results reported in our paper (e.g., in Table 1) are based on the model's independent problem-solving ability, ensuring a fair and direct comparison against vanilla GRPO, LUFFY, and other baselines.
> - **Hint Generation for Training**:
> The hierarchical hints are exclusively a training-time scaffolding mechanism. We generate these tiered hints for every problem within our curated training dataset. This is a one-time, offline data preparation step. These hints are then strategically injected into the prompt *only during training* and only when the model demonstrates an inability to solve a problem on its own (i.e., after a "learning cliff" is detected). This process is designed to provide minimal, temporary support to enable learning, not to create a dependency. As shown in Figure 5, the model learns to internalize these reasoning skills, eventually solving problems without any guidance.

---

> ### Author Response · Authors · 2025-11-21
> **Response to Question 3  (3/3)**
>
> ## Response to Question 3: Clarification of "The Most Concrete Hint" and the Progressive Guidance Mechanism
>
> Thank you for your excellent question. In our framework, "the most concrete hint" refers specifically to the Solution Hint ($H_{\text{solution}}$). A detailed explanation is as follows:
>
> The term **"the most concrete hint"** refers to the **Solution Hint ($H_{\text{solution}}$)**, which is the final and most explicit level in our three-tiered guidance hierarchy. It provides a direct, step-by-step calculation or a segment of the ground-truth solution (e.g., providing the direct calculation step "$6a^3+9b^3+32c^3 \ge 36abc$"). Its purpose is to show the model precisely what to do.
>
> This contrasts sharply with the other, more abstract hints:
> *   **Knowledge Hint($H_{\text{knowledge}}$)**: Points to the key underlying concept or formula (e.g., "The AM-GM inequality").
> *   **Planning Hint($H_{\text{planning}}$)**: Outlines a high-level strategic approach to the solution (e.g., "Recognize the expression can be minimized using AM-GM by splitting it into manageable parts").
>
> We have clarified this definition in the revised manuscript; please refer to lines 414-415.

---

### Author Response · Authors · 2025-11-21
**General Rebuttal**

Dear Program Chairs, Senior Area Chairs, Area Chairs, and Reviewers,

We would like to express our sincere gratitude for your thorough and insightful review of our submission, Scaf-GRPO. We deeply appreciate your constructive feedback and for recognizing the strengths of our work.

In particular, we appreciate that all reviewers recognized the importance of the **"learning cliff"** problem: standard RL fails on hard problems because the model consistently answers incorrectly, resulting in zero reward and stalled progress. We thank **Reviewer 9rLe** and **Reviewer KDuR** for recognizing the elegance and intuition of our scaffolding solution, which overcomes this barrier by providing minimal hints that enable the model to reason out the correct solution by itself, thereby restoring the learning signal, and **Reviewer QtTi** and **Reviewer HrXm** for commending the clarity of our presentation and the comprehensiveness of our experimental validation.

We also valued the constructive suggestions that have been instrumental in significantly improving the rigor and clarity of our work. During the rebuttal period, we provided the following responses and substantial additions:

1.  **Clarification on On-Policy Formulation and Theoretical Soundness:** Following the suggestions of **Reviewers HrXm** and **KDuR**, we have provided a detailed theoretical analysis and empirical evidence (including clip ratio comparisons) to demonstrate why our on-policy formulation maintains better training stability compared to off-policy alternatives.
2.  **Quality Control and Scalability of Hint Generation:** In response to **Reviewers QtTi**, **HrXm**, and **9rLe**, we have introduced a multi-dimensional rubric and an LLM-Judge mechanism to quantify hint quality. We also clarified the scalability of our data preparation process using existing datasets.
3.  **Comprehensive Efficiency Analysis:** Addressing the concerns of **Reviewers KDuR** and **9rLe**, we provided detailed data on training time, memory usage, and trigger proportions. The results confirm that Scaf-GRPO achieves better performance with comparable computational resources to baselines.
4.  **Justification of Hyperparameters and Design Choices:** To address questions from **Reviewers KDuR** and **9rLe**, we conducted extensive ablation studies regarding the "guidance exemption period" (verifying the 15% choice) and the hierarchical hint structure, proving the necessity of these design elements.
5.  **Enhanced Evaluation Rigor and Generalization:** Following the advice of **Reviewers KDuR** and **9rLe**, we expanded our evaluation to include Out-of-Distribution (OOD) testing on general-purpose models and adopted the more robust `avg@16` metric to confirm the stability of our performance gains.

We have provided comprehensive details and additional results in our individual responses, which we believe fully address all the reviewers' concerns. We have already incorporated all the above additions into the revised manuscript, and these new analyses continue to support the primary conclusions of our original work.

Once again, we sincerely thank the Program Chairs, Senior Area Chairs, Area Chairs, and Reviewers for your time and dedication in reviewing our submission!

Best,

All Authors of Submission #6005

---

### Meta-Review · Area_Chair_77GC · 2026-01-13

**Summary:**

The main concerns were: (1) Methodological validity - whether hint-augmented prompts introduce a task distribution shift that undermines the "on-policy" claim; (2) Limited performance gains - marginal improvements (1-2%) on strong long-CoT models and underperformance compared to recent methods like QuestA in certain settings; (3) Scalability questions - dependency on high-quality hint generation and inconsistent gains across different model scales, raising doubts about practical applicability beyond the primary tested scenarios.

**Reviewer Concerns:**

Computational efficiency, hint quality control, hyperparameter robustness, and cross-architecture generalization were convincingly resolved with new experiments. Reviewers QtTi and HrXm confirmed satisfaction.

The task distribution shift debate remains conceptually unresolved despite empirical evidence of skill transfer. Marginal gains on strong baselines (DeepSeek-R1-Distill: 1-2%) and incomplete QuestA comparison (different training budgets) leave questions about competitive positioning and performance ceiling unanswered.

**Reviewer Scores:**

QtTi and HrXm would likely maintain their scores (8 and 6) as they explicitly confirmed their concerns were resolved. KDuR would probably stay at 4 given the persistent conceptual disagreement about task distribution shift despite the score increase from 2. 9rLe is most uncertain - their expressed inclination to lower the score from 6 suggests they might drop to 5, though engagement with the authors' context-dependent performance arguments (training budget differences, ceiling effects on strong models) could keep them at 6.

---

### Decision · Program_Chairs · 2026-01-26

Accept (Poster)